# The grain-scale signature of isotopic diffusion in ice

Felix S. L. Ng[1]

[1]Department of Geography, University of Sheffield, Sheffield, UK

*Correspondence*: Felix Ng (f.ng@sheffield.ac.uk)

**Abstract.** Diffusion limits the survival of climate signals on the water stable isotopes in ice sheets. Diffusive smoothing acts not only on annual signals near the surface, but also on long time-scale signals at depth as they shorten to decimetres or centimetres. Short-circuiting of the slow diffusion in crystal grains by fast diffusion along liquid veins can explain the "excess diffusion" found on some ice-core isotopic records. But experimental evidence is lacking whether this mechanism operates as theorised; theories of the short-circuiting also under-explore the role of diffusion along grain boundaries. The nonuniform patterns of isotopic deviation $\delta$ across crystal grains induced by short-circuiting offer a testable prediction of these theories. Here, we extend the modelling for grain boundaries (as well as veins) and calculate these patterns for different grain-boundary diffusivities and thicknesses, temperatures, and vein-water flow velocities. Two isotopic patterns are shown to prevail in ice of millimetre grain size: (i) an axisymmetric "pole" pattern with excursions in $\delta$ centred on triple junctions, in the case of thin, low-diffusivity grain boundaries; (ii) a "spoke" pattern with excursions around triple junctions showing the impression of grain boundaries, when these are thick and highly diffusive. The excursions have widths ~ 10–50 % of the grain radius and variations in $\delta$ ~ $10^{-2}$ to $10^{-1}$ times of the bulk isotopic signal for oxygen and deuterium, which set the minimum measurement capability needed to detect the patterns. We examine how the predicted patterns vary with depth through a signal wavelength to suggest an experimental procedure, based on laser-ablation mapping, of testing ice-core samples for these signatures of isotopic short-circuiting. Because our model accounts for veins and grain boundaries, its predicted enhancement factor (quantifying the level of excess diffusion) characterises the bulk-ice isotopic diffusivity more comprehensively than past studies.

# 1  Introduction

The water stable isotope records ($\delta^{18}$O, $\delta$D) in polar ice cores contain diverse palaeoclimatic signals. Owing to isotopic diffusion in firn and ice, signals at decimetre and centimetre or shorter scales experience pronounced smoothing as they descend the ice column. This postdepositional process limits the integrity and resolution of climatic information at different depths. The smoothing rate needs to be known for recovering the original (e.g. annual) $\delta$-variations at the surface by "back diffusing" an isotopic record (Johnsen, 1977), for reconstructing surface temperatures in the past from spectrally-derived

diffusion lengths (Gkinis et al., 2014), and for predicting how deep climatic signals of different time scales survive into an ice core (e.g. Grisart et al., 2022). The last aspect, which matters particularly for long records, is of major interest to the ongoing ice-coring campaigns at Little Dome C, East Antarctica, which aim to retrieve ice reaching back $\approx$ 1–1.5 Ma; see the Beyond EPICA - Oldest Ice project webpage (https://www.beyondepica.eu/en/) and the Million Year Ice Core project webpage (https://www.antarctica.gov.au/science/climate-processes-and-change/antarctic-palaeoclimate/million-year-ice-core/).

"Excess diffusion" in the ice below the firn is a key concern in this subject. Analysis of the GRIP (Greenland Ice Core Project) ice core by Johnsen et al. (1997, 2000) showed that the annual $\delta^{18}$O signals in the Holocene section of this core decay $\approx$ 10–30 times faster than expected from the self-diffusion rate measured in single ice crystals (Ramseier, 1967), implying a large enhancement of the bulk-ice isotopic diffusivity above the monocrystalline diffusivity. Theories put forward to explain this excess diffusion invoke *short-circuiting* – the idea that, in polycrystalline ice, fast diffusion in the network of liquid veins

(located at triple junctions) and along grain boundaries bypasses the slow diffusion within ice grains to cause the enhancement. After Nye (1998) made pioneering calculations to show that the presence of veins causes excess diffusion by short-circuiting, Johnsen et al. (2000) adapted the firn isotope diffusion model of Whillans and Grootes (1985) to gauge the separate contributions of grain boundaries and veins to the mechanism. Later, Rempel and Wettlaufer (2003) refined Nye's model to account for the finite isotopic diffusivity of the vein water; they calculated the diffusivity enhancement in ice at $\approx$ –32 °C as a

function of signal wavelength, grain size, and vein radius. In a recent study, Ng (2023) extended the Nye–Rempel–Wettlaufer framework to show that water flow in the veins amplifies excess diffusion, and that vein-water flow velocities of $\sim 10^1$–$10^2$ m yr$^{-1}$ yield a ten to hundred fold enhancement, able to explain the GRIP findings and sections of ice with anomalously high diffusion lengths found in the EPICA (European Project for Ice Coring in Antarctica) Dome C ice core by Pol et al. (2010) and found in the WAIS (West Antarctic Ice Sheet) Divide ice core by Jones et al. (2017) – which these authors interpreted as

signs of excess diffusion potentially caused by the short-circuiting mechanism. As pointed out by Ng (2023), the modulation of isotopic diffusion by vein-water flow means that the decay of climate signals at each ice-core site depends on the hydrology and connectivity of veins down the ice column, as well as the ice temperature, grain and vein sizes, and recrystallisation processes affecting these geometries.

   Here, we take the modelling of excess diffusion in a new direction to enable a critical research gap to be addressed.

Besides those records displaying signs of excess diffusion (accelerated signal decay or anomalous diffusion lengths) and motivating the theories in the first place, no direct observations have been made to show that isotopic short-circuiting actually

operates. Independent evidence is needed to verify the mechanism at the grain scale, for ice-core samples deemed affected by excess diffusion, and for polycrystalline ice generally. One way of testing the theories is to compare their predicted signal smoothing rate against the rate measured in ice doped with isotopic signals, but the slowness of diffusion makes such experiments prohibitively long at low temperature. A different laboratory-based approach, proposed herein, is to analyse ice affected by excess diffusion to look for the distinct grain-scale isotopic variations which the theories predict to result from short-circuiting. For instance, the theories of Nye (1998), Rempel and Wettlaufer (2003) and Ng (2023) – capturing the isotopic exchange between veins and ice in the absence of grain boundaries – imply axisymmetric patterns of $\delta$ around veins, which may be used for this purpose. Knowledge of these patterns is prerequisite to testing for short-circuiting this way. It also helps researchers who are developing techniques of making high-resolution isotopic measurements on ice, who currently lack information on how strong or weak the grain-scale variations in $\delta$ might be. Predicting the variety of isotopic patterns thus forms the main goal of this paper, although we leave the laboratory testing to future studies.

To simulate realistic patterns, we go beyond Nye (1998), Johnsen et al. (2000), Rempel and Wettlaufer (2003) and Ng (2023) by formulating a continuum model that includes grain boundaries, coupling diffusion across all three components: ice, veins, and grain boundaries. This integrated model is necessary, as we wish to test the four theories collectively, and each of them is missing some elements (e.g., Johnsen et al. (2000) did not couple together veins and grain boundaries, whereas the other theories neglected grain boundaries). However, we mean to examine the short-circuiting conceived in these theories, so we do not build more sophistication into the model to account for every conceivable process in polycrystalline ice. We are not trying to advance a new theory to describe isotopic diffusion in the most complete manner possible.

The model geometry, which remains simplified, allows us to explore the combined effect of veins and grain boundaries on the bulk-ice isotopic diffusivity. An outstanding question in this regard is whether diffusion along grain boundaries matters in ice with glaciological grain sizes (~ mm). Their effect on the bulk diffusivity is assumed to be significant in ultra fine-grained ice with ≈ 10–30 nm sized crystals (Lu et al., 2009), where grain-boundary surfaces have a high volumetric density (Jones et al. 2017). In contrast, for glacier ice, a much weaker effect may be suspected based on the calculations of Johnsen et al. (2000), who estimated that the grain boundaries in the GRIP Holocene ice (mean grain diameter ≈ 3 mm) need to be unrealistically thick (50 nm) to explain the observed excess diffusion, even if they are liquid films with the high isotopic diffusivity of water. Studying this question with a fully-coupled model has not been done before and forms our second goal. We compute the enhancement factor $f$ for ice of millimetre grain size at –32 °C and –52 °C (which approximate the upper column temperatures at the GRIP and EPICA core sites, respectively) for different grain-boundary properties and vein-water flow velocities.

Including grain boundaries in the modelling brings challenges. Most obviously, the grain-boundary thickness $c$ and grain-boundary diffusivity $D_b$ need to be specified; but as we will elaborate in Sect. 2, these parameters are not well constrained. In our calculations, we cover potential scenarios by experimenting with different assumptions for $c$ and $D_b$ in a sensitivity analysis. Another issue is that the model geometry does not permit analytical solution, unlike in the theories of Nye (1998), Rempel and Wettlaufer (2003), and Ng (2023). We tackle this by developing a bespoke numerical solution method.

The paper is organised as follows. Section 2 details our model formulation and solution method. Section 3 presents the computed isotopic patterns and enhancement factors for a range of parameters, including the end-member cases of thick, diffusive and thin, non-diffusive grain boundaries and intermediate scenarios. In Sect. 4, we discuss the prospects of detecting the isotopic signatures of excess diffusion in laboratory measurements on ice, focussing on techniques based on laser-ablation sampling (e.g. Malegiannaki et al., 2023). Readers keen to see the predicted patterns are advised to turn to Figs. 4–11. Those seeking to compute the diffusivity enhancement factor for conditions not covered by us can find our numerical code in the repository linked to the paper.

The importance of testing the theories cannot be understated, and several points are worth emphasising in this connection before we start. Given the idea of querying the short-circuiting mechanism, we do not claim that the modelled patterns will necessarily be found – or found at the predicted amplitudes – during grain-scale testing of ice. And while the mechanism has not been experimentally confirmed, ice-core studies seeking to understand excess diffusion on specific isotopic records should not automatically invoke it as if it is firmly established. With those core sections showing excess diffusion at GRIP, EPICA Dome C and WAIS Divide, their explanation by means of vein or grain-boundary short-circuiting remains plausible, but tentative, and the causal factors (e.g. why excess diffusion apparently occurs in those sections and not others) are unclear. In terms of probing the origin of excess diffusion at those sites, the most advanced analyses to date are probably the ones by Jones et al. (2017) and Ng (2023), who used models of the enhancement factor to calculate diffusion lengths to inform hypotheses about the cause. We refer the reader to these studies for more details on this subject.

## 2 Mathematical model

### 2.1 Model geometry

We use the set-up in Fig. 1a – adapted from Nye (1998), Rempel and Wettlaufer (2003) and Ng (2023), which represents ice crystal grains surrounding a vein by a vertical annular cylinder, in $a \leq r \leq b$, where $r$ is the radial coordinate, $a$ is the vein radius (~ $\mu$m), and $b$ approximates the mean grain radius (~ mm). The water vein is kept liquid by dissolved ionic impurities, which lower the melting point (Mulvaney et al., 1988; Nye, 1991; Mader, 1992b). We consider depth-varying isotopic signals in the bulk ice, with $z$ denoting depth. For a list of mathematical symbols used in this paper, see Table A1 in the Appendix.

Grain boundaries leading from the vein are modelled as planes of thickness $c$ ($\ll a$) at $\theta = 0$, $L$, and $2L$, where $\theta$ is the azimuth and $L = 2\pi/3$. Introducing them makes the problem non-axisymmetric, but their periodicity means that it suffices to solve the model in $0 \leq \theta \leq L$.

In plan view, the cylinder approximates a unit cell centred upon triple junctions in ice whose structure is idealised as honeycomb-like (Fig. 1b). In this picture, the radius $b$ reaches out roughly half-way along each grain boundary or to the middle

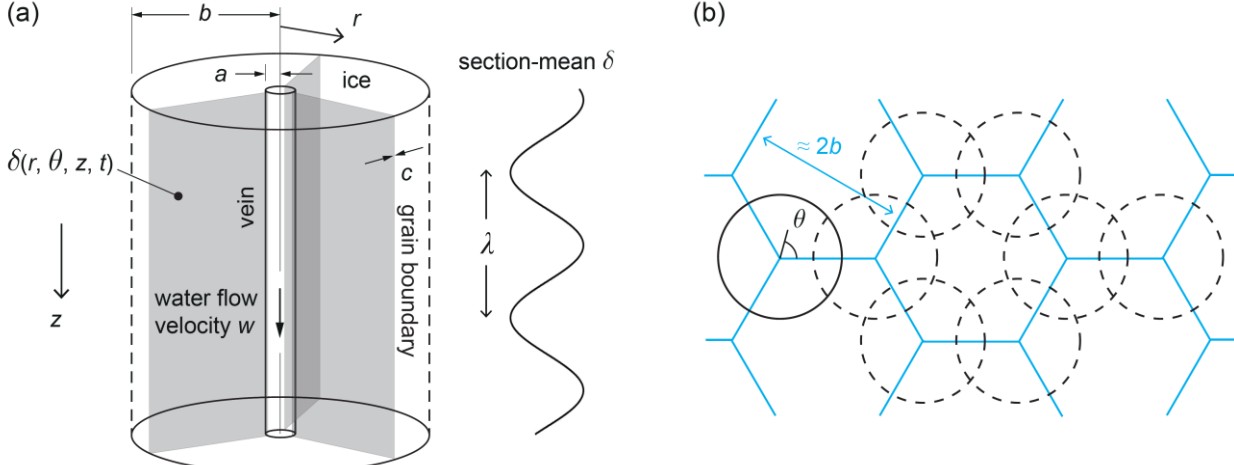

Figure 1. (a) Model geometry for calculating coupled isotopic diffusion in ice, vein, and grain boundaries near a triple junction. (b) Approximate view of what the cell in panel (a) represents in polycrystalline ice with hexagonal grains.

of grains; for convenience, we refer to $r \approx b$ at either location as the "interior". As in the theories of Nye (1998), Rempel and Wettlaufer (2003) and Ng (2023), our extended model geometry still idealises many aspects of the real system: (1) It ignores the detailed vein cross-section, which consists of three convex walls (Nye, 1989; Mader, 1992a; Ng, 2021), although their small length-scale implies perturbations only to the local isotopic concentrations near $r = a$. (2) Veins and grain boundaries are assumed stationary rather than migrating under recrystallisation processes. (3) Horizontal or near-horizontal veins and

grain boundaries are disregarded, so the model does not account for additional short-circuiting arising from these boundaries, which may distort the isotopic patterns near them and affect the enhancement factor. We discuss the last two limitations in Sect. 4.

### 2.2 Material properties

Prior to modelling signal evolution, we consider the isotopic diffusivities in the three components (ice, vein water, grain boundaries) and the grain-boundary thickness and, where relevant, explain values chosen for simulations. All diffusivities discussed here – referring to molecular diffusion – are applicable to the transport of oxygen and deuterium.

For the isotopic diffusivity in ice or "solid diffusivity" $D_s$, we use Ramseier's (1967) formula for self-diffusion in single ice crystals:

$$D_s = 9.1 \times 10^{-4} \exp\left(-\frac{7.2 \times 10^3}{T}\right) \ \text{m}^2\,\text{s}^{-1}, \tag{1}$$

in which $T$ denotes temperature in Kelvin. For the isotopic diffusivity in vein water or "liquid diffusivity" $D_v$, we use the composite exponential formula:

$$D_{\mathrm{v}} = \cfrac{1}{\cfrac{1}{1.085 \times 10^{-6} \exp\left(\cfrac{-1870}{T}\right)} + \cfrac{1}{2.942 \times 10^{7} \exp\left(\cfrac{-9474}{T}\right)}} \quad \mathrm{m^2\ s^{-1}} . \tag{2}$$

This formula was derived by Ng (2023) by fitting self-diffusivity data between –12.8 °C and –60.8 °C, which Xu et al. (2016) obtained by modelling crystal-growth rates measured in laboratory experiments. Equation (2) is consistent with the established formula of Gillen et al. (1972) for $T$ down to –31 °C but covers a greater temperature range. Figure 2 plots Eqs. (1) and (2). These formulas do not account for pressure dependence, which should cause only a minor correction under the glaciostatic overburden in ice sheets (a few % on $D_{\mathrm{v}}$; Prielmeier et al., 1988), nor the influence of dissolved impurities, whose characterisation is presently very limited. Thus, $D_{\mathrm{s}}$ and $D_{\mathrm{v}}$ might vary from the formulas. However, the temperature dependences shown in Fig. 2 should be robust, and departures from the formulas by a few times (e.g. see uncertainty for $D_{\mathrm{s}}$ indicated by Lu et al. (2009) in their Fig. 8) or even an order of magnitude are much smaller than the diffusivity contrast $D_{\mathrm{v}}/D_{\mathrm{s}}$ ~ $10^6$, which governs the qualitative interactions during vein short-circuiting.

What of the grain-boundary diffusivity $D_{\mathrm{b}}$ and thickness $c$? The physico-chemical influences on these parameters are poorly understood across the range of ice-core temperatures (≈ 0 to –55 °C); their values are uncertain and lack reliable formulas. Several empirical and theoretical constraints come to our rescue, as detailed below. But first we sketch more background on the grain-boundary properties of ice, as a step towards explaining our choices for these parameters.

Grain boundaries are disordered interfaces between crystals. Determining their properties experimentally is difficult because the microscopic scale concerned often means that a property can only be inferred from bulk measurements that mix crystal and grain-boundary effects (e.g. Lu et al., 2007). In ice, the grain-boundary thickness must be at least several times crystal lattice spacing (O–O distance: 0.276 nm; Hobbs, 1974). It may be higher in the presence of impurities (Thomson et al., 2013) but is generally expected to depend in complex ways on impurity type and concentration (Benatov and Wettlaufer, 2004). *Premelting* occurs at high temperature (Dash et al., 2006): that is, grain boundaries thicken and start to exhibit quasi-liquid behaviour near the melting point $T_{\mathrm{m}}$ as this is approached from below, at $T_{\mathrm{m}} - T = 0$ to ~ 10 K. Grain-boundary premelting in ice has been studied by (i) theoretical modelling of the forces and thermodynamics controlling the premelted film thickness (Wettlaufer, 1999; Benatov and Wettlaufer, 2004), (ii) laboratory measurements of the film thickness (Thomson et al., 2013), and (iii) classical molecular dynamical simulations (e.g. Moreira et al., 2018). Premelting in ice diminishes beyond a few °C below $T_{\mathrm{m}}$ and is expected to be negligible below ~ –10 °C. For instance, Lu et al. (2007, 2009) argued from experimental results for $D_{\mathrm{b}}$ (reported below) that premelting does not occur below –2 °C in pure ice, although it starts to occur at ≈ –8 °C in ice doped with HCl at 0.04% by mass (≈ 0.01 M bulk concentration). On the other hand, the notion of premelted grain boundaries features in Johnsen et al.'s (2000) and Rempel and Wettlaufer's (2003) theories of excess diffusion, even though their analyses considered much colder ice. Johnsen et al. (2000), particularly, referred to the grain boundaries in ice at $T = -32$ °C as "supercooled water films" and took the liquid diffusivity $D_{\mathrm{v}}$ at that temperature ($1.87 \times 10^{-10}$ m$^2$ s$^{-1}$, blue cross in

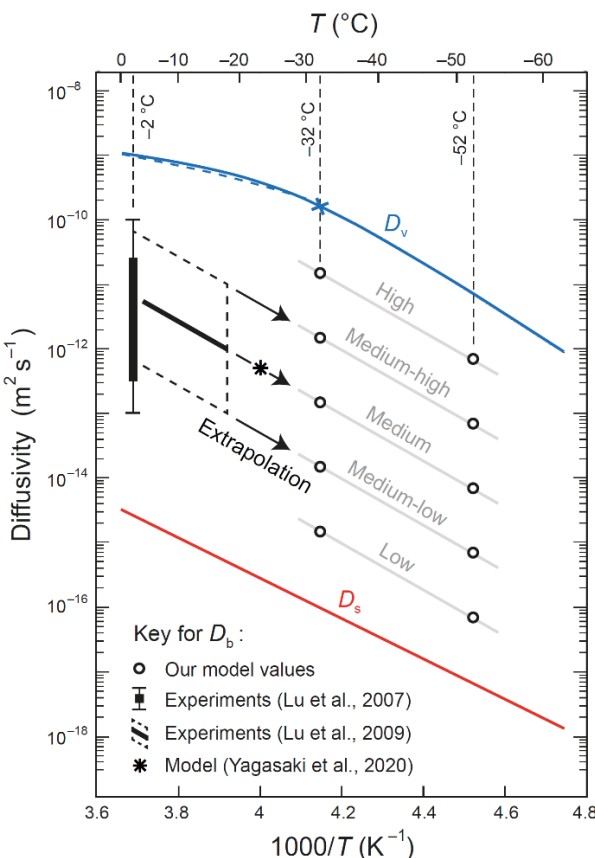

Figure 2. Arrhenius plot of the isotopic diffusivities $D_s$ (ice), $D_v$ (vein water) and $D_b$ (grain boundaries). The curves for $D_s$ and $D_v$ come from
Eqs. (1) and (2). Blue dashed curve plots Gillen et al.'s (1972) relation for $D_v$. Blue cross locates the diffusivity used by Johnsen et al. (2000)
(Sect. 2.2). Box and whiskers at –2 °C (bar: likely range; whiskers: maximal range) and the inclined black line (dashed box: uncertainty
range) plot the laboratory-based estimates for $D_b$ of Lu et al. (2007) and Lu et al. (2009), respectively. The star plots $D_b$ at 250 K from
molecular dynamical simulation (Yagasaki et al., 2020). To pose values of $D_b$ for modelling at –32 and –52 °C, we extrapolate the trend of
Lu et al.'s (2009) results to those temperatures and expand the uncertainty to form two sets of grain-boundary diffusivities (circles; values
in Table 2), which we address by using descriptive labels (grey wording). Grey lines indicate the same system of referring to the size of $D_b$
at other temperatures.

Fig. 2) to be the grain-boundary diffusivity $D_b$ in calculations. As we will see, this estimate for $D_b$ is probably too high.

The question whether grain boundaries in ice are watery or more like solid lattice has bearing on where isotope
fractionation (during phase change) is envisaged to occur in the system – whether at (1) the transition between them and crystal
lattice within the grain interior, or (2) where they meet veins. If one assumes grain boundaries to be liquid, then fractionation
occurs at the liquid-to-solid phase change at location 1, not at location 2, where there is no phase change. If one envisages
them to be solid, closely resembling crystal lattice, then fractionation occurs at location 2 and not (or negligibly) at location 1.

In our model, we assume the latter scenario because our simulations explore temperatures far below the premelting regime,
making the former scenario less plausible (the fractionation coefficients will be described in Sect. 2.3). Note, however, that the question is unsettled given the lack of experimental determination, and fractionation may occur at both places in reality (e.g. in a hybrid scenario where grain boundaries have microstructural properties intermediate between solid and liquid).

We turn to the parameter choices, treating grain-boundary thickness $c$ first. Information comes from two sources. Thomson et al. (2013) used optical scattering to measure $c$ in ice at $T \approx -1.5$ °C with different dissolved impurity concentrations
(NaCl) and different grain-boundary orientations. The impurity concentration at grain boundaries was estimated from the bulk concentration as it cannot be measured directly. They found $c$ from 1 to 8 nm, generally increasing with impurity level (this factor promotes interfacial molecular disorder) for different crystal misorientation angles. For $T < -1.5$ °C, no experimental measurements of $c$ have been made so far, but molecular-scale dynamical simulations offer a handle. Yagasaki et al. (2020) used the TIP4P/Ice model to study molecular transport at grain boundaries in impurity-free ice at 250 K ($\approx -23$ °C) and found
$c \sim 1$ nm under a variety of conditions. Given these studies, we choose three values of $c$ for our modelling: 1 nm, 5 nm and 10 nm (Table 1). The highest value accounts for the possibility of thick grain boundaries resulting from high impurity levels.

For the grain-boundary diffusivity $D_b$, we rely on guidance from the experimental results of Lu et al. (2007, 2009), which are the only results available to date on ice. For $T$ from $-18$ °C to $-1$ °C, these authors determined that $D_b$ lies intermediate between $D_s$ and $D_v$, several orders of magnitude from each of them (Fig. 2). Their experiments measured the inter-diffusivity
$D_{eff}$ of H and D in nanocrystalline sandwiches of $H_2O/D_2O/H_2O$ ice by monitoring the reaction zones at the interfaces with thermal desorption spectroscopy, a technique that ablates the ice with laser and analyses the vapour composition. They estimated $D_b$ from $D_{eff}$ by a model inversion based on the Hart–Mortlock equation ($D_{eff}$ as a linear combination of the compon-

Table 1: Grain-boundary thicknesses
investigated in our modelling.

| Description | $c$ (nm) |
|---|---|
| Thin | 1 |
| Intermediate | 5 |
| Thick | 10 |

Table 2: Isotopic diffusivities (in $m^2 s^{-1}$) used in our modelling. At each temperature, we study five values of the grain-boundary diffusivity, $D_b$.

| | Description | $T = -32$ °C | $T = -52$ °C |
|---|---|---|---|
| $D_v$ | | $1.65 \times 10^{-10}$ | $7.08 \times 10^{-12}$ |
| | High | $1.5 \times 10^{-11}$ | $7 \times 10^{-13}$ |
| | Medium-high | $1.5 \times 10^{-12}$ | $7 \times 10^{-14}$ |
| $D_b$ | Medium | $1.5 \times 10^{-13}$ | $7 \times 10^{-15}$ |
| | Medium-low | $1.5 \times 10^{-14}$ | $7 \times 10^{-16}$ |
| | Low | $1.5 \times 10^{-15}$ | $7 \times 10^{-17}$ |
| $D_s$ | | $9.83 \times 10^{-17}$ | $6.60 \times 10^{-18}$ |

ent diffusivities, weighted by the component volume fractions). In their 2007 study, conducted at –2 °C, their $D_b$ estimate spans 3 orders of magnitude, although they suggested a likely range of 1–2 orders (bar and whiskers, Fig. 2). Their 2009 study extended the measurements of $D_b$ down to –18 °C (sloping black line, Fig. 2), with an order of magnitude uncertainty on either side, finding for $D_b$ an Arrhenius-type temperature dependence with an activation energy of $\approx 69$ kJ mol$^{-1}$.

Below –18 °C, $D_b$ has not been experimentally measured. To pose $D_b$ at –32 and –52 °C for modelling, we extrapolate the estimates of Lu et al. (2009) down the Arrhenius trend (Fig. 2), assuming the same activation energy and $D_b$ to lie between $D_s$ and $D_v$ at lower temperatures. This approach finds support in the modelled value of $D_b$ at 250 K from Yagasaki et al. (2020) (star, Fig. 2). However, we widen the uncertainty range of the Lu et al. (2009) estimates by an order of magnitude, because (i) the Hart–Mortlock equation crudely approximates the bulk diffusivity[1], and the version of the equation used in their inversion ignores the presence of veins, and (ii) they showed that doping the ice with HCl increased $D_{eff}$ by $\approx 20$ times above the pure-ice value, indicating that dissolved impurities can raise $D_b$ substantially. Uncertainties in their inversion from assumptions about the grain-boundary width are discussed by Lu et al. (2007) also. Based on the extrapolation, we choose two sets of five values for $D_b$ (circles, Fig. 2): one set for –32 °C and the other set for –52 °C, as listed in Table 2. In each set, which spans a generous range for sensitivity analysis, the middle three values of $D_b$ represent direct extrapolations of the laboratory measurements and the uncertainty range of Lu et al. (2009). The lowest and highest values, respectively, mimic the more extreme scenarios of coupled diffusion near the no-grain-boundary limit and of high impurity concentration at grain boundaries. For convenience, we refer to the values in each set as *low*, *medium-low*, *medium*, *medium-high*, and *high* (Fig. 2, Table 2). This descriptive scale for $D_b$ is applicable to other temperatures (grey lines in Fig. 2) on the basis of the assumed trend. We explore select values of $D_b$ and $c$ in this paper, given the impracticality of covering a large number of parameter combinations when computing isotopic patterns and enhancement factors.

That $D_b$ is bracketed by $D_s$ and $D_v$ corroborates insights from classical molecular dynamical simulations. Moreira et al. (2018) found that a few degrees below $T_m$, the simulated molecular transport along premelted grain boundaries resembles diffusion in glassy systems and is sub-diffusive in character (with mean-square displacement of molecules $\sim t^\gamma$, where $t$ denotes time and $\gamma < 1$), reflecting lateral confinement of the grain boundaries by adjacent crystal lattice. The grain boundaries at 250 K simulated by Yagasaki et al. (2020) structurally resemble low-density liquid water. Besides estimating a corresponding value for $D_b$, Yagasaki et al. (2020) studied diffusion along triple junctions, finding a diffusivity of $3.4D_b$ for them. We cannot adopt this as the vein diffusivity $D_v$, because their model does not recognise water-filled veins at triple junctions, whose presence in ice has been confirmed by optical (Mader, 1992a) and nuclear magnetic resonance (Brox et al., 2015) methods.

---

[1] That the Hart–Mortlock equation may only roughly approximate the bulk/effective diffusivity in some applications has been recognised (e.g. Lundy, 1978). Moreover, for the coupled diffusion studied here, our results (Sect. 3.2) imply a bulk diffusivity varying with signal wavelength, not what the Hart–Mortlock equation would predict.

## 2.3 *Continuum formulation*

For the system in Fig. 1, let us denote the concentrations of a trace isotope ($^{18}$O or D) in the ice, vein, and grain boundaries by $N_s(r, \theta, z, t)$, $N_v(z, t)$, and $N_b(r, z, t)$, respectively, where $t$ is time. We assume $N_v$ to be independent of $r$ and $\theta$, and $N_b$ to be uniform across the grain-boundary thickness. The concentrations satisfy the conservation equations

$$\frac{\partial N_s}{\partial t} = D_s \left( \frac{1}{r} \frac{\partial}{\partial r} \left( r \frac{\partial N_s}{\partial r} \right) + \frac{1}{r^2} \frac{\partial^2 N_s}{\partial \theta^2} + \frac{\partial^2 N_s}{\partial z^2} \right) , \tag{3}$$

$$\frac{\partial N_v}{\partial t} = D_v \frac{\partial^2 N_v}{\partial z^2} - w \frac{\partial N_v}{\partial z} + \frac{3D_s}{\pi a} \int_0^L \frac{\partial N_s}{\partial r} \bigg|_{r=a} \mathrm{d}\theta + \frac{3c D_b}{\pi a^2} \frac{\partial N_b}{\partial r} \bigg|_{r=a} , \tag{4}$$

$$\frac{\partial N_b}{\partial t} = D_b \left( \frac{\partial^2 N_b}{\partial r^2} + \frac{\partial^2 N_b}{\partial z^2} \right) + \frac{2D_s}{rc} \frac{\partial N_s}{\partial \theta} \bigg|_{\theta=0} , \tag{5}$$

where $w$ is the vein-flow velocity in the downward ($z$-) direction, and other symbols have been introduced. These equations account for isotopic exchange across the vein wall, between grain boundaries and vein, and between grain boundaries and ice. Respectively, the second-last term in Eq. (4), the last term in Eq. (5), and the last term in Eq. (4) – which are source terms in those equations – describe the isotope fluxes leaving the ice radially and azimuthally, and grain boundaries radially. The factor 3 sums flux contributions to the vein from all directions. Taylor dispersion along the vein is ignored as the corresponding Péclet number ($\lesssim 10^{-1}$) would only raise the vein liquid diffusivity by $< 0.1\%$. We specify the boundary conditions $\partial N_s/\partial r = \partial N_b/\partial r = 0$ at $r = b$ (zero gradient in the interior) and anticipate $\partial N_s/\partial \theta = 0$ at $r = a$, because the vein wall at different azimuths contacts the same vein isotopic concentration. Rotational periodicity implies solution symmetry in $0 \leq \theta \leq L$ about $L/2$.

Deriving a model for the isotopic deviation $\delta$ follows Rempel and Wettlaufer's (2003) method. If $N_{s0}$, $N_{v0}$ and $N_{b0}$ are the number densities of the major isotope ($^{16}$O or H) in the three components, then fractionation at the vein wall and at the vein-end of grain boundaries yields $\alpha N_v/N_{v0} = N_s|_{r=a}/N_{s0} = N_b|_{r=a}/N_{b0}$, where $\alpha$ is the fractionation coefficient. Following Rempel and Wettlaufer (2003), we assume equilibrium fractionation. Following them also, we will set $\alpha = 1$, which seems to be a plausible approximation because $\alpha(^{18}\mathrm{O}/^{16}\mathrm{O}) \approx 1.0029$ and $\alpha(\mathrm{D/H}) \approx 1.021$ at 0 °C (O'Neil, 1968; Árnason, 1969; Lehmann and Siegenthaler, 1991), but note that the temperature dependence of $\alpha$ in $T < 0$ °C for either element is unknown[2]. We assume no fractionation on the side walls of grain boundaries (Sect. 2.2), so $N_b = N_s|_{\theta=0}$ in $r \geq a$. By rewriting Eqs. (4) and (5) in terms of $N_s$ (with $N_{v0} \approx N_{b0} \approx N_{s0}$ taken as constant), eliminating their time derivatives with Eq. (3), and using the definition

$$\delta = \delta(r, \theta, z, t) = \frac{N_s}{N_{s0}} - 1 , \tag{6}$$

---

[2] We have not found published values of $\alpha$ (for the liquid–solid phase change) in $T < 0$, certainly not at –32 and –52 °C. It is unsurprising that laboratory measurements of $\alpha$ have not been made at the strongly-depressed melting temperatures specific to the vein system.

we obtain the diffusion equation

$$\frac{\partial \delta}{\partial t} = D_s \left( \frac{1}{r}\frac{\partial}{\partial r}\left(r\frac{\partial \delta}{\partial r}\right) + \frac{1}{r^2}\frac{\partial^2 \delta}{\partial \theta^2} + \frac{\partial^2 \delta}{\partial z^2} \right) , \tag{7}$$

with the boundary conditions

$$\left.\frac{\partial \delta}{\partial r}\right|_{r=b} = 0 \;, \qquad \left.\frac{\partial \delta}{\partial \theta}\right|_{r=a} = 0 \;, \tag{8}$$

$$\frac{\partial^2 \delta}{\partial r^2} - \beta_v \frac{\partial^2 \delta}{\partial z^2} + \frac{w}{D_s}\frac{\partial \delta}{\partial z} + \frac{1}{a}\left(\frac{\partial \delta}{\partial r} - \frac{3\alpha}{\pi}\int_0^L \left.\frac{\partial \delta}{\partial r}\right|_{r=a} \mathrm{d}\theta - \frac{3\alpha\varepsilon}{\pi}(\beta_b + 1)\left.\frac{\partial \delta}{\partial r}\right|_{r=a,\,\theta=0}\right) = 0 \quad \text{at } r=a, \tag{9}$$

$$\frac{1}{r}\frac{\partial \delta}{\partial r} - \beta_b\left(\frac{\partial^2 \delta}{\partial r^2} + \frac{\partial^2 \delta}{\partial z^2}\right) + \frac{1}{r^2}\frac{\partial^2 \delta}{\partial \theta^2} - \frac{2}{rc}\frac{\partial \delta}{\partial \theta} = 0 \quad \text{on } \theta = 0, \; a \le r \le b. \tag{10}$$

The boundary conditions in Eqs. (9) and (10), derived from Eqs. (4) and (5), encapsulate advection and diffusion along the vein and diffusion within the grain-boundary planes. The boundary condition at $\theta = L$ is met automatically, given the solution symmetry. We have introduced the thinness parameter

$$\varepsilon = \frac{c}{a} \quad (\ll 1) , \tag{11}$$

which measures the grain-boundary thickness scaled to the vein radius. The parameters

$$\beta_v = \frac{D_v}{D_s} - 1 \quad \text{and} \quad \beta_b = \frac{D_b}{D_s} - 1 \tag{12}$$

quantify the diffusivity contrasts of water to ice and grain boundary to ice, respectively. As noted in Sect. 2.2, typically $\beta_v \sim 10^6$ (Fig. 2); $\beta_b$ ($< \beta_v$) is also large, but depends on the chosen grain-boundary diffusivity. Notice one cannot lump all grain-boundary properties into a single parameter (e.g. the diffusivity–thickness product $cD_b$ or $\varepsilon(\beta_b + 1)$) in this model.

The partial differential equation problem for $\delta$ in Eqs. (7) to (10) is linear. To quantify signal decay, we study how sinusoidal signals of different wavelength $\lambda$ – or wavenumber $k_z = 2\pi/\lambda$ – smooth out in time (Nye, 1998; Rempel and Wettlaufer, 2003; Ng, 2023) by posing the trial solution

$$\delta \propto H(r,\theta)\exp(-D_s\zeta t + ik_z z) , \tag{13}$$

where $\zeta = \zeta_R + i\zeta_I$ is a complex decay-rate parameter. The enhancement factor measuring the level of excess diffusion is given by the ratio of the signal decay rate $D_s\zeta_R$ in Eq. (13) to the baseline decay rate $D_s k_z^2$ in monocrystalline ice (ice without grain boundaries and veins). On defining $\zeta_R = k_z^2 + k_r^2$, the enhancement factor is

$$f = 1 + \frac{k_r^2}{k_z^2}.$$ (14)

In Eq. (13), the function $H(r, \theta) = H_R + iH_I$ determines the spatial pattern of isotopic signals in three dimensions (3D). At depth $z$, $Re[H\exp(ik_z z)]$ gives their amplitude across the annular sector $0 \leq \theta \leq L$, $a \leq r \leq b$, and the section-mean isotopic signal (ignoring the exponential time decay factor) is

$$\frac{3}{\pi b^2} \int_0^L \int_a^b Re[rH(r,\theta)\exp(-ik_z z)] \, dr \, d\theta.$$ (15)

The sinusoidal signals at different radii and azimuths have the phase angle $\phi = \tan^{-1}(H_I/H_R) + k_z z - D_s \zeta_I t$. Therefore, when $\zeta_I$ is non-zero, the signals migrate at the velocity $\zeta_I D_s/k_z$ in the $z$-direction.

### 2.4 Scaled model

When addressing isotopic patterns later, it will be useful to reference the features on them (e.g. size or radial position) to the grain radius $b$. To facilitate this, we non-dimensionalise the model by letting

$$r* = \frac{r}{b},$$ (16)

at the same time scaling other variables as follows:

$$z* = \frac{z}{b}, \quad t* = \frac{t}{(b^2/D_s)}, \quad \lambda* = \frac{\lambda}{b}, \quad \zeta* = b^2\zeta, \quad [k_z*, k_r*] = b[k_z, k_r].$$ (17)

The scaled model equivalent to Eqs. (7) to (10) is then

$$\frac{\partial\delta}{\partial t} = \frac{1}{r}\frac{\partial}{\partial r}\left(r\frac{\partial\delta}{\partial r}\right) + \frac{1}{r^2}\frac{\partial^2\delta}{\partial\theta^2} + \frac{\partial^2\delta}{\partial z^2},$$ (18a)

$$\partial\delta/\partial r\big|_{r=1} = 0, \quad \partial\delta/\partial\theta\big|_{r=\xi} = 0,$$ (18b)

$$\frac{\partial^2\delta}{\partial r^2} - \beta_v\frac{\partial^2\delta}{\partial z^2} + \chi\frac{\partial\delta}{\partial z} + \frac{1}{\xi}\left(\frac{\partial\delta}{\partial r} - \frac{3\alpha}{\pi}\int_0^L \frac{\partial\delta}{\partial r}\bigg|_{r=\xi} d\theta - \frac{3\alpha\varepsilon}{\pi}(\beta_b+1)\frac{\partial\delta}{\partial r}\bigg|_{r=\xi,\,\theta=0}\right) = 0 \quad \text{at} \quad r = \xi,$$ (18c)

$$\frac{1}{r}\frac{\partial\delta}{\partial r} - \beta_b\left(\frac{\partial^2\delta}{\partial r^2} + \frac{\partial^2\delta}{\partial z^2}\right) + \frac{1}{r^2}\frac{\partial^2\delta}{\partial\theta^2} - \frac{2}{r(\varepsilon\xi)}\frac{\partial\delta}{\partial\theta} = 0 \quad \text{on} \quad \theta = 0,$$ (18d)

where we have dropped the stars for convenience (we work with dimensionless variables from now on). The parameter

$$\xi = \frac{a}{b} \quad (\ll 1)$$ (19)

is the dimensionless vein radius ($c \ll a \ll b$ in glacier ice translates to $\varepsilon\xi \ll \xi \ll 1$), and

$$\chi = \frac{wb}{D_s} \tag{20}$$

is a Péclet number measuring the importance of vein-flow driven advection relative to solid-state diffusion. The trial solution in Eq. (13) becomes

$$\delta \propto H(r,\theta)\exp(-\zeta t + ik_z z)\,, \tag{21}$$

while Eq. (14) for the enhancement factor $f$ is unchanged under the scaling.

2.5 *Eigenvalue problem*

It remains to solve for the pattern $H(r,\theta)$ for signals of any wavenumber $k_z$. Substituting $\delta$ from Eq. (21) into Eq. (18) leads to

$$\frac{\partial^2 H}{\partial r^2} + \frac{1}{r}\frac{\partial H}{\partial r} + \frac{1}{r^2}\frac{\partial^2 H}{\partial \theta^2} + s^2 H = 0\,, \tag{22}$$

with the boundary conditions

$$\partial H/\partial r\big|_{r=1} = 0\,, \quad \partial H/\partial\theta\big|_{r=\xi} = 0\,, \tag{23}$$

$$\frac{\partial^2 H}{\partial r^2}\bigg|_{r=\xi} + p_1 H\big|_{r=\xi} + \frac{1}{\xi}\left( \frac{\partial H}{\partial r}\bigg|_{r=\xi} - \frac{p_2}{L}\int_0^L \frac{\partial H}{\partial r}\bigg|_{r=\xi} \, d\theta - \frac{p_3}{L}\frac{\partial H}{\partial r}\bigg|_{r=\xi,\,\theta=0} \right) = 0\,, \tag{24}$$

$$\frac{1}{r}\frac{\partial H}{\partial r}\bigg|_{\theta=0} - \beta_b\left( \frac{\partial^2 H}{\partial r^2}\bigg|_{\theta=0} - k_z^{\,2}H\big|_{\theta=0} \right) + \frac{1}{r^2}\frac{\partial^2 H}{\partial\theta^2}\bigg|_{\theta=0} - \frac{2}{r(\varepsilon\xi)}\frac{\partial H}{\partial\theta}\bigg|_{\theta=0} = 0\,. \tag{25}$$

Here, we have defined

$$s^2 = k_r^{\,2} + i\zeta_I \tag{26}$$

and introduced the parameters

$$p_1 = \beta_v k_z^{\,2} + ik_z\chi, \quad p_2 = 2\alpha, \quad p_3 = 2\alpha\varepsilon(\beta_b + 1)\,. \tag{27}$$

Equations (24) and (25) may be further simplified by using Eq. (22) to reduce the number of high-order derivatives; thus, we find

$$(p_1 - s^2)H\big|_{r=\xi} = \frac{1}{\xi L}\left( p_2\int_0^L \frac{\partial H}{\partial r}\bigg|_{r=\xi} \, d\theta + p_3\frac{\partial H}{\partial r}\bigg|_{r=\xi,\,\theta=0} \right) \tag{28}$$

and

$$\frac{\partial^2 H}{\partial r^2}\bigg|_{\theta=0} - \frac{\beta_b k_z^{\,2} - s^2}{\beta_b + 1}H\big|_{\theta=0} = -\frac{2}{(\beta_b + 1)r\varepsilon\xi}\frac{\partial H}{\partial\theta}\bigg|_{\theta=0}\,. \tag{29}$$

Equations (22), (23), (28) and (29) need to be solved to determine the isotopic patterns. They constitute a homogeneous boundary value problem for $H$ with the eigenvalue $s^2$, whose real part $k_r^2$ leads to the enhancement factor (see Eqs. (26) and (14)) and whose imaginary part $\zeta_1$ is non-zero if the vein water flows ($w, \chi \neq 0$); thus, vein-water flow causes the signals to migrate in the same direction, as in Ng's (2023) model. The slowest-decaying eigenmode (with minimum $\mathrm{Re}(s^2) > 0$) yields the desired pattern, as the other eigenmodes decay faster, leaving this mode to be observed in long time. The problem is non-trivial because of mixed boundary conditions at the vein wall and grain boundaries. Solution by the separation of variables $H = H_1(r)H_2(\theta)$ could exploit the periodicity in $\theta$ for $H_2$; equivalently, one could take the cosine transform azimuthally (e.g. $\sqrt{2/L} \int_0^L H \cos(n\theta/L)\mathrm{d}\theta$) and the Hankel transform in the radial direction. However, we find that analytic solution does not seem feasible by these conventional approaches – a fundamental obstacle being mismatch between the Fourier kernel of the grain-boundary condition in Eq. (29) and the Hankel kernel of the differential operator in Eq. (22). We therefore solve the problem numerically. Readers not interested in the associated details might skip on to the last paragraph of Sect. 2.6.

### 2.6 Numerical method

We use the pseudo-spectral method, employing Chebyshev collocation in the $\theta$-direction to achieve "spectral accuracy" in approximating the solution (Boyd, 2000; Trethethen, 2000). Although the angular periodicity suggests using trigonometric basis functions instead (i.e. Fourier spectral method), the corresponding approximation lacks spectral accuracy and converges much more slowly than Chebyshev polynomials, as $H$ is nonsmooth (with discontinuous gradient) across the grain boundaries. We use the finite-difference approximation in the radial direction.

The solution on each grain boundary can be written as $G(r) \equiv H(r, 0)$. This enables us to work with alternative variables, by splitting $H$ into the sum

$$H(r,\theta) = F(r,\theta) + G(r), \tag{30}$$

where the field $F$ represents variations in the ice sector unaccounted for by $G$. Usefully, $F$ is zero along the grain boundaries and on the vein wall (as $\partial H/\partial\theta = 0$ there). The decomposition converts Eqs. (22) and (23) to the partial differential equation

$$F_{rr} + \frac{F_r}{r} + \frac{F_{\theta\theta}}{r^2} + s^2 F = -\left(G'' + \frac{G'}{r} + s^2 G\right) \tag{31}$$

with the homogeneous boundary conditions

$$F_r\big|_{r=1} = 0, \qquad F(r,\ 0) = F(r,\ L) = 0, \qquad F(\xi,\ \theta) = 0. \tag{32}$$

Meanwhile, Eqs. (29) and (28) become the ordinary differential equation

$$G'' - \frac{\beta_b k_z^2 - s^2}{\beta_b + 1} G = -\frac{2}{(\beta_b + 1)r\varepsilon\xi} F_\theta\big|_{\theta=0}, \tag{33}$$

with boundary conditions at the vein wall and in the grain interior given by

$$(p_1 - s^2)G(\xi) = \frac{1}{\xi}\left[\frac{p_2}{L}\int_0^L F_r\big|_{r=\xi}\,\mathrm{d}\theta + \left(p_2 + \frac{p_3}{L}\right)G'(\xi)\right], \tag{34}$$

$$G'(1) = 0. \tag{35}$$

We have used the prime (subscript) notation to denote ordinary (partial) derivatives above. The differential equations for $G$ and $F$ are coupled via their source terms.

Because the vein short-circuits diffusion in the ice, we expect the solution to vary rapidly just outside the vein wall and slowly in the grain interior, notably away from grain boundaries. To resolve the variations near $r = \xi$ (vein wall) with sufficient grid points, without over-introducing grid points in the interior (which slows numerical computation), we make a change of the radial variable

$$R = 1 - \ln r, \text{ i.e., } r = e^{-(R-1)}. \tag{36}$$

The interior and the vein wall are located at $R = 1$ and $R = R_{\max} = 1 - \ln\xi$, respectively (Fig. 3a).

        Next, we set up the Chebyshev collocation points

$$x = \cos\left(\frac{n\pi}{N}\right), \quad n = 0, 1, 2 \dots, N, \tag{37}$$

choosing

$$\theta = \frac{L}{2}(x+1) \tag{38}$$

such that the interval $x = [-1, 1]$ maps onto the angular range $\theta = [0, L]$ of the sector (Fig. 3b, c). With these transformations, the coupled problem for $F$ and $G$ becomes

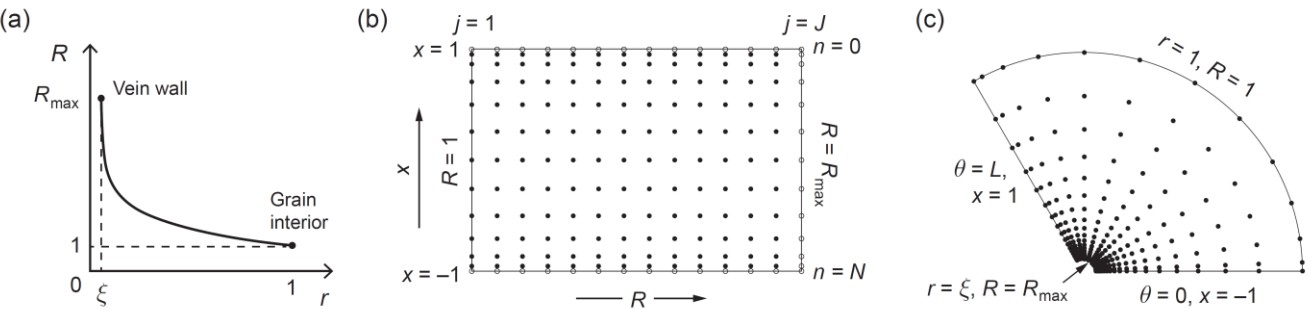

Figure 3. Elements of the mixed spectral–finite difference numerical method. (a) Radial coordinate transformation used to increase spatial
resolution near the vein. (b) Numerical grid for $F(R, x)$. Filled dots indicate solution points; open circles, zero boundary values. (c) The same grid points on the ice domain. Panels (b) and (c) are illustrative; we use many more grid points ($N = 100$, $J = 201$) than shown.

$$-e^{2(R-1)}\left[F_{RR}+\left(\frac{2}{L}\right)^2 F_{xx}\right]+e^{2(R-1)}\left[\beta_b G''+(\beta_b+1)G'\right]-\beta_b k_z^2 G+\frac{4e^{R-1}}{(\varepsilon\xi)L}F_x\Big|_{x=-1}=s^2 F\ ,\tag{39a}$$

$$\text{Boundary conditions:}\quad F_R\big|_{R=1}=0\ ,\quad F(R,\ x=\pm1)=0\ ,\quad F(R_{max},\ x)=0\ ,\tag{39b}$$

and

$$-(\beta_b+1)e^{2(R-1)}(G''+G')+\beta_b k_z^2 G-\frac{4e^{R-1}}{(\varepsilon\xi)L}F_x\big|_{x=-1}=s^2 G\ ,\tag{40a}$$

$$\text{Boundary conditions:}\quad \frac{1}{\xi^2}\left[\frac{p_2}{2}\int_{-1}^{1}F_R\big|_{R=R_{max}}\mathrm{d}x+\left(p_2+\frac{p_3}{L}\right)G'(R_{max})\right]+p_1 G(R_{max})=s^2 G(R_{max})\ ,\quad G'(1)=0\ .\tag{40b}$$

We have written these results with $s^2$ on the right-hand side to facilitate the eigenvalue calculation.

The method proceeds by discretising the $x$-axis with the Chebyshev points and the $R$-axis as $J$ equidistant points (Fig. 3b)
and using the spectral differentiation matrix of Trefethen (2000; p.53) and finite differencing to compute derivatives in these
respective directions. With $F$ zero on three edges of the solution domain, there are $(N-1)(J-1)$ unknowns in $F_{n,j}$ and $J$ unknowns
in $G_j$, for $n=0, 1, 2 \ldots, N$ and $j=1, 2 \ldots, J$. The scheme converts Eqs. (39) and (40) into a system of linear equations $\mathbf{M}\mathbf{v}=s^2\mathbf{v}$, where the solution eigenvector (a column vector)

$$\mathbf{v}=[\ F_{1,1}\ F_{1,2}\ldots F_{1,J-1}\quad F_{2,1}\ F_{2,2}\ldots F_{2,J-1}\quad F_{3,1}\ F_{3,2}\ldots F_{3,J-1}\quad\ldots\quad F_{N-1,1}\ F_{N-1,2}\ldots F_{N-1,J-1}\quad G_1\ G_2\ldots G_J\ ]^{\mathrm{T}}\tag{41}$$

has $(N-1)(J-1)+J$ elements, and $\mathbf{M}$ is a sparse-banded matrix (detailed in Section S1 and Fig. S1 in the Supplement). After
using the MATLAB function eig to compute $s^2$ from $\mathbf{M}$, we find $\mathbf{v}$ corresponding to the slowest-decaying eigenmode and put
$F_{n,j}$ and $G_j$ back in cylindrical polar coordinates to build the solution $H$. Our computation used $N=100$ and $J=201$ points,
and we checked for numerical convergence and convergence at diminishing grain-boundary thickness $c$ towards Ng's (2023)
analytic solution, which describes the vein-only system without grain boundaries.

All isotopic patterns reported below display $H$ after it has been regridded at a constant $\theta$-spacing by Lagrange
interpolation from the Chebyshev grid values, normalised by the value of $H$ at $r=1$, $\theta=L/2$, and copied from $0\le\theta\le L$ into
the other sectors to fill the ice annulus. At $r=1$, $\theta=L/2$, a position which we call the "mid-grain interior", the vertical sinusoidal
signal in $\delta$ has maximum amplitude because diffusion short-circuiting subdues the signal amplitude more strongly elsewhere,
especially near the vein and grain boundaries. Consequently, the mid-grain interior signal closely approximates and has a
slightly higher amplitude than the bulk vertical isotopic signal in Eq. (15) derived through horizontal averaging or,
equivalently, the signal measured by ice-core continuous flow analysis (CFA) (Kaufmann et al., 2008; Bigler et al., 2011). The
normalisation thus puts our pattern amplitudes in Sect. 3.1 in a dimensionless unit, scaled (approximately) to the bulk vertical
signal. It allows the absolute amplitude of the $\delta$-variations of the predicted patterns to be inferred for any bulk-signal amplitude.

## 3   Results and analysis

We proceed to examine computed isotopic patterns (Sect. 3.1) and bulk-diffusivity enhancement factors (Sect. 3.2) for different model parameters. In our model runs, we set the vein and grain sizes at $a = 1$ $\mu$m and $b = 1$ mm and assume the fractionation coefficient $\alpha = 1$, so the results can be compared with those of Rempel and Wettlaufer (2003) and Ng (2023) and applied to either $\delta^{18}$O or $\delta$D. Using precise fractionation coefficients at 0 °C ($\approx 1.0029$ for oxygen, $\approx 1.021$ for hydrogen; Sect. 2.3) changes the results numerically in a minor way that does not alter our qualitative findings. We will report only briefly on the qualitative effects of changing $a$ and $b$, which were examined in more detail by Rempel and Wettlaufer (2003).

There are 30 parameter combinations from the choices of temperatures $T$ (–32 °C, –52 °C), grain-boundary thicknesses $c$ (1, 5, 10 nm; Table 1), and grain-boundary diffusivities $D_b$ (Table 2). For each combination, we compute results for signal wavelengths $\lambda$ across the range 0.005–0.15 m and different vein-water flow velocities $w$ in 0–50 m yr$^{-1}$ when $T = $ –32 °C and 0–5 m yr$^{-1}$ when –52 °C. These ranges enable study of the enhancement factor $f$ as a function of $\lambda$ and $w$ in Sect. 3.2. Note that $w$ at ice-core sites is unknown and has not been measured (Ng, 2023). We chose the $w$-ranges here based on flow velocities of $\sim 10^1$ m yr$^{-1}$ in microns-thick veins that Nye and Frank (1973) estimated in their theory of water percolation in ice sheets, and the expectation that blockage or disconnection of the vein network (e.g. by dust particles) can drastically reduce $w$.

In Sect. 3.1, we analyse selected runs to highlight the effect of grain-scale short-circuiting on the isotopic patterns, focussing on results for $\lambda = 10$ cm. This wavelength is chosen for illustration because (i) short signals at $\lambda \sim$ 10–30 cm are common on the isotopic records from polar ice cores, (ii) the shortest surviving signals (despite stronger diffusive smoothing at smaller $\lambda$) are of interest, and (iii) some signals with $\lambda$ as short as 10 cm are found on the high-resolution (5 mm) records from the WAIS Divide ($\delta^{18}$O and $\delta$D; Jones et al., 2017) and South Pole ($\delta^{17}$O, $\delta^{18}$O and $\delta$D; Steig et al., 2021). Perusing other ice-core datasets, we do not find signals at $\lambda \leq 10$ cm on the NGRIP $\delta^{18}$O record (Gkinis et al., 2014; 5 cm resolution), whereas the GRIP $\delta^{18}$O record (Johnsen et al., 1997; 55 cm resolution) and EPICA Dome C $\delta$D record (Grisart et al., 2022; 11 cm resolution) are too coarse for discerning signals at $\lambda \approx 10$ cm. However, the Dye-3 ice core exhibits annual variations in $\delta^{18}$O and $\delta$D as short as a few centimetres (Vinther et al., 2006; down to $\approx 2$ cm in their Fig. 6, and $\approx 5$ cm in their Fig. 5).

When addressing grain-boundary properties below, we use the qualitative descriptors for $D_b$ and $c$ introduced earlier (Fig. 2; Tables 1 and 2). Not all parameter combinations will be analysed for their isotopic patterns, e.g. not the patterns for medium-low $D_b$, which typically resemble and fall between the low and medium $D_b$ cases. As we shall see, the more interesting pattern transitions occur as $D_b$ varies from medium to high.

### 3.1 *Isotopic patterns in 3D*

#### 3.1.1 *Archetypal patterns at –32 °C: effects of grain-boundary properties and vein-water flow*

Figure 4 shows the predicted patterns at $T = –32$ °C, $\lambda = 10$ cm, and $w = 0$ for three runs with intermediate (5 nm thick) grain boundaries having high, medium-high and medium diffusivities. They illustrate the change from an axisymmetric "pole" pattern to a "3-spoke" pattern as $D_b$ increases, which is one of our key findings. We use the word "spoke" by analogy to the radial elements of a bicycle wheel; "pole" refers to a central peak without such elements. In each panel, the colour charts show the dimensionless variations in $\delta$ – i.e. $Re[H(r, \theta)\exp(ik_z z)]$ after normalisation by $H(1, L/2)$ (Sect. 2.6) – at three depths in the

range spanning $1\lambda$. We are looking down the domain in Fig. 1a and taking horizontal slices of its isotopic deviation, analogous to cuts perpendicular to a vertical triple junction in ice. The far-left plot shows depth profiles of isotopic variations at the mid-grain interior ($r = 1$, $\theta = L/2$; black curve), grain-boundary interior ($r = 1$, $\theta = 0$; blue) and along the vein (red). The first profile always has unit amplitude under the normalisation; in these runs, the latter two profiles have amplitudes very slightly less than 1, so they obscure the first profile when plotted. To emphasise where fast changes occur on each pattern, the colour scale is

always fitted to its maximal range of variations, and we use one of two colour schemes depending on whether $\delta$ at the vein is higher or lower than $\delta$ in the interior. Note that the isotopic signals decay in time following Eqs. (13) and (21), and the patterns occur on a background (mean) isotopic concentration that would first be subtracted when studying real ice samples.

First we analyse Fig. 4b – the medium-high diffusivity run – to explain salient features and how the patterns relate to the short-circuiting. This solution overall shows what the axisymmetric theories (Nye, 1998; Rempel and Wettlaufer, 2003; Ng,

2023) predict, with isotopes diffusing radially towards the vein (e.g. at $z = z_1, z_3$), up and down along the vein, and back into ice and radially outwards ($z_2$). As in those theories, these exchanges bypass solid diffusion in the ice to cause excess diffusion and accelerates the signal decay – the computed enhancement factor $f$ is 2.70 ($> 1$) – and they induce radial variations in $\delta$ that are the most rapid immediately outside the vein. These $\delta$-excursions cause the pole ($z_2$) and reverse-pole ($z_1, z_3$) patterns, which respectively reflect the role of the vein as a source and sink of isotopes in different horizontal sections in the short-circuiting.

The distinct depth intervals where isotopes diffuse radially inwards and outwards (identified by where $\delta$ in the ice exceeds $\delta$ in the vein, and vice versa) are indicated by white and grey bars on the far-left plot. In each interval, the patterns' strength (magnitude of horizontal isotopic variations) varies with depth according to the difference in $\delta$ between the vein and interior. But the patterns themselves hardly change with depth, except very near the transitions where the difference in $\delta$ between vein and interior changes sign (i.e. transitions between the bars). This near-invariance arises because $\lambda \gg b$, so that, away from

these transitions, vertical gradients in $\delta$ are much smaller than horizontal gradients in the system, and the diffusion problems

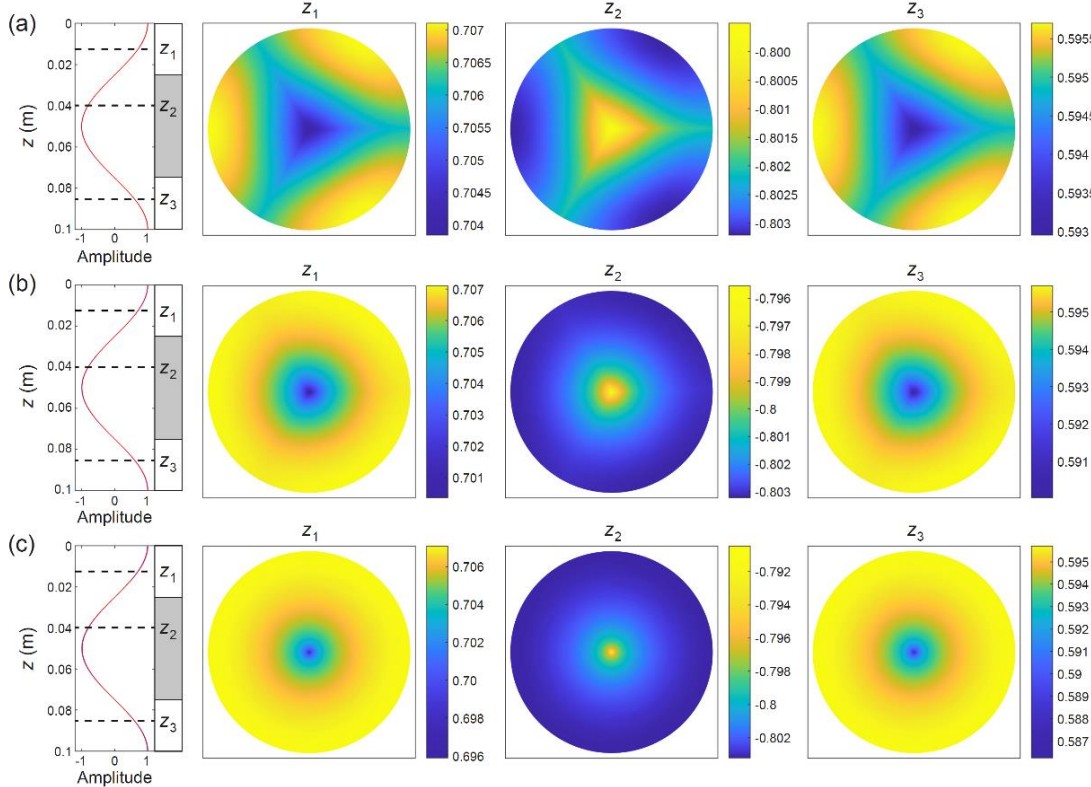

Figure 4. Horizontal isotopic patterns computed in three model runs with $T = -32$ °C, $\lambda = 10$ cm, $c = 5$ nm, $w = 0$ m yr$^{-1}$, and $D_b$ = (a) $1.5 \times 10^{-11}$ m$^2$ s$^{-1}$, (b) $1.5 \times 10^{-12}$ m$^2$ s$^{-1}$, (c) $1.5 \times 10^{-13}$ m$^2$ s$^{-1}$, compiled by sampling the $\delta$-variations in the annular domain of Fig. 1a at three depths ($z_1$, $z_2$, $z_3$). The colour charts reach out to the ice grain radius; the vein at centre is too small to be visible. One of two colour schemes is used, depending on whether $\delta$ at the vein exceeds $\delta$ in the grain interior or vice versa. In each run, the $\delta$-variations have been normalised by the value of $\delta$ in the mid-grain interior at $z = 0$ (Sect. 2.6), so the colour-scale numbering and amplitudes are dimensionless. At far left, curves show the depth profiles of $\delta$-variations at three sites – the vein wall (red), grain-boundary interior (blue), and mid-grain interior (black) – over a signal wavelength. The curves have very similar amplitudes in these runs, so the red curve overlies the other two. White and grey bars indicate the distinct depth intervals where the vein-versus-interior difference in $\delta$ has the same sign. The enhancement factors in these runs are $f$ = (a) 3.37, (b) 2.70, and (c) 2.64.

determining the pattern at different depths are similar[3]. At the transitions, as the vein-to-interior difference in $\delta$ switches sign,

the pattern flips from a pole to a reverse pole or the other way. Movie S1 shows the complete pattern evolution over $1\lambda$. The

stable "archetypal patterns" in the white and grey depth intervals are paired, with the same form but oppositely signed, so

---

[3] Mathematically, $\lambda \gg b$ (dimensionally) translates to $k_z \ll 1$ and $p_1 \ll 1$ in the scaled model of Sects. 2.4 and 2.5, so that Eqs. (22) to (25) for $H$ approximate a boundary value problem with terms representing vertical gradients neglected. Near where the isotopic pattern changes polarity, this approximation breaks down because $H \approx 0$ and those terms become comparable to the radial and azimuthal gradients.

hereafter we write "pole" for both pole and reverse pole. Detailed examination shows that within a narrow distance about each transition, the pattern evolves continuously, with a pole weakening to zero strength and reversing sign. This behaviour is not resolved in Movie S1 but can be gauged from its dynamic colour ranges.

The poles in Fig. 4b are not axisymmetric: they exhibit deformities reflecting the grain boundaries, whose impression is faint in this case. In contrast, Fig. 4a (high-diffusivity run, where $D_b$ is ten-fold) shows a much stronger grain-boundary imprint that causes 3-spoke patterns. Here, the vein plays a similar short-circuiting role as before; the archetypal patterns again flip where the vein-to-interior difference in $\delta$ switches sign. But isotopes also diffuse from ice to grain boundaries and along them to the vein (vice versa at other depths), and diffusion occurs vertically within the grain boundaries. Fast diffusion along them extend the poles to form the spokes and cause extra short-circuiting across the ice sectors, which raises the excess diffusion ($f$ = 3.37). The 3D isotopic field is more complex than in the run of Fig. 4b. Azimuthal variations are evident from the spokes, which indicate difference in $\delta$ between the ice interior and grain-boundary interior. The strongest azimuthal gradients occur just outside the vein next to grain boundaries, so lateral short-circuiting dominates near each ice sector's apex. The increased short-circuiting also reduces the vein-to-interior difference in $\delta$ compared to the last run (see colour-scale numbering).

Going the other way, lowering $D_b$ to medium diffusivity (Fig. 4c) suppresses the grain-boundary imprint and shrinks the poles, which still show corners but only at tiny radius. These changes are expected given the diminishing diffusion along grain boundaries. Although this solution thus closely approximates the axisymmetric solution of Rempel and Wettlaufer (2003) and Ng (2023), its $f$-value (2.64) is slightly less than what they found (2.65) for the same conditions in the absence of grain

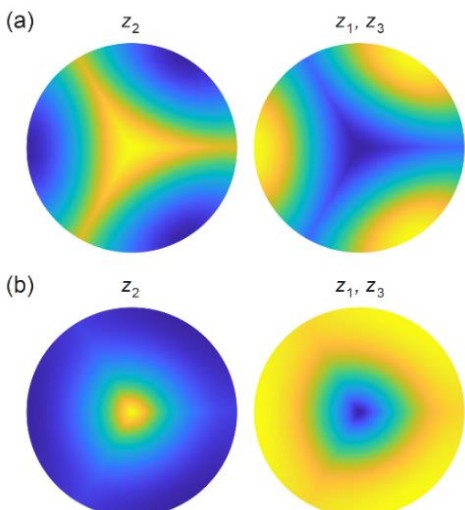

Figure 5. Archetypal isotopic patterns computed in two model runs assuming $T = -32$ °C, $\lambda = 10$ cm, $c = 10$ nm, $w = 0$ m yr$^{-1}$, and $D_b =$ (a) $1.5 \times 10^{-11}$ m$^2$ s$^{-1}$ and (b) $1.5 \times 10^{-12}$ m$^2$ s$^{-1}$, sampled at the same depths as those in Fig. 4 ($z_1$, $z_2$, and $z_3$), and shown with the scheme used there. The enhancement factors in these runs are $f =$ (a) 4.06 and (b) 2.77.

boundaries; this is also the case if $D_b$ is further reduced to medium-low or low. In other words, as we increase $D_b$ from the solid diffusivity $D_s$, $f$ decreases before rising. The initial decrease is due to radial short-circuiting of the ice near crystal apices by the grain boundaries, which reduces the radial gradients in isotopic concentration there, and thus the vein's short-circuiting effect; we will see more drastic examples of this behaviour shortly (Fig. 6). For the interested reader, Movies S2 and S3 document the depth-evolving isotopic patterns in the runs of Fig. 4a and 4c.

Next we vary the grain-boundary thickness $c$. Figure 5 presents archetypal patterns in two runs at –32 °C assuming thick grain boundaries ($c$ = 10 nm) of high and medium-high diffusivities. Compared to the runs in Fig. 4a and b, which used the same $D_b$ values, these patterns have more developed grain-boundary imprints and enlarged central excursions, and the associated enhancement factors are higher. As expected, thickening the grain boundaries here has a similar effect as raising $D_b$ in terms of enhancing grain-boundary short-circuiting, so the transition from a pole to 3-spoke pattern occurs at lower diffusivity. We experimented also with thin grain boundaries ($c$ = 1 nm), finding in this case that the pole-to-spoke transition shifts to higher diffusivity instead. The corresponding archetypal patterns will feature in Fig. 9 described later.

All experiments so far assume no vein-water flow, so their vertical isotopic variations at different positions ($r$, $\theta$) are in phase (Fig. 4, curves). What if $w \neq 0$? Figure 6 shows the results of four runs assuming intermediate and thick grain-boundaries with high and medium-high diffusivities, where we set $w$ to 5 m yr$^{-1}$, leaving other parameters unchanged. Ng (2023) explained

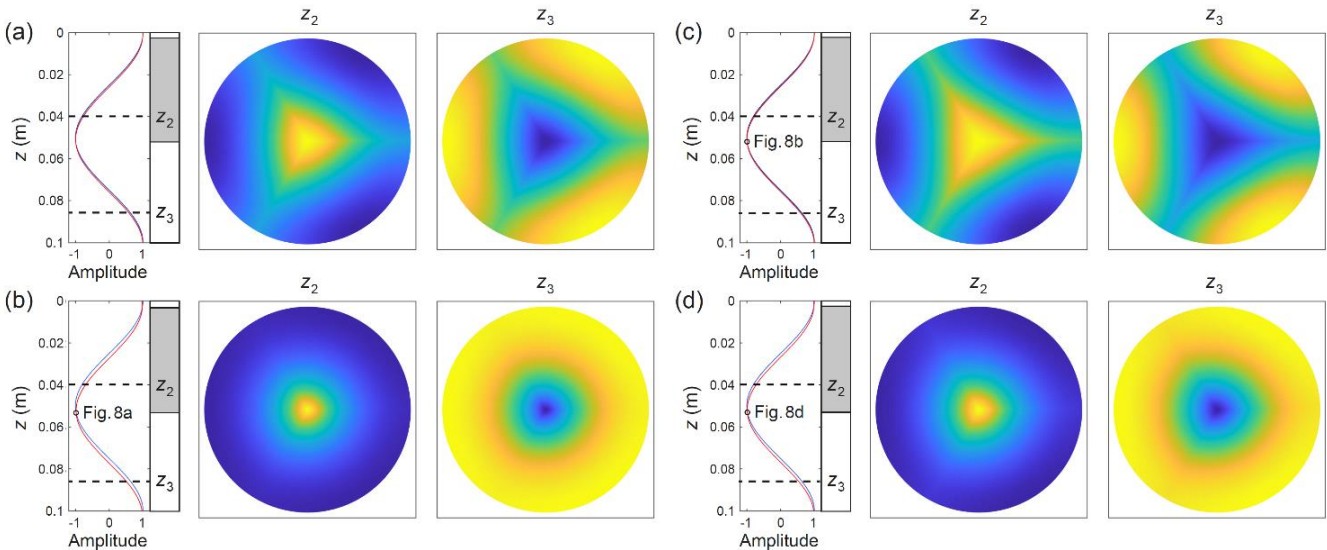

Figure 6. Archetypal isotopic patterns computed in four runs with $T$ = –32 °C, $\lambda$ = 10 cm, $w$ = 5 m yr$^{-1}$ (downward vein-water flow), and the grain-boundary properties (a) $D_b = 1.5 \times 10^{-11}$ m$^2$ s$^{-1}$, $c$ = 5 nm; (b) $D_b = 1.5 \times 10^{-12}$ m$^2$ s$^{-1}$, $c$ = 5 nm; (c) $D_b = 1.5 \times 10^{-11}$ m$^2$ s$^{-1}$, $c$ = 10 nm; (d) $D_b = 1.5 \times 10^{-12}$ m$^2$ s$^{-1}$, $c$ = 10 nm. The layout of Fig. 4 is used, but only the depths $z_2$ and $z_3$ are sampled, and we omit the colour range on each pattern, which is defined by the difference between the vertical isotopic profiles (curves at far left) for the vein wall (red), grain-boundary interior (blue), and mid-grain interior (black); the black curves are overlain by the blue curves in these runs. As in Fig. 4, all signal amplitudes are dimensionless. The enhancement factors in these runs are $f$ = (a) 4.56, (b) 5.96, (c) 4.97, and (d) 5.33.

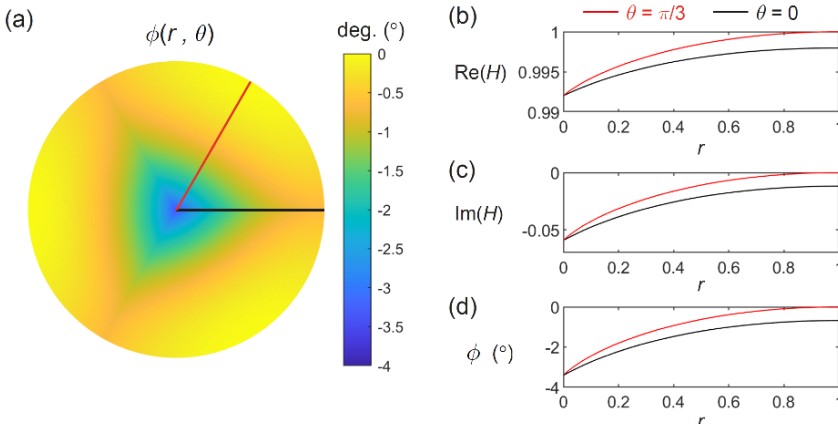

Figure 7. (a) Map of signal phase angle $\phi$ at $z = 0$ in the experiment of Fig. 6a. Corresponding radial transects of (b, c) the real and imaginary parts of the normalised solution $H$ and of (d) $\phi$ at $z = 0$, on $\theta = 0$ (black; i.e. grain boundary) and $\theta = \pi/3$ (red). The location lines in (a) and the curves in the other panels use the same colour coding.

that vein-water flow displaces the vein signal against the interior signal to induce a "shear layer" of phase-shifted isotopic variations outside the vein wall. In turn, the shear layer generates strong radial gradients in isotopic concentration in the ice near the vein, amplifying the diffusive isotope exchange between ice and vein to raise the level of excess diffusion. Figure 6 shows the vein signal displaced in all four runs. Each solution still has two transitions where the vein-to-interior difference in

$\delta$ switches, and paired archetypal patterns occupying equal depth intervals, within $1\lambda$. The archetypal patterns closely resemble the ones found earlier (cf. Figs. 4a, b & 5) because the vein-flow induced shear layers cause only subtle changes to them. Figure 7 depicts the shear layer on a map of $\phi$ for the run in Fig. 6a, showing also the radial transects of $H$ at $\theta = 0$ and $L/2$.

The non-zero imaginary part to $H$ causes a phase shift reaching $\approx -3°$ by the vein in this run. Unlike in Ng's (2023) axisymmetric theory, the shear layer here is triangular (non-circular) in planform due to lateral short-circuiting by the grain-

boundaries, so isotopic transport in 3D is complicated by both vein-water flow and the grain boundaries' presence.

All four experiments in Fig. 6 confirm the amplification of excess diffusion by $w$ anticipated by Ng's study: at each combination of $c$ and $D_b$, $f$ is higher than in the runs where $w = 0$ (cf. Figs. 4a–b and 5). Three effects involving the shear layer are noteworthy. First, Fig. 6 shows that at fixed $c$, $f$ is actually reduced as $D_b$ increases from medium-high to high. This arises from grain-boundary short-circuiting of the ice-crystal apices, which, in these runs, limits the radial isotopic gradients of the

shear layers so much that the reduced exchange between vein and ice offsets the enhanced exchange between grain boundaries and ice. Specifically, the higher is $D_b$, the weaker are those gradients at $w = 5$ m yr$^{-1}$, so the lower is $f$. This behaviour, which is observed in other runs with vein-water flow (e.g. $f$-values in red in Figs. 9 and 10 later), will be revisited in Sect. 3.2.

Second, the vertical phase shifts between the vein and interior signals increase the amplitude difference between these signals at most depths, and so strengthen the isotopic patterns (e.g. compare the curves in Fig. 6b to those in Fig. 4b). Vein-

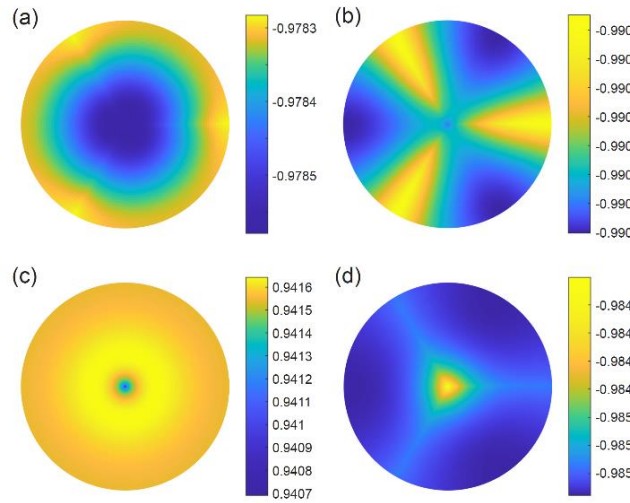

Figure 8. Transitory isotopic patterns from (a, b, d) three of the runs in Fig. 6 (see reference labels there) and (c) a run with $T = -32$ °C, $\lambda =$ 10 cm, $w = 5$ m yr$^{-1}$, $c = 1$ nm, and $D_b = 1.5 \times 10^{-13}$ m$^2$ s$^{-1}$. The patterns have low amplitudes because they occur near transitions in $z$ across which the vein-to-interior difference in $\delta$ switches sign. As before, the colour scales are dimensionless.

water flow thus makes the patterns easier to detect, even though it affects their form only in minor ways. Additional runs at $w$
> 5 m yr$^{-1}$ (not reported) show further increase in the phase shifts and pattern amplitudes with $w$. Note that higher pattern
amplitudes also result from shorter signal wavelength (e.g. Figs. S2 and S3 in Sect. S2, which show repeats of the runs in Figs.
4 and 6 for $\lambda = 2$ cm) or larger grain size (e.g. Figs. S6 and S7 in Sect. S4, which show repeats of the same runs for $b = 5$ mm),
but how $\lambda$ and $b$ affect the anatomy of the 3D isotopic fields will not be analysed extensively herein.

Third, when $w \neq 0$, the phase variations cause unusual patterns to appear in the narrow transitions across which an
archetypal pattern (pole or spoke) evolves to its opposite form. Figure 8 shows examples of these patterns, taken from the last
runs and an extra run at medium $D_b$. They include "wheels" with notable azimuthal variations mid-way along grain boundaries
(near $\theta = 0$, $L$, and $2L$ at $r \approx 1$) and "halos" where isotopic concentration varies with radius non-monotonically. Although we
mention them for completeness, we expect to find them rarely in measurements, because their small amplitudes likely fall
below measurement sensitivity and the sampling has to be made at precisely the right depth against the bulk signal.

3.1.2 *Pattern continuum at different temperatures*

Returning to the archetypal patterns, we summarise and elaborate on the insights gained so far on them with the aid of Fig. 9,
which puts them on the $c$–$D_b$ parameter space. The pattern type at –32 °C depends on the relative amount of vein and grain-
boundary short-circuiting. Thin, non-diffusive grain-boundaries give a pole pattern, since the short-circuiting is done mostly
by the vein. The axisymmetric solution is reproduced at the no-grain-boundary limit $c \to 0$ (dimensionlessly, $\varepsilon \to 0$) or when

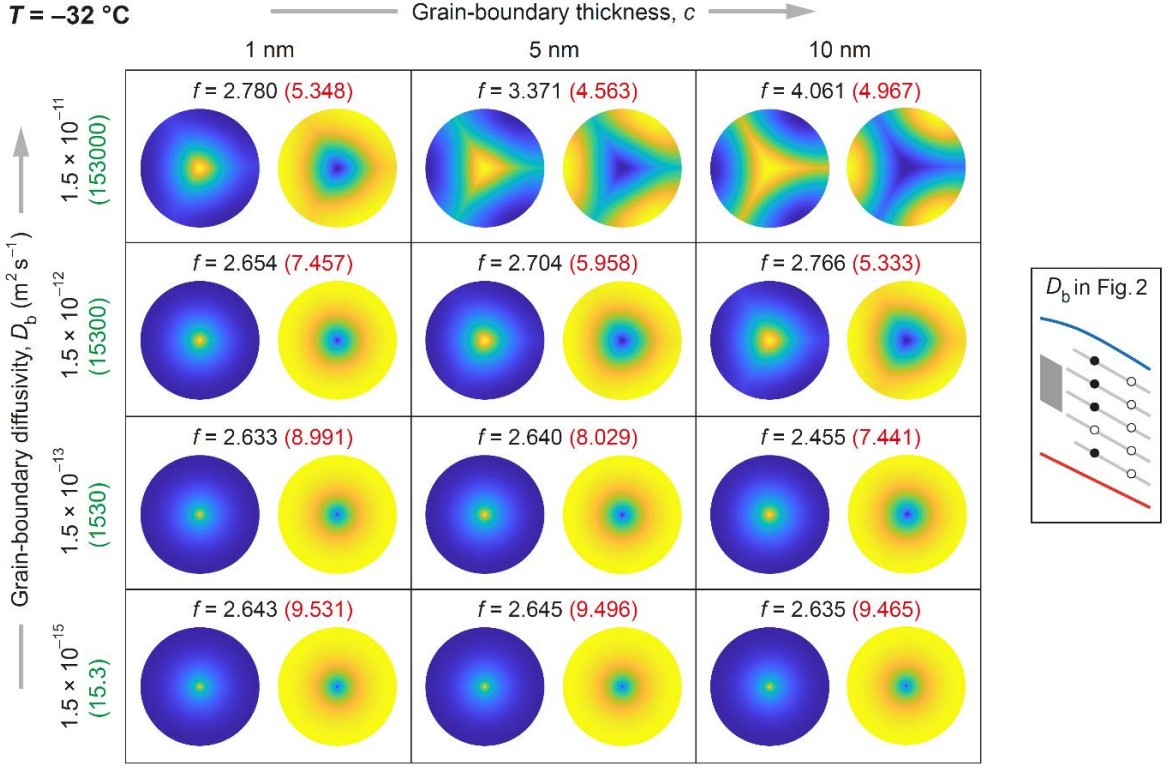

Figure 9. Dependence of archetypal patterns on grain-boundary diffusivity $D_b$ and thickness $c$ at –32 °C for signals with the wavelength $\lambda = $ 10 cm. Key on the right locates the four values of $D_b$ as black filled circles on the scheme of Fig. 2. Numbers in green give the corresponding diffusivity contrasts $\beta_b$. The isotopic patterns and the enhancement factors $f$ in black are for $w = 0$. Bracketed in red are the $f$-values when vein water flows at $w = 5$ m yr$^{-1}$, which produces patterns only slightly different from the ones shown (Fig. 6 and Fig. S4).

$D_b \rightarrow D_s$ (grain boundaries with the solid diffusivity; $\beta_b \rightarrow 0$). Thick, diffusive grain boundaries give 3-spoke patterns, as they serve as radial extensions of the vein in the 3D isotopic exchange; the higher is $c$ or $D_b$, the more developed are spokes. On the pattern continuum, the pole-to-spoke transition at –32 °C occurs roughly at medium-high $D_b$ – higher if the grain boundary is thinner. Figure 9 also indicates that a ten-fold increase in $c$ or $D_b$ leads to almost the same pattern, suggesting the thickess–diffusivity product entirely determines the pattern. However, our model analysis (Sect. 2.3) shows that $cD_b$ (or $\varepsilon(\beta_b + 1)$) isn't the sole control; $c$ and $D_b$ also act independently, which is why the ten-fold increases do not give identical enhancement factors.

How about other temperatures? Calculations at –52 °C for 10-cm long signals reveal a similar array of archetypal pole and spoke patterns on the parameter space (Fig. 10; cf. Fig. 9). Vein-water flow again modifies these patterns slightly (Fig. S5) but increases their amplitude and detectability strongly (we find this at other temperatures). That the pattern arrays for –52 and –32 °C bear close resemblance is unsurprising, because the diffusivity contrast $\beta_b = D_b/D_s – 1$ predominantly determines

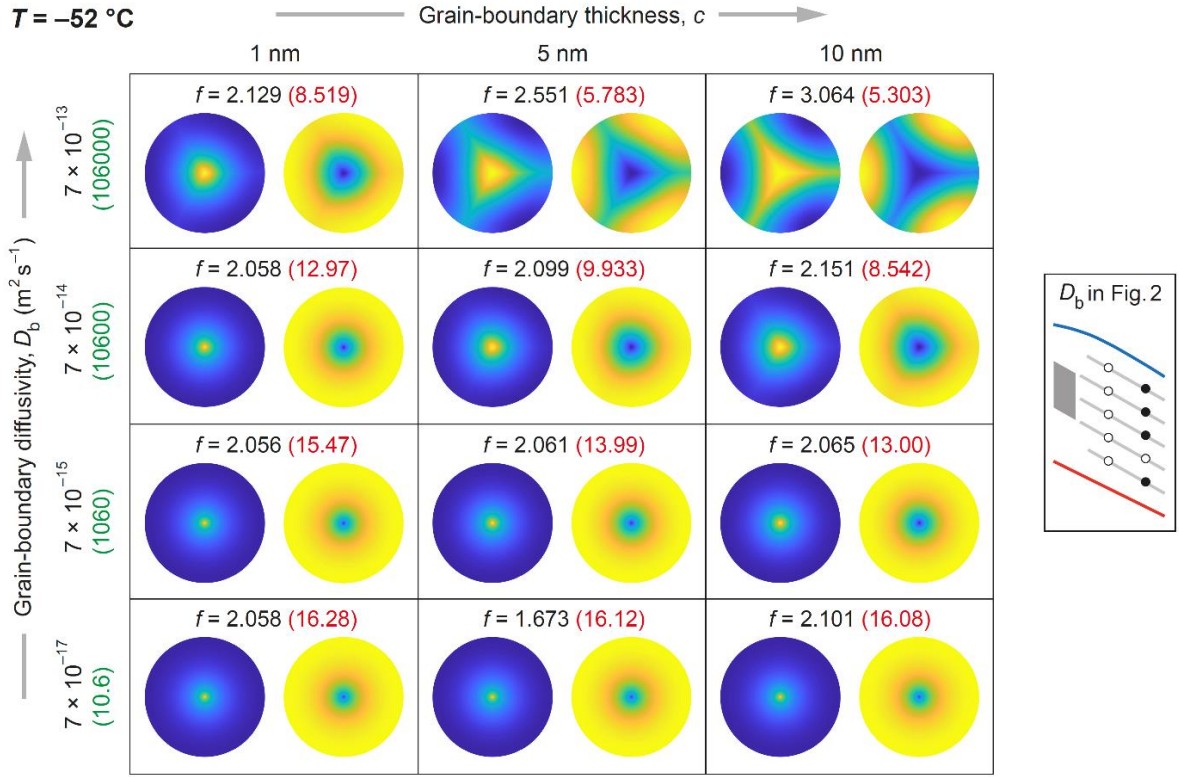

Figure 10. Dependence of archetypal patterns on $D_b$ and $c$ at $T = -52$ °C and $\lambda = 10$ cm. The layout of Fig. 9 is used here. The isotopic patterns and the enhancement factors $f$ in black are for $w = 0$. Bracketed in red are the $f$-values for vein-water flow at $w = 0.5$ m yr$^{-1}$, which produces patterns only slightly different from the ones shown (Fig. S5).

the pattern at each thickness $c$, and because our $D_b$ values for the two temperatures lie at similar distances above the solid-diffusivity curve (Fig. 2) and convert to similar $\beta_b$ values (Figs. 9 and 10). The liquid diffusivity $D_v$ also influences the patterns, but $\beta_v$ varies weakly with $T$, as $D_v(T)$ and $D_s(T)$ have similar slopes on the Arrhenius plot (Fig. 2). These considerations mean that we can predict the isotopic pattern at any temperature from $D_b$ and $c$, by calculating $\beta_b$ – or gauging it with Fig. 2 – and consulting the arrays in Figs. 9 and 10. For example, at –42 °C, for grain-boundaries with $D_b = 10^{-14}$ m$^2$ s$^{-1}$, Eqs. (1) and (12) give $\beta_b \approx 400$, while these $T$–$D_b$ data plot between the grey lines labelled medium-low and medium in Fig. 2. Both evaluations put the grain-boundary diffusivity between medium-low and medium on our descriptive scale, below the third row of patterns in Figs. 9 and 10, so we predict a pole pattern (regardless of the grain-boundary thickness). An interesting corollary is that isotopic patterns observed in real ice can be used to infer grain-boundary properties (Sect. 4.1).

Hitherto, we have focussed on using the results at $\lambda = 10$ cm to elucidate underlying interactions and pattern controls. For signals of other wavelengths at centimetre and decimetre scale, we find similar effects of $D_b$ and $c$ on the archetypal patterns; e.g. see Figs. S2 and S3 for results at $\lambda = 2$ cm. The patterns are weakly sensitive to $\lambda$ for the reason given earlier (footnote 3).

When $\lambda \gg b$, as is typical for isotopic signals in ice sheets, the diffusion problem for $\delta$ at different depths (within a signal wavelength) is similar, dominated by horizontal gradients, with terms representing vertical gradients being negligible. Note that our results in this section show that a given isotopic pattern does not indicate a fixed enhancement factor, as it can form under different conditions ($T$, $\lambda$, and $D_b$ and $c$ combinations).

### 3.1.3 *On pattern detectability*

We end the section with a few remarks related to pattern detection, in preparation for the work in Sect. 4.1. Since the patterns in Figs. 4 to 10 are based on the normalised $H$, their $\delta$-variation in absolute terms is given by their dimensionless amplitude, as shown by the colour scales or the difference between the vein and grain-interior isotopic profiles, multiplied by the true amplitude of the bulk vertical signal. This scaling conversion applies to both oxygen and deuterium. For instance, if the bulk signal (in $\delta^{18}$O or $\delta$D) is 10‰ peak-to-peak, then the pattern amplitudes in Fig. 4b, $\approx$ 0.005–0.007, translate to $\delta$-variations $\approx$ 0.025–0.035‰, whereas the much higher pattern amplitudes in the runs with vein-water flow in Figs. 6b and 6d, $\approx$ 0.1, translate to $\approx$ 0.5‰. Each result here is an approximation and underestimation, because the bulk signal has a scaled amplitude $\lesssim$ 1 (see the end of Sect. 2.6); we do not quantify the approximation exactly as it varies with the pattern.

The $\delta$-excursions of the patterns reported above have widths ~ 10–50% of the grain radius $b$. Although we do not study grain-size effects extensively, additional runs show that this qualitative finding holds at $b$ = 5 mm (Figs. S6–S9); thus, the $\delta$-excursions are dimensionally wider in coarse-grained ice. However, the pattern forms shift nearer the pole end of the pole-to-spoke continuum as $b$ increases (Figs. S6–S9). This is predicted by the scaled model (Sect. 2.4), where a larger $b$ has no effect on $\beta_b$ (this is the dominant control on pattern type; Sect. 3.1.2), reduces the dimensionless signal wavelength (the pattern is only weakly sensitive to this), raises the Péclet number $\chi$ (the flow-induced shear layer doesn't strongly alter the pattern; Sect. 3.1.1), and reduces the thinness $\xi$ and $\varepsilon\xi$ of the vein and grain boundaries and thus their short-circuiting efficiency: this causes the shift.

Finally, surface and thin sections on real ice will often cross triple junctions at oblique angles to their axes, yielding distorted isotopic patterns for them. Figure 11 exemplifies potential outcomes, made by sampling the solutions in Figs. 4a and 6a–b at tilts of 5°, 10°, 25°, and 50° from the horizontal (Movies S4–S7 show how they evolve as the azimuth of the section-normal varies). While this examination stretches our use of an idealised model geometry, which ignores the irregular shape of real grain boundaries and triple junctions (e.g. neighbouring junctions in real ice typically differ in orientation), these examples suggest that pole and spokes may still have visible impressions at moderate tilt. Generally though, on a given section, only some triple junctions have low or moderate tilts; other junctions with high tilts will have unrecognisable isotopic patterns.

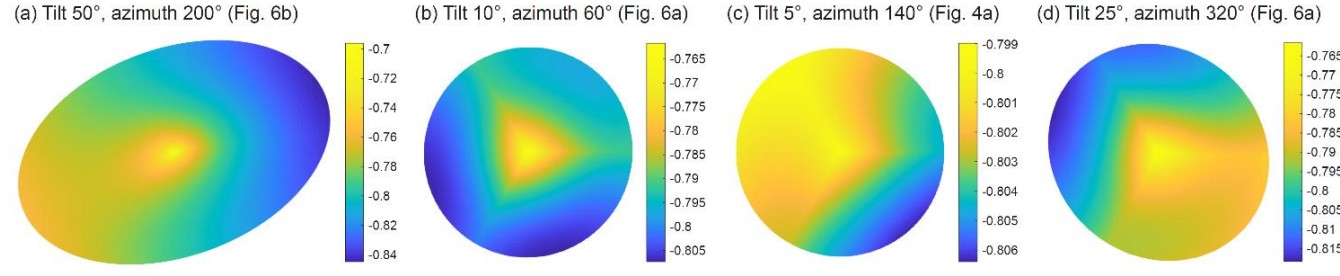

(a) Tilt 50°, azimuth 200° (Fig. 6b)  (b) Tilt 10°, azimuth 60° (Fig. 6a)  (c) Tilt 5°, azimuth 140° (Fig. 4a)  (d) Tilt 25°, azimuth 320° (Fig. 6a)

Figure 11. Isotopic patterns compiled by sampling several solutions in Figs. 4 and 6 at non-zero tilt from the horizontal (constant $z$ in our model), with the sampled sections meeting $z = z_2$ at $r = 0$. The tilt angle, tilt-axis azimuth and model run are indicated in each case. As before, the colour scales are dimensionless.

### 3.2 *Enhancement factor on bulk-ice diffusivity*

Of interest also is how much the presence of grain boundaries affects the enhancement factor $f$ measuring the excess diffusion (and acceleration of signal smoothing) above the rate due to monocrystalline diffusion. Here, we study this by examining the computed surfaces of $f$ as functions of vein-water flow velocity $w$ and signal wavelength $\lambda$.

Ng (2023) reported the surfaces $f(w, \lambda)$ at $T = -32$ and $-52$ °C for the axisymmetric (vein-only) system when $a = 1\ \mu m$ and $b = 1$ mm, which we reproduce in Fig. 12a and 12e as contour maps. Our computed surfaces accounting for grain boundaries show the same valley form as these maps, with $f$ increasing with $\lambda$ and $|w|$. Thus, vein-water flow amplifies excess diffusion in our system with grain boundaries by an amount independent of whether the flow is up or down, as in the vein-only system. Our model also predicts known trends of $f$ against the vein and grain sizes – $f$ increases with $a$ and decreases with $b$ (Rempel and Wettlaufer, 2003), which reflects the way these parameters control the efficiency and density of short-circuiting elements (Ng, 2023). Notably, a larger $b$ increases these elements' spacing relative to the signal wavelength, and so reduces the short-circuiting and $f$ towards 1 (no excess diffusion) asymptotically; this dependence is shown in Fig. 4 of Rempel and Wettlaufer (2003) for the vein-only system. Although we do not characterise this dependence in our system fully, model runs at $b$ from 1 to 5 mm in 1-mm increments confirm a similar behaviour (Fig. S10, Sect. S4). Furthermore, our system exhibits (i) lower $f$ at $-52$ °C than $-32$ °C when $w = 0$ and (ii) stronger modulation of $f$ by $w$ in colder ice (this is why our experiments at $-52$ °C use lower vein-flow velocities; e.g. compare the $f$-values in Fig. 10 resulting from $w = 0.5$ m yr$^{-1}$ to those in Fig. 9 from $w = 5$ m yr$^{-1}$). These aspects have been explained by Ng (2023) with scaling arguments that we do not repeat here.

We focus instead on how the surface $f(w, \lambda)$ deforms when we introduce grain boundaries and vary their diffusivity. Figure 12b–d and f–h present the results at $-32$ and $-52$ °C for intermediate grain boundaries with medium, medium-high and high $D_b$. The results are shown as difference maps $\Delta f(w, \lambda)$ referenced to the surfaces in Fig. 12a and 12e, because we find

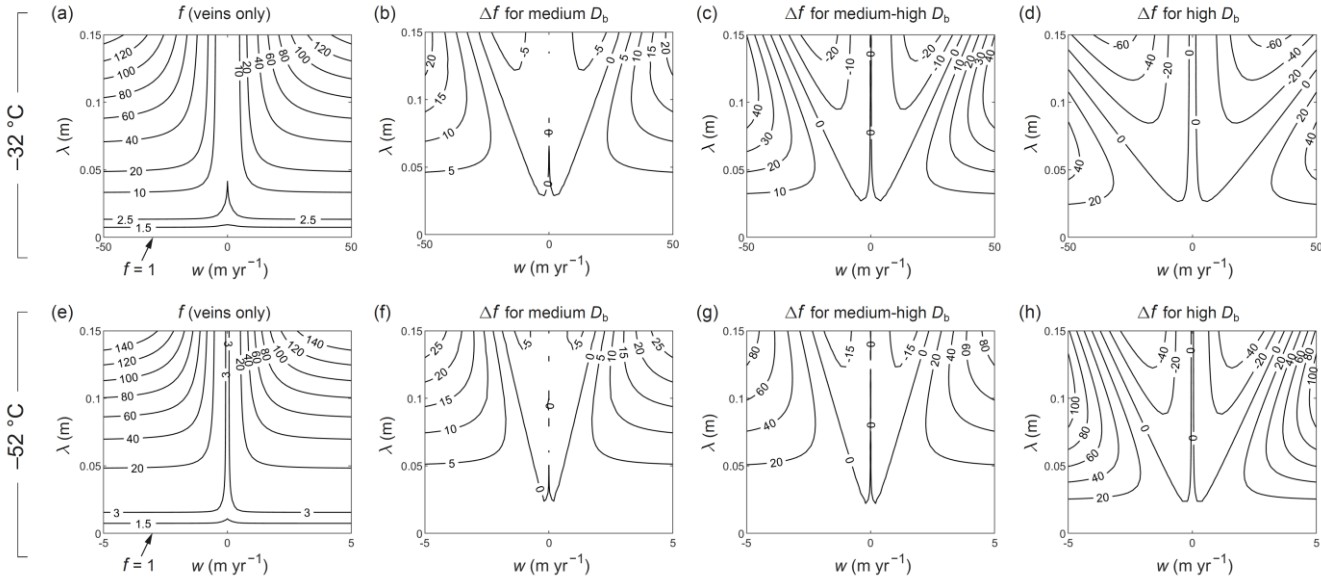

Figure 12. Impact of the presence and diffusivity ($D_b$) of grain boundaries on the level of excess diffusion for different vein-flow velocities $w$ and signal wavelengths $\lambda$ at –32°C and –52 °C when $c$ = 5 nm. (a, e) Contour maps of enhancement factor $f(w, \lambda)$ for the vein-only system without grain boundaries; data from Ng (2023). (b–d) $f(w, \lambda)$ reported as contour maps of the difference $\Delta f$ from (a), for our system at –32 °C when the grain boundaries have medium, medium-high, and high diffusivities. (f–h) Ditto for –52 °C, but referenced to (e).

visualising the changes by comparing different sets of contours of $f$ more difficult.[4] On the difference maps, the interesting feature is the wedges of negative $\Delta f$ straddling the $w = 0$ axis. They indicate reductions in $f$ caused by grain-boundary diffusion when vein-water flows. The reductions increase in magnitude with $D_b$ and $\lambda$, occur at relatively low vein-water velocities in $\lambda \gtrsim 2.5$ cm, and persist to higher vein-water velocities the longer is the signal. Between each pair of wedges is a narrow ridge at $w \approx 0$ where $\Delta f \gtrsim 0$, which matches our finding in Sect. 3.1 that $f$ typically increases with the degree of grain-boundary short-circuiting at zero vein flow (e.g. $f$-values in black in Figs. 9 and 10). Outside the wedges, $\Delta f$ is positive and increases steeply with $|w|$.

For the surfaces $f(w, \lambda)$, these differences mean that grain-boundary short-circuiting flattens their valley bottom – reducing $f$ there compared to the vein-only case for signals longer than ≈ 2.5 cm, while it raises $f$ only at sufficiently high vein-flow velocities. Because our runs at $\lambda = 10$ cm with vein-water flow (Sect. 3.1) assume values of $w$ inside the wedges, they predict less excess diffusion and lower $f$ when $D_b$ is increased ($f$-values in red in Figs. 9 and 10). The mechanism was explained in Sect. 3.1 through study of the 3D isotopic fields: with vein-water flow, diffusion along grain boundaries suppresses the flow-

---

[4] Our computed surfaces at low $D_b$ differ from Fig. 12a and 12e negligibly and could equally serve as the references.

induced shear layer by radial short-circuiting of its concentration gradients. The outcome thus rests on a competition: at low $|w|$, this effect overcomes the enhanced isotopic exchange between ice and vein due to the shear layer (so $f$ decreases overall); it is out-competed by the latter at high $|w|$ (whereupon $f$ increases). The wedge shape arises because the mechanism is more effective for longer signals, which develop weaker shear layers at a given $w$. For completeness, we provide the computed grids of $f$ in the paper's repository and show in Fig. S11 a companion version of Fig. 12 that plots $f$ instead of $\Delta f$.

In summary, although short-circuiting by diffusive grain boundaries leaves stable 3-spokes on isotopic patterns (Sect. 3.1), for decimetre-scale isotopic signals, it increases $f$ only at zero or high vein-water velocities, not at intermediate velocities. Thus, while the presence of grain boundaries or veins in glacier ice ($b \sim$ mm) always causes excess diffusion compared to the monocrystal, and while vein-water flow always increases the level of excess diffusion compared to no flow, whether more diffusive grain boundaries amplify the level has a mixed answer. However, this outcome does not affect the concept of using the grain-scale patterns to diagnose isotopic short-circuiting.

## 4 Discussion

### 4.1 *Detecting isotopic patterns*

Our calculations establish isotopic patterns around triple junctions as an inevitable consequence of excess diffusion that operates by vein and/or grain boundary short-circuiting. As highlighted in the Introduction, we propose looking for this grain-scale prediction in laboratory measurements on ice to test the Nye–Rempel–Wettlaufer genre of theories. Here we discuss this matter, drawing on the results in Sect. 3.1.

The crux is whether such tests reveal systematic excursions in $\delta$ around veins and grain boundaries like the predicted archetypal patterns. Also relevant is whether pole or three-spoke patterns (or both) are found to prevail in natural ice, but the current level of knowledge about the grain-boundary properties of ice precludes a clear expectation on this. Our model predicts a pattern type dependent on grain-boundary diffusivity and thickness – higher $D_b$ and $c$ favour spokes. Particularly, medium-high to high $D_b$ on our descriptive range (Fig. 2) is needed for spokes (Figs. 9 and 10; also, Figs. S2–S3 and S6–S9). This does not necessarily mean that spokes will be rarely observed, given substantial uncertainties about the extent to which impurities and crystallographic factors affect $D_b$ and $c$ (Sect. 2.2). Also, although the HCl bulk concentration ($\approx 0.01$ M) used by Lu et al. (2009) in their diffusivity measurements to explore the impurity effect is much higher than the typical concentration of Cl⁻ in ice cores ($\sim 1$–$10$ $\mu$M), natural ice contains myriad impurities. More likely, any detected isotopic patterns might give us a handle to assess the grain-boundary properties.

In terms of measurement technique, one based on laser-ablation (LA) sampling is promising. Bohleber et al. (2021) used LA-ICP-MS (laser ablation inductively-coupled plasma mass spectrometry) to map the elemental abundances (Na, Mg, Sr) on ice-core surface sections at 35 $\mu$m resolution, gaining new insights into impurity localisation at grain boundaries; see review by Stoll et al. (2023) also. Malegiannaki et al. (2023) have been innovating a system for mapping water isotope ratios in ice by coupling LA sampling with cavity ring down spectroscopy. The resulting isotopic maps will hopefully have a spatial

resolution as good as LA-ICP-MS and measurement sensitivity and accuracy in $\delta^{18}$O or $\delta$D sufficient for our proposed tests. On our simulated patterns at $\lambda = 10$ cm, the $\delta$-excursions have widths $\sim$ 10–50% of the grain radius for $b = 1$–5 mm (Sect. 3.1.3 and Sect. S4) and amplitudes $\approx$ 0.005–0.007 of the vertical bulk signal in the less favourable cases without vein-water flow (Fig. 4); that is, $\approx$ 0.01–0.02‰ if the bulk-signal variation is 5‰ peak to peak. This conversion example suggests achieving high sensitivity in $\delta$ to be the main obstacle for the LA-based technique to detect the patterns, while the technique should plausibly achieve sub-millimetre spatial resolution. But, as noted in Sects. 3.1.3 and 3.1.1, the pattern amplitudes are higher by an order of magnitude in those runs with vein-water flow and still higher at values of $w$ greater than those experimented by us; they are also higher for larger grain radii ($b > 1$ mm) and shorter signals ($\lambda < 10$ cm). Moreover, polar ice cores often exhibit decimetre-scale variations in $\delta$D of up to $\sim$ 20–40‰, in contrast to a few ‰ in $\delta^{18}$O, owing to the different dependences of $\delta$D and $\delta^{18}$O of polar precipitation on condensation temperature (Dansgaard, 1964). The conversion example above thus may be conservative, and optimistically we think that a measurement sensitivity of $\sim$ 0.1‰ has the potential of detecting the stronger isotopic patterns, especially if one targets short, large-amplitude signals in $\delta$D in coarse-grained ice.

For testing ice-core samples with this technique, our findings motivate mapping $\delta$ on horizontal sections at different depths. Figure 13 sketches an experimental design. The bulk isotopic signal should first be determined – e.g. by continuous flow analysis (CFA) measurements of a vertical strip – to guide where to make horizontal sections. If $w = 0$, locations likely to yield stronger and more detectable isotopic patterns are the peaks and troughs of the bulk signal, because a vein isotopic profile in phase with the bulk signal leads to the greatest pattern amplitude at those extrema, and pattern extinction at the bulk-signal inflexions (dotted curve in Fig. 13), where the predicted archetypal patterns switch sign (Sect. 3.1; Fig. 4). If the ice experienced vein-water flow, then we expect the vein isotopic signals to be displaced in the direction of $w$, so the extrema of the bulk signal might give weaker patterns than elsewhere (e.g. dashed-dotted curve in Fig. 13; also, Fig. 6). Given the difficulty of constraining $w$ at ice-core sites (see discussion by Ng (2023)) and other factors behind the shift (e.g. grain-boundary properties in a sample), the amount and direction of shift are not known a priori, so horizontal sections at several places – the bulk-signal extrema and inflexions and intermediate positions – are probably needed to obtain high-amplitude maps.

The map of $\delta$ from each horizontal section is processed by subtracting its section-mean value to isolate variations for spotting patterns akin to the predicted ones. If 2D mapping is not possible, linear transects may be used to detect anomalies in $\delta$ across grain boundaries. Whether mapping in 1D or 2D, obtaining the grain-boundary network independently by LA-ICP-MS or other measurements is desirable. If multiple sections could be made, we suggest sampling across the bulk-signal wavelength to look for the predicted pattern sign reversal (associated with the depth intervals where isotopes diffuse towards and away from veins; Sect. 3.1) and to characterise how pattern amplitudes vary with depth. For ice affected by short-circuiting, our model predicts (i) same-signed patterns on each horizontal section (triple junctions all showing either higher or lower $\delta$ than away from them) and (ii) pattern amplitudes cycling vertically on half the bulk-signal wavelength. Indeed, firm evidence for isotopic short-circuiting includes finding these depth-dependent relationships, besides the archetypal patterns.

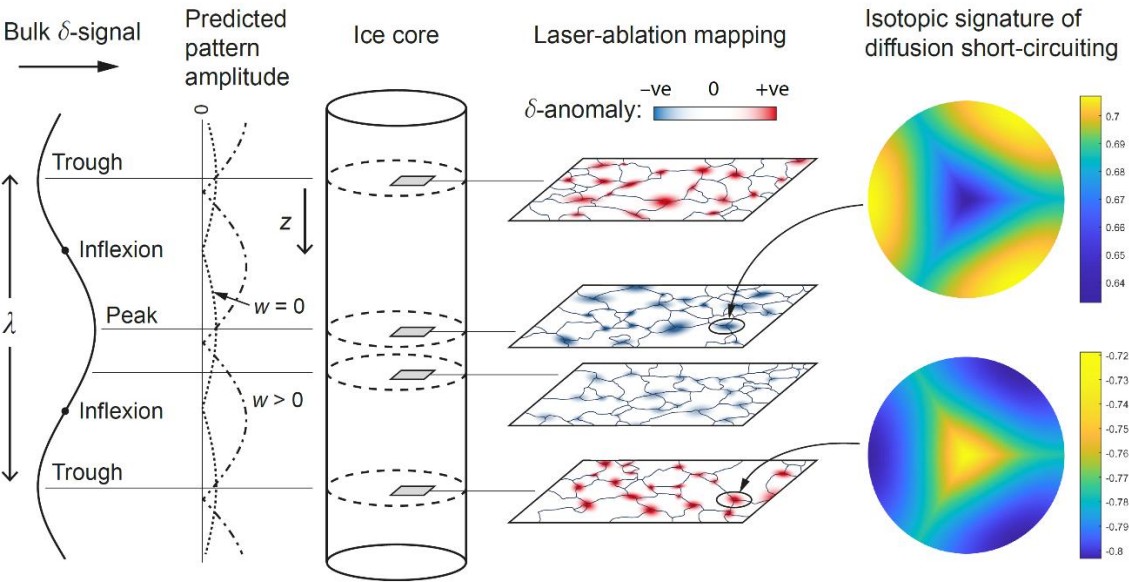

Figure 13. Experimental design for testing ice-core samples for the grain-scale signature of the isotopic short-circuiting causing excess diffusion. Horizontal sections are expected to show anomalies in $\delta$ around triple junctions, whose amplitudes vary with depth in association with the bulk $\delta^{18}O$ or $\delta D$ signal. Laser-ablation measurement is used to map the anomalies, some of them resembling the computed patterns in Figs. 4–6 and 8–11. If the ice experienced no or negligible vein-water flow ($w \approx 0$), the pattern amplitudes are strongest at the peaks and
troughs of the bulk signal, weaken away from these, and are faintest at inflexion points of the bulk signal (dotted curve). If $w > 0$, the amplitude variations are shifted vertically (dashed-dotted curve) so the patterns may be weaker near the bulk-signal peaks and troughs than elsewhere (e.g. Fig. 6). Cartoons right of centre illustrate the polarity and amplitude of the mapped patterns for $w = 0$. Images at far right illustrate how the patterns might look in detail in the case of spokes, on colour scales set to bracket their isotopic variations, after these have been non-dimensionalised by the bulk-signal amplitude.


On maps of $\delta$ yielding successful detection, we expect to see more varieties of triple-junction patterns than simple poles and spokes – patterns with different shape, amplitudes (even some with opposite sign), distortion levels, and patterns unlike the archetypes. Reasons include (i) a non-sinusoidal bulk signal, (ii) anisotopy in the diffusivity $D_s$ within crystals, (iii) curved grain boundaries and triple-junction angles deviating from 120° in real ice, (iv) textural variations in real ice (i.e. different vein
diameters, grain sizes and shapes, and triple-junction orientations (Fig. 11) sampled by each section; recall our simulations used fixed $a$ and $b$), (v) "out of plane" effects of veins and grain boundaries slightly above and below the section (discussed later in Sect. 4.2), and (vi) short-circuiting by subgrain boundaries (Sect. 4.2). Based on the computed pattern arrays in Figs. 9 and 10, the observed assemblage of archetypal or near-archetypal patterns may allow us to gauge the short-circuiting regime: spoke- (pole-) dominated assemblages would suggest thick, diffusive (thin, non-diffusive) grain-boundaries.

Vertical sections can also be mapped in the experiments. They will miss most (if not all) vertically-oriented triple junctions and hence not show our predicted patterns, but out-of-plane effects on them may generate excursions near triple

junctions (Sect. 4.2). Random linear transects across our computed patterns (e.g. Figs. 4 to 6) suggest that vertical sections will cross some of the $\delta$-excursions around grain boundaries in affected ice samples. Again, our model predicts their amplitude to vary vertically in unison – not necessarily in phase – with the bulk signal.

For isotopic maps from either vertical sections or a stack of horizontal sections, consistent phase relationship found between the pattern amplitudes and the bulk signal can be used to infer the direction and relative magnitude of vein-water flow (or its stagnancy). For an ice-core sample, this means the value of $w$ experienced when it was in-situ in the ice column.

     Which ice-core samples should be tested? High on the list are those from the Holocene part of the GRIP core (Johnsen et al., 1997, 2000), $\approx$ 15–18 ka BP in the WAIS Divide core (Jones et al. 2017), and MIS 19 ($\approx$ 3170 m) in the EPICA Dome
C core (Pol et al., 2010), given interpretation that they suffered excess diffusion (Sect. 1). Our model suggests choosing samples carrying bulk $\delta$-signals that are short ($\lambda$ as low as possible) with high amplitude. But since diffusion – especially excess diffusion – damps short signals efficiently, ideal samples may be challenging to find, and compromise between amplitude and wavelength may be necessary. As pointed out above, coarser-grained ice may show stronger patterns with wider excursions that are easier to resolve; this suggests samples deep in the ice column. Therefore, samples dated to MIS 19 from EPICA Dome
C (where $b \approx$ 6 mm; Fig. 8 of Ng (2023)) may be a good candidate; the high vein-water flow velocities ($\sim 10^2$ m$^2$ yr$^{-1}$) needed to model the diffusion lengths in that part of the core (Ng, 2023) also favours these samples. Separately, although the in-situ ice temperature and the time span of the bulk signal (e.g. whether it is annual, centennial, or millennial) do not matter in these tests for the occurrence of excess diffusion, samples with high dust or microparticle content are best avoided because blockage of veins (maybe grain boundaries also) hinders the theorised short-circuiting. We pause with these general ideas on sample
selection here and leave dedicated considerations to future studies.

     It is equally important to test ice-core samples apparently unaffected by excess diffusion. Together with the (purportedly) affected samples, they may help us understand the origin and pattern of occurrence of excess diffusion in individual or multiple cores. From the perspective of the Nye–Rempel–Wettlaufer framework, which includes Ng's (2023) and our present model, it is puzzling why excess diffusion occurs in a patchy manner in ice cores. All three cores mentioned above have depth intervals
where the signal-decay rate or diffusion length can be explained with monocrystalline diffusivity without excess diffusion (Johnsen et al., 2000; Pol et al., 2010; Jones et al., 2017). Yet the short-circuiting theories predict $f > 1$ always, because veins and grain boundaries are always present. One possibility is that blocked or disconnected veins prevent excess diffusion on some intervals, whereas on other intervals, dissolved impurities migrate to grain boundaries (e.g. Bohleber et al., 2021) and then to the veins, thickening them to turn on excess diffusion. A study that tests unaffected and affected samples for grain-
scale isotopic short-circuiting and maps their impurities simultaneously (with LA-ICP-MS) might shed light on the enigma.

     Artificial ice samples can also be tested. Manufacturing these with bulk isotopic signals may be non-trivial, and the long time for isotopic patterns to stabilise seems impractical (the time scale $b^2/D_s$ in Eq. (17) gives 16, 27, and 78 years at –5, –10, and –20 °C, respectively, for $b$ = 1 mm; longer for higher $b$) and may limit insights to the transient stages of short-circuiting.


4.2 *Model limitations and extensions*

Real isotopic patterns at the grain scale will be more varied and complex than predicted because our model geometry is idealised: its cell-like regularity (Fig. 1) ignores grain size and shape variations, for instance (Sect. 4.1). Before finishing, we consider several important limitations of the model in this respect.

First, the true geometry has many non-vertical veins and grain boundaries. To gauge their effects, one might try to add horizontal veins and grain boundaries to the model geometry. Ice with grain size $b \sim$ mm will have many such elements in one $\lambda$ (if $\lambda \sim$ cm to dm). Spaced at intervals $\sim b$, they extend the diffusion pathways laterally from our system. We expect the associated $\delta$-excursions, which modify the isotopic field near these elements, to be thin vertically, just as the radial (azimuthal) excursions around veins (grain boundaries) in our current model are thin. Between the new excursions, the field should

resemble the one computed by us. Therefore, on a given horizontal section, we should still find triple-junction patterns in $\delta$ like the predicted ones, for neighbouring grains crossed by the section near their waist (thick excursions from high $c$ and $D_b$ might distort these patterns). But grains crossed near their top and bottom will show strongly-affected patterns, as their sampled junctions and boundaries lie near or within the new excursions. Consequently, real maps of $\delta$ will show out-of-plane distortion due to horizontal and sub-horizontal veins and grain boundaries above and below the section. The impact of the (sub-)

horizontal elements on the level of excess diffusion is harder to predict. They may increase the isotopic exchange between veins and ice to raise $f$ overall or short-circuit the vertical system sufficiently to reduce $f$ (we infer this possibility from Sect. 3.2, where we saw diffusion along grain boundaries weakening the flow-induced shear layer around veins).

      Second, veins and grain boundaries in the real system generally are not stationary but migrate continually. Their 3D motion will cause lopsided or asymmetric isotopic patterns. Modelling the outcome requires quantifying the relative rates of

isotopic-field evolution and this motion, accounting for the statistical distribution of vein and grain-boundary velocities and impurity factors, which lies beyond the scope of this paper.

      Third, small-angle boundaries within crystals, i.e., subgrain boundaries, may act as short-circuiting pathways that distort the isotopic patterns. In their experiments on ice with gaseous HCl, Dominé et al. (1994) and Thibert and Dominé (1997) interpreted measured depth profiles of the HCl concentration in single crystals for fast HCl diffusion along these defects, using

this to explain the high value and high scatter of apparent diffusivities in their samples. Dominé et al. (1994) estimated the HCl diffusivity along the small-angle boundaries (accounting for segregation of HCl there) at –5 to –15 °C to be $\sim 10^7$ times greater than the "true" HCl diffusivity in the crystal lattice away from the defects, which the two studies estimated to be probably around $10^{-16}$ $m^2$ $s^{-1}$ at –5 to –35 °C. Consequently, one could conjecture fast diffusion of oxygen and deuterium isotopes along the same defects, which would extend the short-circuiting network of grain boundaries and veins into crystals.

Its impact on excess diffusion and the isotopic patterns would presumably depend on the density of small-angle boundaries. In exploring such conjecture, a key question of how well the findings for HCl translate to water self-diffusion, and one way to investigate this is to repeat the experiments on water stable isotopes, instead of HCl.

Studies using the enhancement factor $f$ from our model to simulate signal evolution and diffusion-length profiles in ice cores should bear in mind the above limitations, which apply equally to the short-circuiting theories of Nye (1998), Johnsen et al. (2000), Rempel and Wettlaufer (2003), and Ng (2023). In terms of building more realism and sophistication upon these theories, we are near the end of the road with using simple analytical models to capture the coupled diffusion across ice, veins, and grain boundaries. Looking forward, overcoming the limitations in the mathematical description seems challenging and may require approximate approaches (e.g. using multiscale or homogenisation methods to derive bulk diffusivity) or direct numerical simulation tracking complex mobile interfaces.

## 5   Conclusions

If vein and grain-boundary short-circuiting is responsible for excess diffusion in ice, then isotopic imprints similar to our computed archetypal pole and spoke patterns will occur at the grain scale. The $\delta$-excursion of each imprint reflects isotopic exchange between ice and the short-circuiting pathways, its polarity showing whether the pathways act as a sink or source of isotopes for the crystal lattice. For ice with millimetre grain size, our model predicts excursions ~ 10–50 % of the grain radius – thus, at least 0.1 mm wide, with $\delta$-variations whose amplitude is proportional to the bulk isotopic signal and ranges from ~ $10^{-2}$ to $10^{-1}$ times of its amplitude. Mapping the isotopic patterns probably requires a minimum instrumental sensitivity of $\sim$ 0.1‰ and spatial resolution of a few tens of microns or better. Pattern detectability is improved in ice that carries short bulk signals with high amplitude, has higher mean grain size, and experienced vein-water flow in the ice column, as these factors promote stronger excursions.

These predictions motivate testing ice for these signatures of excess diffusion and the short-circuiting mechanism by mapping their isotopic concentration at high resolution. Given ongoing development of laser-ablation measurement techniques, we outlined a scheme for conducting the tests on 2D sections of ice from ice cores, with thoughts on sample selection (Sect. 4.1; Fig. 13). The proposed tests are independent from known ways of inferring excess diffusion from the signal-decay rates or estimated diffusion lengths on ice-core isotope profiles, which can diagnose its occurrence but not its underlying mechanism.

Our modelling elucidates the controls on the isotopic signatures. Although the isotopic diffusivity ($D_b$) and thickness ($c$) of grain boundaries in ice are poorly constrained, our results show that thin, non-diffusive grain boundaries yield pole patterns, whereas thick, diffusive grain boundaries yield spoke patterns. Figs. 9 and 10 show the predicted continuum of pattern types on the $c$–$D_b$ parameter space, which can be used with the observed patterns in an ice sample to infer its grain-boundary properties. The grain-boundary diffusivity affects the pattern via the parameter $\beta_b = D_b/D_s - 1$, in which $D_b$ and the monocrystalline diffusivity $D_s$ both depend on temperature (Fig. 2). Our results also revise current estimates of the enhancement factor $f$ quantifying excess diffusion above $D_s$. For the full system with veins and grain boundaries, vein-water flow amplifies excess diffusion and increases $f$, as in the vein-only system (Ng, 2023). The presence of grain boundaries can increase or reduce $f$ compared to the vein-only system, depending on the vein-water flow velocity $w$; $f$ is increased at

sufficiently high $w$ for decimetre-scale or shorter signals (Sect. 3.2). The model predicts polycrystalline ice always to exhibit some excess diffusion ($f > 1$) unless the veins are blocked by solid particles or disconnected.

In future extensions, it may be possible to use the assemblage of grain-scale isotopic patterns in ice samples to quantify their level of excess diffusion and constrain their vein and grain-boundary properties. The proposed tests and this avenue will help us understand why excess diffusion occurs on some parts of ice cores and not others.

## Appendix A

Table A1: Variables and parameters in our mathematical model.

| Symbol | Description [square brackets indicate values used in our calculations] |
|---|---|
| $a$ | Liquid-vein radius [1 $\mu$m] |
| $b$ | Mean grain radius [1 mm] |
| $c$ | Grain-boundary thickness [see Table 1 for values] |
| $D_b$ | Grain-boundary (isotopic) diffusivity [see Table 2 for values] |
| $D_s$ | Isotopic diffusivity in ice or "solid diffusivity"; Eq. (1) |
| $D_v$ | Isotopic diffusivity in vein water or "liquid diffusivity"; Eq. (2) |
| $f$ | Enhancement factor on isotopic diffusion rate |
| $F(r, \theta)$ | A part of the function $H$ in the numerical method (Sect. 2.6) |
| $G(r)$ | Radial function representing variation of $H$ along grain boundaries |
| $H(r, \theta)$ | Complex function encapsulating the isotopic pattern |
| $J$ | Number of numerical grid points in the radial direction |
| $k_z$ | Signal wavenumber ($= 2\pi/\lambda$) |
| $k_r$ | Parameter linked to wavenumber in the signal-decay calculation (Eq. (14)) |
| $L$ | $= 2\pi/3$, the angle between grain boundaries |
| **M** | Sparse-banded matrix in the numerical method (Sect. 2.6) |
| $N$ | Number of numerical grid points in the azimuthal direction |
| $N_s, N_v, N_b$ | Concentrations of trace isotope ($^{18}$O or D) in ice, vein, grain boundaries |
| $N_{s0}, N_{v0}, N_{b0}$ | Concentrations of major isotope ($^{16}$O or H) in ice, vein, grain boundaries |
| $p_1, p_2, p_3$ | Parameters used in the calculation of Sect. 2.5 |
| $r$ | Radial coordinate |
| $R$ | Transformed radial variable |
| $R_{max}$ | Vein-wall position in the transformed radial variable |
| $s$ | Square root of the eigenvalue $s^2$ in the problem for $H$ (Sect. 2.5) |
| $t$ | Time |
| $T$ | Temperature |

| | |
|---|---|
| **v** | Eigenvector in the numerical solution |
| $w$ | Vein-flow velocity [values in 0–50 m yr$^{-1}$ used in experiments] |
| $x$ | Chebyshev collocation point positions in the spectral method |
| $z$ | Depth |
| $\alpha$ | Fractionation coefficient, $\approx 1$ [see Sect. 2.3 for information] |
| $\beta_b$ | Diffusivity contrast of grain boundary to ice ($= D_b/D_s - 1$) |
| $\beta_v$ | Diffusivity contrast of water to ice ($= D_v/D_s - 1$) |
| $\delta$ | Isotopic deviation |
| $\varepsilon$ | Dimensionless grain-boundary thickness ($= c/a$) |
| $\zeta$ | Complex decay-rate parameter |
| $\theta$ | Azimuthal coordinate |
| $\lambda$ | Signal wavelength ($= 2\pi/k_z$) |
| $\xi$ | Dimensionless vein radius or dimensionless radial position of vein wall ($= a/b$) |
| $\phi$ | Phase angle of isotopic signal in the $z$-direction |
| $\chi$ | Péclet number (ratio of vein-flow advection to monocrystalline diffusion) |

**Code and data availability**

The MATLAB code for solving the model equations and computed grids of the enhancement factor are archived at https://doi.org/10.15131/shef.data.xxxxxxxx. Use https://figshare.com/s/e42a421e53b02efdaa0f during the review stage.

**Video supplement**

Movies S1–S7 are available at https://doi.org/10.15131/shef.data.xxxxxxxx.
Please use https://figshare.com/s/37cfa936be37610f24e8 during the review stage.

**Supplement**

Sections S1–S4, Movies S1–S7, and Figures S1–S11 are available at https://doi.org/10.15131/shef.data.xxxxxxxx.
Please use https://figshare.com/s/37cfa936be37610f24e8 during the review stage.

**Author contribution**

F. S. L. Ng designed the study, performed all analyses, and wrote the paper.

**Competing interests**

The author has declared that there are no competing interests.

**Acknowledgements**

I thank the City University of Hong Kong for library access during my visits, which gave me a peaceful environment for doing some of the model calculations; acknowledge support from the University of Sheffield Institutional Open Access Fund for covering the publication cost; and thank two anonymous reviewers for their highly constructive comments on the manuscript. For the purpose of open access, the author has applied a Creative Commons Attribution (CC BY) licence to any Author Accepted Manuscript version arising from this submission.

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
