# Peer review of "The grain-scale signature of isotopic diffusion in ice"

_EGUsphere, 2024_

## Referee Comment (RC2)

**The grain-scale signature of isotopic diffusion in ice by Felix S.L. Ng**

**General Comments**

This paper deals with the modeling of the diffusion of stable water isotopes occurring in polycrystalline ice. The overall motivation of this work is to explore mechanisms that could explain the "excess diffusion" of stable water isotopes observed in some ice-cores. Understanding and quantifying diffusion of stable isotopes in ice cores is important as they are used as proxies for the reconstruction of past temperatures and post-deposition effects need to be quantified for a fine interpretation of these paleo-records.

Specifically, the paper models the combined impact of liquid water veins and crystals grain boundaries in the diffusion of water isotopes and to quantify the enhancement of the smoothing of isotopic signals (compared to a scenario considering diffusion through the ice phase only).

To my view, the paper offers three main points:
- The equations to model an idealized polycrystalline ice, including a water vein and grain boundaries, and a numerical methodology to solve them.
- To demonstrate the formation of specific spatial patterns, around the water vein and crystal boundaries, which constitute a robust signature of the proposed mechanisms to explain excess diffusion.
- To quantify the role of water veins and grain boundaries to the enhancement factor of diffusive-smoothing.

The paper is well constructed, follows a sound methodology, and offers valuable results for the interpretation of ice cores. I therefore think it is suited for the The Cryosphere and suggests minor corrections.

My main remarks are:

**1 –** I think the paper would benefit from a more systematic use of citations to published literature. It would help the reader not familiar with the overall scientific context and with the techniques used in the paper. Some examples are given in the specific comments below, but for instance one example that comes to my mind is L249 that refers to "past theories" without mentioning them (I guess the author meant Rempel and Wettlaufer et al., 2003).

**2 –** The author assumes equilibrium fractionation at the interfaces between the grains boundaries, the vein, and the ice (L230). From what I understand, this assumption simplifies the system of equations (as $N_v$ and $N_b$ are now direct functions of $N_s$ at these interfaces). Is it a common assumption for the diffusion of isotopes in polycrystalline ice? Also, since there is transport in the problem, I think that non-equilibrium fractionation could be possible. If so, what could be the impact on the diffusion of water isotopes? Can we justify that it is small?

**3 –** The paper mainly focus on the role of water vein and grain boundaries, and so little emphasis is put on grain size (which was studied before if I got it correctly). I understand this choice, but I'm still wondering to what extent the enhancement factor depends on the grain size. Section 3.2 refers to Rempel and Wettlaufer (2003) for this effect and mentions

that the proposed model predict the same trends. Is it possible to mention them in the paper and to estimate what kind of impact can we expect the grain sizes variations observed in ice cores?

Still on grain size, the abstract and Section 4.1 mention that the patterns of δ correspond to 10 to 50% of the grain radius. Do they really scale with grain radius (i.e. does the 10-50% holds for very large or very small grain sizes or it is valid for millimeter grain sizes only)?

**Specific remarks:**

**L19** I would rather write things in percents when possible, as I find them more natural to read ("10-50% of the grain size radius" rather than "0.1-0.5 of the grain size radius").

**L29** Add references to the use of δ18O and δD in ice cores.

**L32** Perhaps add general references to signal smoothing in ice cores.

**L38** Add references to Beyond-EPICA Oldest Ice and the Little Dome C project.

**L89** Add references to the use of laser-ablation in ice cores studies and its possibilities.

**L139-141** References would be helpful here to justify that grain boundaries needs to be inferred from bulk measurements and that they size can greatly increase with impurities.

**L189** Does "bulk" here refers to the combination of ice and grain boundaries? If so perhaps precise it so bulk is not confused with the ice only (my first thought when I read the paragraph).

**Eqs 3 to 5** I would specify that the sources terms in Eq 4 and 5 also corresponds to BCs for Eqs 3 and 4.

**L230** Is there a reference for $\alpha \sim 1$?

**Eqs 9 and 10** I would specify here that Eqs 9 and 10 are directly derived from Eqs 4 and 5, under the assumption of equilibrium fractionation at the vein/boundary/solid interfaces.

**L250** Add references to the mentioned past theories.

**L261** The form of the trial solution was explicitly chosen as a dampened traveling wave in Eq. 13 but this sentence could leave the feeling that the migration of the signal has been somewhat proven (rather than assumed). I would reformulate with something along the lines "As shown by the phase angle XX, the trial solution of Eq. 13 includes a potential vertical migration at the velocity XX."

I also propose to replace "downward" by "vertical", as I assume it would depend on the sign of the water flow.

**L282** Doesn't $k_r$ require to be solved as well (along side the H pattern)?

**L299** It makes physical sense that the $\xi_i$ is zero when there is no water flux (because then the isotopic signal is not traveling), but can it be justified a priori from the system of equation (perhaps already done in the literature)?

**L300** Why only the slowest-decaying eigenmode?

**L301-305** Perhaps add references about the mathematical techniques discussed here and examples where they have been used.

**L363** I do not really understand the normalization. It is normalized by the value of H at r=1, θ=L/2, and z=0 (the last point missing in the text if I'm not wrong)?

**L373** Why these particular choices for λ and w? I guess they correspond to typical values found in ice core, but extra context and justification would be a plus.

**Section 3.1** This section presents and discusses a lot of results and is therefore a bit dense to read. Perhaps adding sub-sections could help follows the argument set up by the author. For instance something as
*3.1.1 Impact of grain boundary and water flow on isotopic patterns*
*3.1.2 Impact of temperature*
*3.1.3 Impact of the bulk isotopic signal*

**L402** I would say that "the solution *overall* shows what the axisymmetric theories predict", as the presence of small distortions near the grain boundaries discussed later are a notable difference with previous theories.

**L449** I understand that the phase angle (defined in L261) is constant at the given position z, but is it really equals to zero? Or perhaps the phase angle here refers to the phase of H ($\tan^{-1}(Im(H)/Re(H))$).

**L494-500** I would move this paragraph on the impact of the non-orthogonality between thin sections and triple junctions later in the text (L510). Here, I found that it cuts the reasoning on the impact of $D_b$ and c on the spatial patterns.

**L499** "Only some triple junctions may show the archetypal patterns" Is it because the tilt make them unrecognizable or because the pattern itself will deviate from the model (model assumptions being not met in real ice)?

**L539** I would precise that the fact the patterns are unchanged does not presume that the enhancement factor is the same.

**L544** I do not understand why the bulk δ signal has a scaled amplitudes <= 1 in the case of short-circuiting.

**L608** Perhaps add references for the impurities.

**L634** Add a reference to CFA technique.

**Technical remarks:**

**Title** I would precise that the isotopes in questions are (stable) water isotopes.

**L1** I would also mention here that the article discusses stable water isotopes.

**L11** I would use "could explain" rather than "can explain".

**L19** Perhaps rephrase to "[…] and variations in δ ranging from 1 to 10% of the amplitude of the bulk isotopic signal. This sets […]"

**L20** Not sure the mention of the specific technique of laser-ablation mapping is necessary here.

**L126-127** "Equation" and "Figures" should be written in full at the start of a sentence. Also both the abbreviations "Eqn." and "Eq." are used in the article. I think TC guidelines require the use of "Eq.".

**L169** The authors sometimes uses here ≈ when the symbol ~ is used elsewhere to mean what I think is the same idea. Perhaps use the word "about".

**Table 1** If possible perhaps also put the investigated grain boundary thicknesses.

**L204** corroborates

**L206** t is not defined at this point.

**Eq. 6** I would give the definition of δ a bit above, when it first appears L228.

**L324** I would rather define BC the first time Boundary Condition is used.

**L327 and 660** I am not familiar with the use of "/" to mean respectively.

**Figure 4** I It might be a problem from my printing set up, but I cannot easily read the scale and axis values. Perhaps try to increase the resolution of the Figures and to increase the font of the Figures' text. Also the use of dashed could help visualizing the black curve when it covered by the blue one.

**L545** Remove the hyphen in amplitudes

**L546** "Underestimation"?

**L550** Perhaps replace "single-crystal diffusion" by "crystal (or solid) diffusion alone" or by "diffusion in single-crystal".

**L640** Is the reference to Ng (2023)?

---

## Author Comment (AC2)

**Response to RC2**  [egusphere-2024-1012]

Black: Reviewer comments. Blue: Authors' response.

**RC2**: Anonymous Referee #1

**General Comments**

This paper deals with the modeling of the diffusion of stable water isotopes occurring in polycrystalline ice. The overall motivation of this work is to explore mechanisms that could explain the "excess diffusion" of stable water isotopes observed in some ice-cores. Understanding and quantifying diffusion of stable isotopes in ice cores is important as they are used as proxies for the reconstruction of past temperatures and post-deposition effects need to be quantified for a fine interpretation of these paleo-records.

Specifically, the paper models the combined impact of liquid water veins and crystals grain boundaries in the diffusion of water isotopes and to quantify the enhancement of the smoothing of isotopic signals (compared to a scenario considering diffusion through the ice phase only).

To my view, the paper offers three main points:
- The equations to model an idealized polycrystalline ice, including a water vein and grain boundaries, and a numerical methodology to solve them.
- To demonstrate the formation of specific spatial patterns, around the water vein and crystal boundaries, which constitute a robust signature of the proposed mechanisms to explain excess diffusion.
- To quantify the role of water veins and grain boundaries to the enhancement factor of diffusive-smoothing.

The paper is well constructed, follows a sound methodology, and offers valuable results for the interpretation of ice cores. I therefore think it is suited for the The Cryosphere and suggests minor corrections.

Thank you for your review and these appraisals, and for providing a very detailed set of comments and suggestions to help me improve the manuscript.

My main remarks are:

**1** – I think the paper would benefit from a more systematic use of citations to published literature. It would help the reader not familiar with the overall scientific context and with the techniques used in the paper. Some examples are given in the specific comments below, but for instance one example that comes to my mind is L249 that refers to "past theories" without mentioning them (I guess the author meant Rempel and Wettlaufer et al., 2003).

Yes, the submitted text did skip some references and use umbrella terms for them in places to save space. I can revise the text to be specific in those instances. (I have noted the items in your "Specific Remarks" below about references.)

**2** – The author assumes equilibrium fractionation at the interfaces between the grains boundaries, the vein, and the ice (L230). From what I understand, this assumption simplifies the system of equations (as $N_v$ and $N_b$ are now direct functions of $N_s$ at these interfaces). Is it a common assumption for the diffusion of isotopes in polycrystalline ice? Also, since there is transport in the problem, I think that non-equilibrium fractionation could be possible. If so, what could be the impact on the diffusion of water isotopes? Can we justify that it is small?

Equilibrium fractionation was assumed in Johnsen et al.'s (2000) study, and assumed in the equations of Rempel and Wettlaufer (2003), which form the backbone of this part of my model – the formulation on L230 is closely analogous to their formulation. No fractionation was considered in Nye's 1998 model.

The possibility of non-equilibrium fractionation is interesting generally. However, I don't think that I will be elaborating the text to explore it. There are two reasons:
    (1) I don't think that current knowledge allows us to address the process properly as a detail in this study. Experimental data are lacking on non-equilibrium fractionation at the ice–water interface. I looked and struggled to find any relevant literature. Borrowing concepts or results from other fields, e.g., from isotopic studies on igneous and metamorphic rocks or from other phase boundaries (ice–vapour or water–vapour) involve bold and uncertain extrapolations to the ice–water system, and stacking of more theories on top. Those theories will have still more assumptions behind them.
    (2) More importantly, I don't think that the consideration of non-equilibrium fractionation aids this study. The purpose of the study is to bring together existing theoretical elements into a single representative model, and calculate results from it to inform observational testing of those existing theories. The "elements" referred to here include the veins considered by Nye (1998) [who didn't model grain boundaries], the grain boundaries explored by Johnsen et al. (2000) [who considered veins but didn't couple grain-boundary and vein diffusion], the extension to a finite liquid diffusivity by Rempel and Wettlaufer (2003) [whose model has veins but not grain boundaries], and the vein-water flow studied by Ng (2023) [whose model lacks grain boundaries]. Thus, the combined model is necessarily an extension of the individual models/theories. But the purpose of the study is *not* to build even more complexity into it to capture all conceivable aspects of the physical world, to form a 'kitchen sink' model and put it forward as a complete, robust description of excess-diffusion mechanism. Instead, the study emphasises testing the *existing* theoretical framework. Even if, under some conditions, non-equilibrium fractionation has a large effect in the system, the study's purpose is still to report predictions from the existing theories.
    I will modify L230 to emphasise that equilibrium formulation is assumed in the model and in the theories informing its formulation, so it is clear to the reader.

**3** – The paper mainly focus on the role of water vein and grain boundaries, and so little emphasis is put on grain size (which was studied before if I got it correctly). I understand

this choice, but I'm still wondering to what extent the enhancement factor depends on the grain size. Section 3.2 refers to Rempel and Wettlaufer (2003) for this effect and mentions that the proposed model predict the same trends. Is it possible to mention them in the paper and to estimate what kind of impact can we expect the grain sizes variations observed in ice cores?

Thank you for this suggestion. Yes, I can summarise the trends from grain-size variations following L556-557, probably in a passage of several lines. In sketching the kind of impact, I will strive to give description that errs on the quantitative side, probably through giving examples of values. I think it is a difficult to go beyond that, given an already large number of parameters in the model (and here we're considering variation in one more parameter).

Still on grain size, the abstract and Section 4.1 mention that the patterns of δ correspond to 10 to 50% of the grain radius. Do they really scale with grain radius (i.e. does the 10-50% holds for very large or very small grain sizes or it is valid for millimeter grain sizes only)?

Good question. Yes, they scale with the grain radius, because the calculations are made on the scaled model using a radial variable that has been non-dimensionalised with the grain radius b (Sect. 2.4). This finding holds for very large or very small grain sizes, as long as the grain boundaries are much thinner than the vein radius, and the vein radius is much less than b; see Eq. (19) and the line after it. Strictly speaking, the pattern is scaled exactly to the grain size when the Péclet number representing the vein-water flow velocity doesn't change. But, as reported in the Results, the pattern changes only slightly with vein-water flow velocity (because noticeable changes occur only very close to the vein).

**Specific remarks:**

Thank you for the comments below. I will put them to use in the revision. Most of them are minor, some concerning clarity and precision (including addition of references), so I hope it is ok for me to omit a point-by-point answer in this initial reply, and write only to those items where I have more substantive things to say.

**L19** I would rather write things in percents when possible, as I find them more natural to read ("10-50% of the grain size radius" rather than "0.1-0.5 of the grain size radius").
**L29** Add references to the use of δ18O and δD in ice cores.
**L32** Perhaps add general references to signal smoothing in ice cores.
**L38** Add references to Beyond-EPICA Oldest Ice and the Little Dome C project.
**L89** Add references to the use of laser-ablation in ice cores studies and its possibilities.
**L139-141** References would be helpful here to justify that grain boundaries needs to be inferred from bulk measurements and that they size can greatly increase with impurities.
**L189** Does "bulk" here refers to the combination of ice and grain boundaries? If so perhaps precise it so bulk is not confused with the ice only (my first thought when I read

the paragraph).

**Eqs 3 to 5** I would specify that the sources terms in Eq 4 and 5 also corresponds to BCs for Eqs 3 and 4.

**L230** Is there a reference for $\alpha \sim 1$?
Yes, I can provide references.

**Eqs 9 and 10** I would specify here that Eqs 9 and 10 are directly derived from Eqs 4 and 5, under the assumption of equilibrium fractionation at the vein/boundary/solid interfaces.
**L250** Add references to the mentioned past theories.

**L261** The form of the trial solution was explicitly chosen as a dampened traveling wave in Eq. 13 but this sentence could leave the feeling that the migration of the signal has been somewhat proven (rather than assumed). I would reformulate with something along the lines "As shown by the phase angle XX, the trial solution of Eq. 13 includes a potential vertical migration at the velocity XX."
Thanks for this suggestion. I will probably borrow or adapt this wording to improve the emphasis of the existing passage.

I also propose to replace "downward" by "vertical", as I assume it would depend on the sign of the water flow.

**L282** Doesn't $k_r$ require to be solved as well (along side the H pattern)?
Thank. In this section (Sect. 2.5), I will add a sentence to say that after finding the eigenvalue s^2, k_r is found via Eq. (26) and then used to calculate the enhancement factor f.

**L299** It makes physical sense that the $\xi_I$ is zero when there is no water flux (because then the isotopic signal is not traveling), but can it be justified a priori from the system of equation (perhaps already done in the literature)?
If I understand your idea correctly, this aspect has already been treated by Ng (2023), on page 3067 in that paper, 4 lines above their Eq. (10), by symmetry considerations. In the current study, I refrain from repeating this highly technical aspect, as it is far from the thread of Sect. 2.5 explaining the approach to solving the equations.

**L300** Why only the slowest-decaying eigenmode?
I will add a sentence to clarify that the other eigenmodes decay faster, leaving the slowest mode to be the one observed. Therefore we are interested in calculating the slowest-decaying mode.

**L301-305** Perhaps add references about the mathematical techniques discussed here and examples where they have been used.

**L363** I do not really understand the normalization. It is normalized by the value of H at r=1,

θ=L/2, and z=0 (the last point missing in the text if I'm not wrong)?

The z-variations have been handled by the exponential in Eq. (21), i.e., in mathematical terms, they are treated as the sum of Fourier modes in *z* in a "separation of variables" approach to solving the scaled model in Eq. (18).

Therefore, as indicated on L284, H is a function of r and θ only, not a function of z. However, your comment is probably made with the amplitude of the vertical signal in mind. I can clarify (around L363) that H at r=1, θ=L/2 describes the maximum value of that amplitude, to help the reader.

**L373** Why these particular choices for λ and w? I guess they correspond to typical values found in ice core, but extra context and justification would be a plus.

The λ-range is motivated partly by both the wavelengths explored in the earlier models of Nye (1998) and Rempel and Wettlaufer (2003), and partly by typical wavelengths seen on ice-core isotope records. The w-range is motivated by theoretical predictions of the water percolation through ice sheets – Reviewer 1 has asked me to add context about that. Yes, I will add suitable background for both choices in this section.

**Section 3.1** This section presents and discusses a lot of results and is therefore a bit dense to read. Perhaps adding sub-sections could help follows the argument set up by the author. For instance something as

*3.1.1 Impact of grain boundary and water flow on isotopic patterns*
*3.1.2 Impact of temperature*
*3.1.3 Impact of the bulk isotopic signal*

Agree. Thanks for coming up with these headings – this is very helpful. I will consider how to sub-section the material and what headings best suit the different parts.

**L402** I would say that "the solution *overall* shows what the axisymmetric theories predict", as the presence of small distortions near the grain boundaries discussed later are a notable difference with previous theories.

**L449** I understand that the phase angle (defined in L261) is constant at the given position z, but is it really equals to zero? Or perhaps the phase angle here refers to the phase of H ($\tan_{-1}(\mathrm{Im}(H)/\mathrm{Re}(H))$).

**L494-500** I would move this paragraph on the impact of the non-orthogonality between thin sections and triple junctions later in the text (L510). Here, I found that it cuts the reasoning on the impact of $D_b$ and c on the spatial patterns.

**L499** "Only some triple junctions may show the archetypal patterns" Is it because the tilt make them unrecognizable or because the pattern itself will deviate from the model (model assumptions being not met in real ice)?

**L539** I would precise that the fact the patterns are unchanged does not presume that the enhancement factor is the same.

**L544** I do not understand why the bulk δ signal has a scaled amplitudes <= 1 in the case of short-circuiting.

Yes, I will add a sentence to explain.

**L608** Perhaps add references for the impurities.

**L634** Add a reference to CFA technique.

**Technical remarks:**

As for the "Specific remarks" above, I will put your suggestions here to use, and hope that it is ok for me to omit a point-by-point response to them in this initial reply.

**Title** I would precise that the isotopes in questions are (stable) water isotopes.

Thanks for this suggestion. This is a good idea. I will consider whether to clarify this in the title or in the abstract (or in both places).

**L1** I would also mention here that the article discusses stable water isotopes.

Yes, certainly.

**L11** I would use "could explain" rather than "can explain".

**L19** Perhaps rephrase to "[…] and variations in δ ranging from 1 to 10% of the amplitude of the bulk isotopic signal. This sets […]"

**L20** Not sure the mention of the specific technique of laser-ablation mapping is necessary here.

**L126-127** "Equation" and "Figures" should be written in full at the start of a sentence. Also both the abbreviations "Eqn." and "Eq." are used in the article. I think TC guidelines require the use of "Eq.".

**L169** The authors sometimes uses here ≈ when the symbol ~ is used elsewhere to mean what I think is the same idea. Perhaps use the word "about".

**Table 1** If possible perhaps also put the investigated grain boundary thicknesses.

**L204** corroborates

**L206** $t$ is not defined at this point.

**Eq. 6** I would give the definition of δ a bit above, when it first appears L228.

**L324** I would rather define BC the first time Boundary Condition is used.

**L327 and 660** I am not familiar with the use of "/" to mean respectively.

**Figure 4** I It might be a problem from my printing set up, but I cannot easily read the scale and axis values. Perhaps try to increase the resolution of the Figures and to increase the font of the Figures' text. Also the use of dashed could help visualizing the black curve when it covered by the blue one.

**L545** Remove the hyphen in amplitudes

**L546** "Underestimation"?

**L550** Perhaps replace "single-crystal diffusion" by "crystal (or solid) diffusion alone" or by "diffusion in single-crystal".

**L640** Is the reference to Ng (2023)?

---

## Author Response (AR1)

**Response by the author detailing the revision of manuscript "egusphere-2024-1012"**

25 July 2024

Dear Editor of *TC*,

Please see in this file a **point-by-point description of my revision** of the manuscript **(in red)**, alongside the **comments of the ED and RC1 and RC2 (in black)** and, for reference, **my responses during the interactive discussion phase (blue)**.

**OL** refers to Old Line numbers, and **NL** to New Line numbers.

Following the advice of ED, RC1 and RC2, I have undertaken a detailed revision of the manuscript, including making the following changes:
- Elaborated the study's scope and rationale in the Introduction, in response to the first point raised by RC1;
- Illustrated all of the modelling with computed results at signal wavelength $\lambda$ = 10 cm (instead of 2 cm), including using the corresponding new versions of Figs. 4, 5, 6, 7, 8, 9, 10, 11; updating all related captions and textual passages and pitching the focus of analysis/discussion on those results; updating the supplementary Figs. S4 and S5 that are counterparts to Figs. 9 and 10; and updating the Movie Supplements;
- Added Supplementary results to show the effect of larger grain size $b$ (up to 5 mm) in Figs. S6 to S10 in Sect. S4, so Reviewer 2 can see that the dependence of enhancement factor $f$ on $b$ described in Sect. 3.2 is indeed found in the model, and that the excursion widths are still 10-50 % of $b$ when $b$ ranges up to 5 mm. In the Supplement file, I also insert the original $\lambda$ = 2 cm results (in two figures in Sect. S2), which are relevant for comparative purpose;
- Grouped some of the material in the Supplement into sections (Sects. S1 to S4) to help the reader;
- Explained my "descriptive scale" for grain-boundary diffusivity $D\_b$ much earlier, in Sect. 2.2 (instead of Sect. 3.1), at the same time improving Figure 2 and its caption, updating Table 2 (the original Table 1) to show the descriptive scale for $D\_b$, and adding a new Table 1 to list the grain-boundary thicknesses used in the model runs;
- Clarified the "normalisation" of $H(r, \theta)$ in a new passage at the end of Sect. 2.6 (NL 418-424), at the same time stating the dimensionless unit of the normalised isotopic patterns in the captions of relevant figures;
- Explained the choices of model-run parameters near the start of Sect. 3.1, notably for the vein-water flow velocity $w$ and signal wavelength $\lambda$, signposting there also that grain-size effects will be addressed briefly;
- Improved the passages about isotopic fractionation (NL 190-198, NL 276-279);
- Added sub-section headings (3.1.1, 3.1.2, 3.1.3) to help partition the existing material of Sect. 3.1 to help readers, and moved the paragraph OL 494-500 and the accompanying Old Fig. 9 to NL 648-654 (in Sect. 3.1.3) and to New Fig. 11, following RC2's suggestions;
- Mentioned diffusion along small-angle boundaries when discussing the model limitations in Sect. 4.2) (see NL 843-854);
- Added Table A1 to list all key mathematical symbols;

- Attended to every comment of RC1 and RC2, following most of their suggestions or coming up with alternative solutions, in order to improve the text as far as possible.

Alongside these revisions, I refined the text to ensure coordination between the changes, to remove redundancies, and to address minor typos or mistakes (e.g. Gkinis et al. (2014) on OL 36 was missing from the original reference list, and the front end of Eqn. (15) had missed out the factor $1/\pi b^2$; these have now been readded).

Thank you for considering and handling the manuscript. I wish to thank the reviewers for their highly constructive comments and suggestions.

Best wishes,
Felix Ng

--- --- --- ---
**ED:** Editor comments

Dear Author,

Both reviews are in general positive but do make constructive suggestions that will help you prepare an improved revised version. In particular:

Dear Editor,

Thank you for handling the manuscript. I hope it is appropriate for me to post this reply as my *Final Response* in the interactive review process, as I have replied to the two reports RC1 and RC2, responding in detail to their points (except those on minor corrections regarding typos, local precision in wording, and inclusion of references). I appreciate the detailed, constructive and insightful comments by the reviewers.

My plan of revising the manuscript consists of (i) implementing changes in response to the reviewers' points and suggestions – I think that I should be able to address most of them, and (ii) extending the text in the Introduction (probably also in Sect. 4.1 or Conclusions) to put across the rationale of the study more strongly.

In the revision, I have done most of (i) (more below) and all of (ii). The study's rationale and scope are now substantiated on NL 69-72, NL 75-79 and a new paragraph on NL 103-112.

- Please ensure that all hypotheses and simplifications that you use are explicitly stated. You may also briefly discuss their limitations.
- Please ensure that all your equation developments are reasonably easy to follow and to reproduce by scientists in the field. Your choice of parameter values must be justified. Reviewer 2 makes a number of remarks on this aspect that deserve your attention. Among these, your choice of grain size, and the impact of choosing other grain sizes, should be discussed.

  Done – please see point-by-point responses below to RC1 and RC2.

As described in my response to the RC1 and RC2 reports, I will (i) give more background behind the chosen ranges of wavelength and vein-water flow velocity, (ii) say more about the impact of grain-size variations on the isotopic pattern and the enhancement factor, and (iii) explain Figure 2 better in order to link the experimental and model values of D_b (grain-boundary diffusivity) more clearly.

- The relationship to ice core measurement must be improved, as recommended by Reviewer 1.

  Yes, I plan to follow Reviewer 1's advice of illustrating the study with computed results for a signal of longer wavelength of 10 cm or so. Accordingly, in Sect. 4, I will pitch the inferences regarding the amplitude of the predicted patterns and the required experimental measurement sensitivity to detect them in terms of those results (rather than results for a 2 cm signal). As described in my reply to RC1, numerical results were computed across the parameter space so I have the relevant data at hand, and while Figs. 4–11 will be updated, their isotopic patterns and pattern transitions will not change much.

  I now illustrate the study with computed results for a signal with wavelength of 10 cm. Done. Please see details below.

- I mentioned in my initial evaluation that the possible impact of small angle boundaries may also deserve consideration as diffusion short-circuits. Their presence is likely because of ice deformation. Please consider addressing this topic. How would considering these defects affect your equations and conclusions?

  Thank you for this idea. Yes, I can discuss this topic in the revised text. Its discussion will enrich the manuscript. This will probably be added to Sect. 4.2, where model limitations are discussed.

  The revision will probably explore a thread similar to the following. I perused the studies by Dominé et al. (1994) and Thibert and Dominé (1997), who interpreted experimentally-measured depth profiles of HCl concentration in ice single crystals for the occurrence of fast diffusion along small-angle boundaries inside crystals, i.e., sub-grain boundaries, and attributed this process as the cause of the high value and scatter of the apparent diffusivities of their samples. Dominé et al. (1994) estimated the HCl diffusivity along the small-angle boundaries (accounting for segregation of HCl there), at -5 to -15 °C, to be $\sim 10^7$ times greater than the "true" HCl diffusivity in the crystal lattice (away from the defects), which the two studies estimated to be probably $\sim 10^{-16}$ $m^2\,s^{-1}$ at -5 to -35 °C. Therefore, when considering oxygen and deuterium isotopes, it is possible to conjecture that fast diffusion along small-angle boundaries could raise the amount of excess diffusion and complicate the isotopic patterns, by extending the short-circuiting network of grain boundaries and veins further into crystals; the outcome would depend on the density of small-angle boundaries. In exploring such a conjecture, a key question of whether (or how reliably) the findings for HCl translate to the problem of water self-diffusion, and one

way to examine this question is to repeat the kind of experiments performed by these authors on water stable isotopes, instead of HCl.

Given that my study's purpose is to compute the predictions of a short-circuiting model encapsulating the theories of Nye (1998), Johnsen et al. (2000), Rempel and Wettlaufer (2003) and Ng (2023), for informing the testing of those four theories, I think that adding other processes to the current mathematical model should not be necessary (in any case, this is difficult to do when the self-diffusivity of $H_2O$ along small-angle boundaries and the thickness of these are not well constrained).

Done. Please see NL 843-854.

Please explain how you plan to respond to these suggestions, as well as the other Reviewers' comments.

I look forward to reading your responses.

Best regards,

Florent Domine

Editor

**Citation**: https://doi.org/10.5194/egusphere-2024-1012-EC1
* * *
**Response to RC1** [egusphere-2024-1012]

Black: Reviewer comments. Blue: Authors' response.

**RC1**: Anonymous Referee #1

This manuscript is a pure modelling study (following a recent study of the same type by the same author) of the diffusion through vein and grain boundary in addition to slow diffusion in ice. Initially, larger than expected diffusion (excess diffusion) has been observed in several cases; those cited in this study are GRIP Holocene, WAIS deglaciation and EPICA Dome over Marine Isotopic Stage 9. Studies of grain boundaries and vein contributions to the diffusion were already performed, e.g., in the book chapter of Johnsen et al. (2000) cited in the present study. The present study wants to go one step further compared to Johnsen et al. (2000) and following his recent paper to show the expected isotopic patterns to be observed at the grain scales for a range of parameters.

Thank you for your review. It is useful to read this summary, and your comments and suggestions in this report are valuable.

The main problem is that no observation is provided and even if laser ablation techniques are progressing, we are still far to the point where such observation can be done and it is also not clear that observations will actually be possible in a near future (see following comments). As it stands now, this study is focused on the resolution of the differential equations for diffusion through ice, grain boundary and vein in cylindrical coordinates. This is a serious calculation work giving the expected changes of patterns with changing parameters. I do not see how it can really be used by others as long as no observation is provided and there are no clear other scientific perspectives for this work. But the study is seriously conducted and well detailed.

I appreciate these comments, and thank the reviewer for positive appraisals on the mathematical calculations undertaken. I agree with the observations that the weak isotopic variations calculated cannot yet be resolved by analytical techniques, and laser ablation methods may have some way to go before being able to measure them.
I feel differently about this situation being a "main problem" with the study. I think that it is permissible for theoretical studies to produce results beyond the measurement capability of the time. In this case, analytical techniques are now actively being developed by researchers, who will benefit from knowing what they are up against – how strong or weak are the predicted/expected signals and what the signals look like, according to theories (which must be tested). As far as I know, prior to the submitted study, no information on this was available, at least not in sufficient detail. More generally, theoretical papers and theoretical results serve an important role in stimulating researchers to take up the challenge to push observational limits, through instrumentation design and innovation. Examples abound in the physical sciences where such interactions happen; observational advances do get made (often in a non-linear manner), and many observations made possible by new technical advances could have seemed far-fetched before those advances. Separately, there are countless publications across science that report theoretical analysis and that do not – in the same paper – measure or observational results pertaining to the phenomenon studied. Thus, I think that the "main problem" being alluded to here lies rather with today's state-of-play of this research area in glaciology and ice physics.
A key angle of the submitted work is that the short-circuiting mechanism of the Nye, Rempel and Wettlaufer, and Ng genre of theories has never been tested – unconfirmed by observations at the grain scale. These theories are at best "working hypotheses", obviously able to match the accelerated signal decay seen in the core sections that they were formulated to explain, but unproven. The theories may be far from reality. We won't know until independent observations are made. I haven't seen this realisation put across in the literature, and it is important that ice-core studies do not automatically invoke vein or grain-boundary short-circuiting as the explanation wherever an isotope record indicates excess diffusion, as if the explanation is firmly established. I think these points form a strong scientific perspective to be offered by a study, and cannot see what other clear perspectives are needed.
Having gathered these thoughts to answer the comments, I think that in a planned revision of the text, I will elaborate along these lines more in the Introduction (perhaps also in the Conclusions) in order to emphasise my study's rationale even more strongly.

The rationale and scope of the study are now substantiated – along the lines described in blue above – in the Introduction, on NL 69-72, NL 75-79 and a new paragraph on NL 103-112.

I concentrate below on comments along the text.

- Introduction : the introduction is well written and summarizes previous findings on diffusion. It is interesting that the author mention the 3 cases where excess diffusion has been identified but I am wondering if the origin of the excess diffusion is the same for all three cases. Indeed in the case of EPICA Dome C, the ice is very deep and old, with relatively large ice crystals and the origin of the excess diffusion may be different than for WAIS and GRIP Holocene. It would be nice to have a discussion on these differences and perhaps discuss the possible mechanisms in each cases.

Thanks for this suggestion. Yes, to help the reader, I can add a passage in the Introduction to sketch the general state regarding the published/existing explanations for those observations, differences, and potential factors, notably, (i) that Pol et al. (2010) invoked fast diffusion in the vein system as a possibility when discussing long diffusion lengths deep in the EDC core; that (ii) Ng (2023) was able to recreate the vertical pattern of diffusion lengths in those core sections of the GRIP Holocene and EDC MIS 19 showing excess diffusion, by using the model with vein-water flow, and finding vein-water flow to be necessary; and that (iii) Jones et al. (2017) explored multiple hypotheses for the WAIS core section showing unusual isotopic diffusion rates, finding that no single hypothesis could straightforwardly explain the observations. Descriptions along these lines will enrich the contextualisation of the study, so I am happy to add them. (I won't be able to discuss those differences and factors on a firm or detailed basis, as we don't know whether or not the theory of vein/grain-boundary short-circuiting is correct, or even roughly correct.)

Done. In a passage near the end of the Introduction (NL 107-112), I outline the state of existing explanations for those core sections showing excess diffusion at GRIP, WAIS Divide and EDC. The gist is that while the short-circuiting mechanism has been invoked as the origin for all these sites, it is not known whether such attribution is correct, as we don't know whether the mechanism really operates. Thus, I am giving readers more context while staying on the main thread of this manuscript. I point them to the papers by Jones et al. (2017) and Ng (2023) should they wish to know more. I cannot give much more detail about those 3 cases of excess diffusion, not without starting to write a different paper whose purpose strays far from the two stated goals (on NL 71-72 and NL 87) of the current study.

- 2.1- the system is clearly described

- 2.2- Figure 2 is complicated to understand and it was only much later during the reading of the manuscript that I could really understand what « high », « medium-high », …, « low » mean. It should be explained in this section as well as in the caption of figure 2.
Thank you for pointing this out. Yes, my descriptive scale for the chosen model values of Db should be better explained at the outset. I can do that, by extending the text on page 8 (around Lines 197-201) and the caption of Fig. 2 caption and coordinating them.

Figure 2 and its caption have been revised. I now use clearer layout and better symbolisation and have improved the key in the figure panel.

Originally, the descriptive scale for Db was given too late (in Sect. 3). I now explain it upfront in Sect. 2.2 (page 9), expanding and splitting the old paragraph (OL 176-204) into 2 paragraphs (NL 208-232, NL 233-248) to walk the reader through the extrapolation of the findings of Lu et al. (2009) to form the chosen sets of Db values for -32 and -52 degC. Both paragraphs are closely coordinated with Figure 2 and its caption. The descriptive scale is explained on NL 244-246 as well as the figure caption and is included in the new Table 2 (old Table 1). Accordingly, I have shortened the passage in Sect. 3.1 about the descriptive scale; see NL 451-452.

Also, in some places the description is not accurate, e.g. « departures from the formulas **by a few times** », « Pre-melting occurs **at high temperature** », « ice-core samples can be **very variable** in these », … and should be precised.

Thank you for prompting me to revisit these passages for their accuracy.

« departures from the formulas **by a few times** » :   The long sentence reads "departures from the formulas by a few times or even an order of magnitude are much smaller than…". This sentence is constructed to describe a hypothetical comparison, where "a few times or even an order of magnitude" indicates an approximate range. I don't think that writing a precise number X in "by X times" is necessary. Therefore I haven't changed the text (NL 155).

« Pre-melting occurs **at high temperature** »:   The quantitative information was deliberately given at the end of the passage, so readers can first get to the physical idea (given after the colon), before having to process detailed numerical information that qualifies what high temperature means. To improve the passage, I now insert "that is" after the colon, to tie the two sentences together more strongly, clarifying the qualification; NL 167-168.

In the short space, it is difficult to be more precise than the numerical information being given. Theoretically, in the *typical* case (but not all cases), the interfacial thickness at low T tends to a negatively-sloped power-law function of the amount of sub-cooling, $T_m - T$; thus, thickness goes to zero as the sub-cooling amount goes to infinity. In other cases, the (theoretical) thickness reaches zero abruptly at a fixed amount of sub-cooling. These complications are the reason why I cite Dash et al. (2006), who expound the details.

« ice-core samples can be **very variable** in these »: I have removed this vague and uninformative phrase.

Line 160 : the author refers to the hypothesis of liquid diffusion at -32°C for Db and say that this estimate will be ruled out but they do not give the value for Db in this case and we can not really see it in Table 1.
Done. On that line (NL 188), I now give the value of Db used by Johnsen et al. (2000). In their paper, that value is written nine lines below their Eq. (25). I do not show the value in Table 1 (new Table 2), for reasons explained below.

In general, it would be nice to make a more clear link between the hypotheses listed in p. 7 and the values for Db explored in this manuscript and listed in Table 1. Perhaps the authors

could take in Table 1 the values of Db from the litterature (or explain how the range of chosen values are linked to previous estimates) and refer to the different papers in Table 1 so that we can make the link between the present study and the previous studies.

Thank you for this suggestion. Extending Table 1 to clarify the links is tricky, as there are no laboratory measurements of Db below –18 °C (Lu et al. (2007, 2009) measured at –1 to –18 °C), and the single Db value simulated by Yagasaki et al.'s (2020) model is for –23.3 °C, while my model Db values lie at still lower temperatures: –32 and –52 °C. Given how scattered these values are on the diffusivity–temperature space, showing all of them (plus Johnsen et al.'s (2000) value) in one table to draw links between them may not work. Currently, I do this graphically with Figure 2. I think what your comments are telling me goes back to the point that Figure 2 should be better explained. Therefore, I plan to improve the text, to guide readers more clearly regarding what Figure 2 shows, how the different data on it should be read, and the interactions and linkages between them, including how extrapolation of the results of Lu et al. is used to choose model Db values, and my descriptive scale for those values. The current text covers these elements too quickly.

As explained in blue above, there are no laboratory measurements of Db below –18 °C, and the values concerned are scattered across the temperature–diffusivity space, so linking them in a table is difficult, and I rely on using Figure 2.

Thus, the key is improvement of Figure 2, and better explanation of how the experimental results of Lu et al. (2009) are extrapolated and used to motivate our test values of Db at -32 and -52 degC. I have done both of these. As described earlier, Figure 2, its caption, the relevant text explaining the extrapolation and its assumptions, and the explanation and use the "descriptive scale" for Db, have all been improved in the revision. Please see NL 233-248, especially 233-235 and 240-247.

- The fractionation explanation is not very clear from l. 162. Which fractionation does the author refer to ? No data for fractionation is given nor its dependency to temperature.

I agree that the passage on l. 162-166 is brisk and might come across as unclear. I will revise it for clarity and detail and indicate fractionation values for oxygen and hydrogen where known (note that laboratory data are incomplete) and give references.

Essentially, if one assumes grain boundaries to be completely liquid, then fractionation will occur at the liquid-to-solid phase transition between them and the crystal lattice within grain interiors; there will be no fractionation where the grain boundaries meet veins, because both of these are liquid. In contrast, if one envisages grain boundaries to be solid-like (almost the same as crystal lattice), then no or negligible fractionation occurs between them and crystal lattice; but in this case, fractionation occurs where the (solid-like) grain boundaries meet (liquid) veins. The paragraph's last sentence on line 165-166 also suggests the possible hybrid scenario where grain boundaries behave as "disrupted lattice", with structural-molecular properties intermediate between solid and liquid. Then fractionation may be expected to occur at both locations. I hope that this description already improves upon l. 162-166. I plan to use what I have written here as the basis for revising the text.

I have rewritten the paragraph, making it clearer. Please see NL 190-198. The paragraph addresses where fractionation occurs, not the fractionation coefficient values, which are addressed in Sect. 2.3 (see my reply below to your comment about OL 369).

- 2.3 – The formulation and resolution follow a classical mathematical approach. Still, because many parameters and variables are involved in this resolution, it would help to have somewhere a table gathering the different parameters and variables, explaining their meaning and giving their values. As it is now, it is difficult to read.

Done. I have added Table A1 to list and describe all key mathematical symbols (except a few peripheral ones) and a sentence at the beginning of Sect. 2.1 (on NL 119) pointing to it. The table is long, so I put it in the Appendix.

It is also certainly possible to have the details on the resolution (e.g. the sections 2.4, 2.5 and 2.6) in an appendix and go directly to the results once the problem and the way to solve it has been defined.

Thanks for this suggestion. I have considered this, and prefer to keep these sections in the main text. Sections 2.4 and 2.5 are central to the description of the mathematical problem (for H) being solved, especially its formulation as eigenvalue problem; and H(r, theta) defines the isotopic fields shown and analysed in the Results sections. I doubt if these sections could be moved to the appendix. If I move them, I end up having to explain the same essential things at the beginning of the Results (section 3). For Section 2.6, I need to cater for readers interested in how the problem is actually solved, so I prefer keeping it in the text; doing this also maintains flow from 2.4 and 2.5. By analogy to experimental studies, I feel that moving sections 2.4, 2.5 and 2.6 to the appendix would be like moving their Experimental Methods section (detailing critical steps, choices and assemblage/units of the new technique that enabled the advance) to behind the text. I don't prefer such a move unless the material is really peripheral.
As explained (in blue) above, I have decided not to move these sections to the appendix.

-l. 369 : no reference nor explanation for choice of the fractionation coefficients are given
In the revised passage (NL 277-279), I now describe the origin of the assumption of setting the fractionation coefficient $\alpha$ to 1, giving three references and coefficient values at T = 0 degC for oxygen and hydrogen. Lack of knowledge on the temperature dependence of $\alpha$ at sub-zero temperatures is pointed out; I added footnote 2 to elaborate on this context. The assumption of $\alpha$ = 1 traces to Rempel and Wettlaufer's (2003) theory, and their theory of isotopic short-circuiting is one of the four theories being proposed for grain-scale testing – see the 4th paragraph in the Introduction (NL 73-79).

Uncertainty whether $\alpha$ deviates negligibly or substantially from 1 at -32 and -52 degC is an outstanding scientific problem, and I refrain from making conjectures about its potential temperature dependence based on the behaviour at other phase boundaries or based on the behaviour of other materials. ($\alpha$ at sub-zero temperatures is known for the ice-vapour phase boundary in studies of atmospheric science, but we are talking about the solid-liquid boundary here.)

- l. 373 : I do not see why studying signal wavelength of 5 mm ? Such signal is not detectable in ice core, even in CFA because of mixing in the system – at best the resolution could be 1 cm which will not enable capturing such signal. Also the following discussion (section 3.1) and display of the results (figure 4 and all figures after figures 4) with a wavelength of 2 cm is not very realistic. Also because firn diffusion occurs with a diffusion length of several cm, it would be more realistic to discuss signals of wavelength 10 cm at least. This is a major

limitation of this study and it is important to address it for more realism and potential use of this study for ice core people. Some figures are shown in the supplement (showing much less excess diffusion) but not really discussed in the main text.

Thank you. This is an important and valuable comment. I agree – it is much more useful to illustrate the study by analysing results for 10 cm or similarly long signals. The original choice of 2 cm was suboptimal. It is easy for me to update the text and figures to showcase findings for a longer signal, e.g. 10 cm or 12 cm, because I have the relevant results at hand (no new computation is needed; I had computed results across the parameter space) and the code and workflow to make Figs. 4 to 11 for any parameter combination; also, the conclusions of the study are not hinged on the 2 cm signal wavelength.
In the update, Figures 4–11 and the interpretations and conclusions drawn from them will in fact change only slightly. This is because the pole and spoke patterns, their transitions and excursion widths, and the parametric controls on their transitions, are mostly unchanged. The reason is that the wavelength affects the variation amplitude (as you pointed out) but not the patterns; examples already showing this are Supplementary Figures S3 – S6 for an 8 cm long signal (cf. Figs. 10 and 11). At longer wavelength, the patterns will still have higher variation amplitudes – and become more detectable – if vein-water flow occurs. Therefore, the inferences regarding instrumental sensitivity (e.g. on page 26) will probably be adjusted but not change drastically. In the revision, in various places I will of course modify the text for emphasis, order, or numerical detail, to coordinate with the changes.

Thank you again for your suggestion. All these changes have been made. The modelling throughout Sect. 3.1 is now illustrated with results for a signal wavelength of 10 cm. The choice of 10 cm (not a longer wavelength) is explained in a new paragraph in the opening in Sect. 3, on NL 442-450, where I describe various isotopic records in this connection, including corresponding references. Hopefully, readers will understand that testing at 10 cm isn't unreasonable or unrealistic. In Sect. 3, Figs. 4–11 have also been replaced by the corresponding new versions for $\lambda$ = 10 cm, and the associated text and captions updated to reflect the changes. As expected (anticipated in blue above), the isotopic patterns have hardly changed; the pattern amplitudes have decreased, as reported in the original manuscript; but vein-water flow still amplifies them to make them more detectable.

>> why studying signal wavelength of 5 mm ? The wavelength ($\lambda$) range extends to small values (as low as 5 mm) to allow the enhancement factor $f$ to be determined as a function over the parameter space of $\lambda$ and $w$ in Sect. 3.2, where it is compared to the earlier results of Ng (2023). (It would be awkward to study a part-surface that misses out the lowest several centimetres in $\lambda$.) I now signpost this on NL 438.

Some figures are shown in the supplement (showing much less excess diffusion) but not really discussed in the main text.

Since the modelling is now illustrated with results for signals at a wavelength of 10 cm, I removed the original Figs. S3–S6 (which issued results for the 8 cm wavelength); and, for completeness and partly in response to the other reviewer interested in grain-size effects, I have made the following changes/additions to the **Supplement file**:

- new Figs. S2 and S3: these are the Original Figs. 4 and 6 for $\lambda$ = 2 cm. I retain them here (but move them to the supplement) to demonstrate the *stronger* pattern amplitudes as $\lambda$ is reduced from 10 cm. Note that the Dye-3 ice core shows isotopic signals as short as 2 cm (see NL 449-450).

- new Figs. S4 and S5, showing the archetypal patterns for non-zero vein-water flow at -32 degC and -52 degC. These are the counterparts to Figs. 9 and 10. The wavelength used in these figures is $\lambda$ = 10 cm.

- new Figs. S6–10, showing the results of repeating the model runs in Figs. 4, 6, 9 and 10 for a larger grain size b = 5 mm and the dependence of enhancement factor f on b (all at $\lambda$ = 10 cm). The other reviewer is interested in seeing these.

- l. 376 : intermed**i**ate
Corrected

- l. 408 and after : some terms need to be better explained : « vertical stretches », « stretch transition », « transitions » (from what to what ?), « hole type», « spoke type»

"Pole" and "spoke" are now defined on NL 463-464.  Hole is no longer used; instead, I write "reverse pole" (e.g. NL 479).

"Vertical stretches" is rewritten as "depth intervals", e.g. NL 481-2.

"Stretch transition / transition(s)" rewritten as "the transitions where the difference in $\delta$ between vein and interior changes sign" on NL 484-485, which serves as the definition of "transitions" for the rest of this paragraph. Elsewhere, this type of transition is clarified with similar wording (e.g. NL 497, 581-582).

- l. 447 : The sentence lacks a word.
I have reworded the sentence to avoid the "this" construction, and added "will" to the next sentence; please see NL 534-535.

- l. 464 : when should we have vein-water flow – please explain this case a little bit
On NL 438-441, I have now added the relevant description, referring to Nye and Frank (1973), who estimated vein-water velocities associated with percolation through ice sheets. I mention that the vein network may be blocked, so it should be clear why we explore vein-flow velocities reaching down to zero. On NL 438, I also indicate that actual vein-flow velocities have not been measured. The reference list now includes Nye and Frank (1973).

- when discussing enhancement factors in this study (e.g. figure 6), it should be compared to what has been measured (if possible on signals with similar wavelength in the data and modeling approach).

I am not sure that this study is the right place for such comparison. On the observational side of the subject, quantification of the diffusive smoothing rate on ice-core isotopic signals is done via the "diffusion length" estimated from the (Fourier) power spectral density of signals. An entirely different study is necessary for comparing theoretical-predicted

enhancement factors and enhancement factors estimated from observations. Such study would need to simulate the diffusion-length profiles down core. Actually, this is the subject of the second half of the paper by Ng (2023); see their Section 4 and their Figs. 8–11. In the current manuscript, I might briefly outline this matter and refer readers to that paper.

As explained in my initial reply (in blue above), comparison is possible between the *diffusion-length estimates* retrieved from ice-core isotopic records by spectral analysis of their signals (this constitutes "measurement") and *theoretical diffusion lengths* simulated from the down-core variation of enhancement factor *f*, at an ice-core site. Such analysis is a substantial piece of modelling that involves the diffusion-length theory of Johnsen (1977) and it must account for parameter variations (e.g. T, b) down-column at the site of interest. An example of such work appears in the second half of the paper by Ng (2023).

As far as I am aware, there has been no direct/local measurement of the diffusivity enhancement factor f for ice-core samples at any specific depth. f isn't the same as the diffusion length; it is related to the *rate of change* of diffusion length or *rate of change* of signal amplitude (e.g. for annual signals). Retrieving it in a robust manner requires mathematical inversion of the empirical diffusion-length profile. There is another way of finding f for a specific part of an ice core -- to monitor and measure how fast its isotopic signals smooth out in time in the laboratory; such experiments are possible but take a long time, as noted on NL 63-65 and NL 817-819.

I appreciate your interest in seeing such comparisons, but I think that these analyses fall outside the scope of the current manuscript.

- When showing isotopic patterns in the figure, the unit should be provided

Yes, I will clarify in the text in Section 3.1 and several captions (notably Figs. 4 and 5) that the unit shown in the figures is dimensionless, and elaborate on the reason. As explained in Section 3.1, the magnitude of the isotopic variations across a given isotopic pattern is scaled to the amplitude of the vertical bulk signal inducing it. Thus, its absolute magnitude (in per mil) can be known only if the absolute amplitude of the vertical bulk signal is known; i.e. the former changes proportionally to the latter. Real bulk signals can have a variety of absolute amplitudes --- there isn't a fixed value. The normalisation on H (prior to plotting) ensures that the figures show a unit (dimensionless) amplitude for the bulk signal, so that scaling can be used.

In final paragraph of Sect. 2.6, the normalisation procedure is now clarified in a new passage (NL 417-424). There, I explain that the patterns and isotopic variations reported later (in Sect. 3.1) are dimensionless, scaled to the signal amplitude at the mid-grain interior position (r = 1, theta = L/2) in the respective model runs. Also, I now repeat the idea of "dimensionless" on NL 465 where the Results (Sect. 3.1.1) starts to explain the findings from Figure 4. I also refer to "dimensionless" or "dimensionless unit" in the captions of Figs. 4, 6 and 8.

- l. 634 : I do not really see how a signal of 2 cm wavelength can be well determined by CFA, see comment above.

Please see my reply above to your comments regarding l.373. I will update the text to illustrate the analysis with a longer-wavelength signal.

As mentioned above, the model results are now illustrated with results for 10 cm signal wavelength. The whole of Section 4.1, including the sentence about CFA (NL 747), is now contextualised on these results for 10 cm (not 2 cm) wavelength.

- l. 670 and below: the author correctly notes that it is really difficult to find ice to test this effect since the isotopic signal usually measured has a wavelength much higher than 2 cm and for short wavelength, the amplitude of the signal is accordingly small.

Thank you for this observation. In Section 4.1, the text has been realigned with the context of a longer signal wavelength of a decimetre. All passages involving the isotopic pattern amplitudes and the measurement sensitivity required, where relevant, have been revised, starting from mid-way in the third paragraph of the section (NL 734 onward), through to the second-last paragraph of the section referring to ice samples from GRIP, WAIS Divide and EPICA Dome C (NL 794-806).

Importantly, although the predicted pattern amplitude at $\lambda$ = 10 cm is small when $w$ = 0 (no vein-water flow), requiring a measurement sensitivity potentially as good as 0.01-0.02 per mil (NL 734-738), several factors mean that stronger patterns may exist in ice samples, making less demand on instrumental sensitivity. These factors include the larger-amplitude bulk signal variations in $\delta D$ compared to $\delta^{18}O$ in ice-core records, the strong amplification of pattern amplitude by vein-water flow (reported in Sect. 3.1), and increase of the pattern amplitude as signal wavelength decreases or as grain size increases. Please see NL 738-744.

- l. 729 : a sensitivity of 0.1 per mill is required but for which isotopic delta ? d18O or dD ?

Thanks for pointing this out. I will clarify this in the Conclusion passage, probably also on page 26, by saying more about the separate cases of d18O or dD, in view of their different ranges of variation observed on CFA records.

Thanks for reminding me to clarify the distinction of d18O and dD. The conversion from dimensionless to dimensional pattern amplitude (by using the bulk signal) is first explained in Sect. 3.1.3, on NL 635-637. There, I now clarify that the conversion applies to both species (NL 635). I clarify this also in the Abstract, on NL 20.

In Sect. 4.1, where isotopic measurement sensitivity is discussed, I indicate the distinction on NL 740-744, here inferring also that under the scale conversion, the typically larger variations on dD (compared to d18O) result in high-amplitude patterns for dD.

**Response to RC2**  [egusphere-2024-1012]

Black: Reviewer comments. Blue: Authors' response.

**RC2**: Anonymous Referee #1

**General Comments**

This paper deals with the modeling of the diffusion of stable water isotopes occurring in polycrystalline ice. The overall motivation of this work is to explore mechanisms that could explain the "excess diffusion" of stable water isotopes observed in some ice-cores. Understanding and quantifying diffusion of stable isotopes in ice cores is important as they are used as proxies for the reconstruction of past temperatures and post-deposition effects need to be quantified for a fine interpretation of these paleo-records.

Specifically, the paper models the combined impact of liquid water veins and crystals grain boundaries in the diffusion of water isotopes and to quantify the enhancement of the smoothing of isotopic signals (compared to a scenario considering diffusion through the ice phase only).

To my view, the paper offers three main points:
- The equations to model an idealized polycrystalline ice, including a water vein and grain boundaries, and a numerical methodology to solve them.
- To demonstrate the formation of specific spatial patterns, around the water vein and crystal boundaries, which constitute a robust signature of the proposed mechanisms to explain excess diffusion.
- To quantify the role of water veins and grain boundaries to the enhancement factor of diffusive-smoothing.

The paper is well constructed, follows a sound methodology, and offers valuable results for the interpretation of ice cores. I therefore think it is suited for the The Cryosphere and suggests minor corrections.

Thank you again for your review and these appraisals, and for providing a detailed set of comments and suggestions to help me improve the manuscript.

My main remarks are:

1 – I think the paper would benefit from a more systematic use of citations to published literature. It would help the reader not familiar with the overall scientific context and with the techniques used in the paper. Some examples are given in the specific comments below, but for instance one example that comes to my mind is L249 that refers to "past theories" without mentioning them (I guess the author meant Rempel and Wettlaufer et al., 2003).

Yes, the submitted text did skip some references and use umbrella terms for them in places to save space. I can revise the text to be specific in those instances. (I have noted the items in your "Specific Remarks" below about references.)
Thanks. I now write out the references as far as possible; e.g. NL 73-74
(what "the four theories" on NL 75 and "these theories" on NL 76 mean should be clear).

>> L249 that refers to "past theories" without mentioning them (I guess the author meant Rempel and Wettlaufer et al., 2003). I now reference the the relevant studies on NL 299-300, placing their reference bracket in a place to avoid interfering with Eq. (13), which contains an azimuthal ($\theta$-) dependence not considered by those studies.

**2** – The author assumes equilibrium fractionation at the interfaces between the grains boundaries, the vein, and the ice (L230). From what I understand, this assumption simplifies the system of equations (as Nv and Nb are now direct functions of Ns at these interfaces). Is it a common assumption for the diffusion of isotopes in polycrystalline ice? Also, since there is transport in the problem, I think that non-equilibrium fractionation could be possible. If so, what could be the impact on the diffusion of water isotopes? Can we justify that it is small?

Equilibrium fractionation was assumed in Johnsen et al.'s (2000) study, and assumed in the equations of Rempel and Wettlaufer (2003), which form the backbone of this part of my model – the formulation on L230 is closely analogous to their formulation. No fractionation was considered in Nye's 1998 model.

The possibility of non-equilibrium fractionation is interesting generally. However, I don't think that I will be elaborating the text to explore it. There are two reasons:
    (1) I don't think that current knowledge allows us to address the process properly as a detail in this study. Experimental data are lacking on non-equilibrium fractionation at the ice–water interface. I looked and struggled to find any relevant literature. Borrowing concepts or results from other fields, e.g., from isotopic studies on igneous and metamorphic rocks or from other phase boundaries (ice–vapour or water–vapour) involve bold and uncertain extrapolations to the ice–water system, and stacking of more theories on top. Those theories will have still more assumptions behind them.
    (2) More importantly, I don't think that the consideration of non-equilibrium fractionation aids this study. The purpose of the study is to bring together existing theoretical elements into a single representative model, and calculate results from it to inform observational testing of those existing theories. The "elements" referred to here include the veins considered by Nye (1998) [who didn't model grain boundaries], the grain boundaries explored by Johnsen et al. (2000) [who considered veins but didn't couple grain-boundary and vein diffusion], the extension to a finite liquid diffusivity by Rempel and Wettlaufer (2003) [whose model has veins but not grain boundaries], and the vein-water flow studied by Ng (2023) [whose model lacks grain boundaries]. Thus, the combined model is necessarily an extension of the individual models/theories. But the purpose of the study is *not* to build even more complexity into it to capture all conceivable aspects of the physical world, to form a 'kitchen sink' model and put it forward as a complete, robust description of excess-diffusion mechanism. Instead, the study emphasises testing the *existing* theoretical

framework. Even if, under some conditions, non-equilibrium fractionation has a large effect in the system, the study's purpose is still to report predictions from the existing theories.

I will modify L230 to emphasise that equilibrium formulation is assumed in the model and in the theories informing its formulation, so it is clear to the reader.

On NL 276-277, I clarify the assumption of equilibrium formulation and its origin. This assumption was made in the theory of Rempel and Wettlaufer (2003). The 4th paragraph in the Introduction (NL 73-79) now states that their theory of isotopic short-circuiting is one of the four theories being proposed for grain-scale testing, and that we are not developing a more sophisticated model with more processes and complications to be put forward for the testing.

**3** – The paper mainly focus on the role of water vein and grain boundaries, and so little emphasis is put on grain size (which was studied before if I got it correctly). I understand this choice, but I'm still wondering to what extent the enhancement factor depends on the grain size. Section 3.2 refers to Rempel and Wettlaufer (2003) for this effect and mentions that the proposed model predict the same trends. Is it possible to mention them in the paper and to estimate what kind of impact can we expect the grain sizes variations observed in ice cores?

Thank you for this suggestion. Yes, I can summarise the trends from grain-size variations following L556-557, probably in a passage of several lines. In sketching the kind of impact, I will strive to give description that errs on the quantitative side, probably through giving examples of values. I think it is a difficult to go beyond that, given an already large number of parameters in the model (and here we're considering variation in one more parameter). Briefly (not exhaustively), I now cover the effects of larger grain size b on:
(i) the isotopic patterns (pattern magnitude, pattern type, width of the isotopic excursions on them) on NL 577-580 and NL 640-647;
(ii) the enhancement factor f in Sect. 3.2, on NL 669-674.
In these new passages, I refer to the results for b = 5 mm and b > 1mm, given in Supplementary Section S4 in five figures (Figs. S6 to S10). Figs. S6 to S9 show repeats of the model runs in Figs. 4, 6, 9 and 10 with b = 5 mm instead of 1 mm, with other parameters unchanged. Figs. S10 shows the computed dependence of f on b, for b in 1-5 mm in increments of 1 mm (under various conditions). These trends found for f(b) support and illustrate the description given on NL 669-672.

Still on grain size, the abstract and Section 4.1 mention that the patterns of δ correspond to 10 to 50% of the grain radius. Do they really scale with grain radius (i.e. does the 10-50% holds for very large or very small grain sizes or it is valid for millimeter grain sizes only)?

Good question. Yes, they scale with the grain radius, because the calculations are made on the scaled model using a radial variable that has been non-dimensionalised with the grain radius b (Sect. 2.4). This finding holds for very large or very small grain sizes, as long as the grain boundaries are much thinner than the vein radius, and the vein radius is much less than b; see Eq. (19) and the line after it. Strictly speaking, the pattern is scaled exactly to the grain size when the Péclet number representing the vein-water flow velocity doesn't

change. But, as reported in the Results, the pattern changes only slightly with vein-water flow velocity (because noticeable changes occur only very close to the vein).

Thanks for asking me to clarify this aspect. Clarification is now given in Sect. 3.1.3 on NL 640-647. Note that the statement of "on the order of 10-50 %" refers to the order-of-magnitude of the **excursion width**, not to the pattern form/type. Thus we are not talking about precise scaling of the pattern with b. The ball-park estimate of "10-50 %" indeed holds when b is increased, at least as far as b = 5 mm, even though the pattern type changes slightly (NL 642-3). To inform that the new passage, I now report results for b = 5 mm in four new figures (Figs. S6-S9) in Supplementary Section S4. Accordingly, I have not changed the sentence in the Abstract (NL 19), which remains correct.

**Specific remarks:**

**L19** I would rather write things in percents when possible, as I find them more natural to read ("10-50% of the grain size radius" rather than "0.1-0.5 of the grain size radius").
Now written in %; please see NL 19.

**L29** Add references to the use of δ18O and δD in ice cores.

Thanks for this suggestion. I decided against it. This is the opening sentence of the paper; it states a general idea, launching the reader into the introduction. If one wants to reference, there are hundreds of potentially relevant references, and selecting a few examples raises the question of why one emphasises those ones, not others. Above all, I don't think referencing here aids the purpose for the opening sentence (NL 31).

**L32** Perhaps add general references to signal smoothing in ice cores.
I decided against this in the original draft, and keep this decision in the revised manuscript, as the relevant 6 to 10 references will be mentioned shortly – on the following lines and in the next two paragraphs, each for a specific purpose. If I pre-list them here, that would interrupt reading without achieving much, and readers would wonder about the duplication. Basically, the "general references" on signal smoothing turn out to be those specific references that are upcoming.

**L38** Add references to Beyond-EPICA Oldest Ice and the Little Dome C project.
Done, on NL 39-40. I now refer directly to the webpages of these projects, adding items to the reference list in TC style. (I failed to find any journal publication that specifically addresses the work of either project. I decided against (i) referencing general news-pieces and (ii) using oblique references such as papers on old-ice dating in the Dome C region that do not focus on describing the projects.)

**L89** Add references to the use of laser-ablation in ice cores studies and its possibilities.

Done. Malegiannaki et al. (2023) now referenced on NL 100. (Note that Bohleber et al. (2021) did not measure isotopic abundances with their LA technique, so I cannot reference them here.)

**L139-141** References would be helpful here to justify that grain boundaries needs to be inferred from bulk measurements and that they size can greatly increase with impurities. Done. References are given on NL 165-166 for the two ideas. I also revised the phrase "may be much higher in the presence of impurities" to "may be higher in the presence of impurities (Thomson et al., 2013) but is generally expected to depend in complex ways on impurity type and concentration (Benatov and Wettlaufer, 2004)", to convey the complexity about the topic.

**L189** Does "bulk" here refers to the combination of ice and grain boundaries? If so perhaps precise it so bulk is not confused with the ice only (my first thought when I read the paragraph). No change made in footnote 1 on p9. From the Introductory section (NL 80-81) onward, "bulk diffusivity" (or bulk-ice diffusivity) has always meant the diffusivity of the polycrystalline material consisting of the collection of monocrystals and their interfaces. The context here, considering the Hart–Mortlock equation, is unchanged.

**Eqs 3 to 5** I would specify that the sources terms in Eq 4 and 5 also corresponds to BCs for Eqs 3 and 4. Thanks for this suggestion. To help readers, I have revised NL 268-269 to spell out what those three source terms represent.
    Please note that defining them as *boundary conditions* is problematic. For instance, the final term in Eq. (5) *can be* posed as a BC of Eq. (3), but that would be a *Neumann* BC for $N\_s$, and its value isn't actually known, so we can't use this BC. The actual BC to be used to solve Eq. (3) (when it is written in terms of δ in Eq. (7)) is a *mixed* BC derived from Eq. (5) through the elimination of time derivatives described on the line before Eq. (6). The mixed BC is stated in Eq. (10) in terms of δ. Mentioning two different boundary conditions *for the same boundary* – one used, one not used – is unnecessary and can confuse readers. I prefer stating only the BC that is used, in Eq. (10).

**L230** Is there a reference for α~1? Yes, I can provide references. References have been added in the passage on NL 276-279, which gives additional clarification in response to suggestions by the other reviewer.

**Eqs 9 and 10** I would specify here that Eqs 9 and 10 are directly derived from Eqs 4 and 5, under the assumption of equilibrium fractionation at the vein/boundary/solid interfaces. On NL 289, I now indicate that they are derived from Eqs. (4) and (5). I do not repeat the assumption of equilibrium fractionation, which was explained in the earlier passage (above Eq. (6), around NL 277) detailing the derivation.

**L250** Add references to the mentioned past theories. Done; see NL 299-300. The other reviewer requested the same addition.

**L261** The form of the trial solution was explicitly chosen as a dampened traveling wave in Eq. 13 but this sentence could leave the feeling that the migration of the signal has been somewhat proven (rather than assumed). I would reformulate with something along the lines "As shown by the phase angle XX, the trial solution of Eq. 13 includes a potential vertical migration at the velocity XX."

Thanks for this suggestion. I have revised the passage. I now indicate that vertical migration is conditional on Im(zeta) being non-zero; see NL 310-311.

I also propose to replace "downward" by "vertical", as I assume it would depend on the sign of the water flow.

I now write "in the z-direction" (NL 311). "Velocity" accounts for the direction by the sign of its value.

**L282** Doesn't $k_r$ require to be solved as well (along side the H pattern)?

Thank. In this section (Sect. 2.5), I will add a sentence to say that after finding the eigenvalue s^2, k_r is found via Eq. (26) and then used to calculate the enhancement factor f.

I have inserted this detail on NL 350-351. I keep the opening sentence of the section (NL 334, old L282) unchanged because it frames the section.

**L299** It makes physical sense that the $\xi_I$ is zero when there is no water flux (because then the isotopic signal is not traveling), but can it be justified a priori from the system of equation (perhaps already done in the literature)?

If I understand your idea correctly, this aspect has already been treated by Ng (2023), on page 3067 in that paper, 4 lines above their Eq. (10), by symmetry considerations. In the current study, I refrain from repeating this highly technical aspect, as it is far from the thread of Sect. 2.5 explaining the approach to solving the equations.

No change made. As described in blue, Ng (2023) addressed this mathematical aspect.

**L300** Why only the slowest-decaying eigenmode?

I will add a sentence to clarify that the other eigenmodes decay faster, leaving the slowest mode to be the one observed. Therefore we are interested in calculating the slowest-decaying mode.

Done, now clarified on NL 352-353: "[comma] as the other eigenmodes decay faster, leaving this mode to be observed in long time".

**L301-305** Perhaps add references about the mathematical techniques discussed here and examples where they have been used.

I considered this and decided not to add references. These techniques for linear problems go back to the 18th and 19th century.

**L363** I do not really understand the normalization. It is normalized by the value of H at r=1, θ=L/2, and z=0 (the last point missing in the text if I'm not wrong)?

The z-variations have been handled by the exponential in Eq. (21), i.e., in mathematical terms, they are treated as the sum of Fourier modes in *z* in a "separation of variables"

approach to solving the scaled model in Eq. (18). Therefore, as indicated on OL 284, H is a function of r and θ only, not a function of z.

However, your comment is probably made with the amplitude of the vertical signal in mind. I can clarify (around OL 363) that H at r=1, θ=L/2 describes the maximum value of that amplitude, to help the reader.

At the end of Sect. 2.6, on NL 418-424 (expanding Old Line 363 into a new paragraph), I now explain the normalization in detail, including the outcome that the normalised solutions represent pattern amplitudes that are dimensionless, scaled to the vertical signal amplitude at r=1, θ=L/2, and thus scaled approximately to the bulk vertical signal as measured by CFA.

**L373** Why these particular choices for λ and w? I guess they correspond to typical values found in ice core, but extra context and justification would be a plus.

The λ-range is motivated partly by both the wavelengths explored in the earlier models of Nye (1998) and Rempel and Wettlaufer (2003), and partly by typical wavelengths seen on ice-core isotope records. The w-range is motivated by theoretical predictions of the water percolation through ice sheets – Reviewer 1 has asked me to add context about that. Yes, I will add suitable background for both choices in this section.

Thanks. Context and justification now added, on NL 438-441 for w, and on NL 443-450 for λ. As described in blue, the other reviewer requested similar additions.

**Section 3.1** This section presents and discusses a lot of results and is therefore a bit dense to read. Perhaps adding sub-sections could help follows the argument set up by the author. For instance something as

*3.1.1 Impact of grain boundary and water flow on isotopic patterns*
*3.1.2 Impact of temperature*
*3.1.3 Impact of the bulk isotopic signal*

Agree. Thanks for coming up with these headings – this is very helpful. I will consider how to sub-section the material and what headings best suit the different parts.#

Done. I have chosen

3.1.1 *Archetypal patterns at –32 °C: effects of grain-boundary properties and vein-water flow*
3.1.2 *Pattern continuum at different temperatures*
3.1.3 *On pattern detectability*
See NL 460, 588 and 631.

**L402** I would say that "the solution *overall* shows what the axisymmetric theories predict", as the presence of small distortions near the grain boundaries discussed later are a notable difference with previous theories.

Done. The word "overall" added; see NL 475.

**L449** I understand that the phase angle (defined in L261) is constant at the given position z, but is it really equals to zero? Or perhaps the phase angle here refers to the phase of H (tan$_{-1}$(Im(H)/Re(H))).

I have shortened the passage to two words: "in phase". See NL 536-7.

**L494-500** I would move this paragraph on the impact of the non-orthogonality between thin sections and triple junctions later in the text (L510). Here, I found that it cuts the reasoning on the impact of $D_b$ and c on the spatial patterns.
Done. I have moved this paragraph and Old Figure 9 to the end of Sect. 3.1, to the new subsection 3.1.3, renumbering the figure as Fig. 11. Please see NL 648-659.

**L499** "Only some triple junctions may show the archetypal patterns" Is it because the tilt make them unrecognizable or because the pattern itself will deviate from the model (model assumptions being not met in real ice)?
Thanks, the reason is the first one you give. It is now clarified on NL 653-654.

**L539** I would precise that the fact the patterns are unchanged does not presume that the enhancement factor is the same.
Done. Clarified on NL 628-630.

**L544** I do not understand why the bulk δ signal has a scaled amplitudes <= 1 in the case of short-circuiting.
Responding to an earlier point, the normalisation procedure is now explained fully at the end of Sect. 2.6 (NL 418-424). There, I describe why the bulk signal has a slightly lower amplitude than the mid-grain interior signal (i.e. the signal used as the scale in the normalisation). It follows that the scaled bulk signal has a scaled amplitude slightly less than 1. No change is made on Old L544-5 except that I now refer to Sect. 2.6; see NL 638-639.

**L608** Perhaps add references for the impurities.
Not added. That glacier ice contains different kinds of impurities is well known and there are too many potential references to add, and it is not clear why one or several of them should be selected and emphasised.

**L634** Add a reference to CFA technique.
Two references are given now (earlier) on NL 422 where CFA is first mentioned. The items are in the reference list.

**Technical remarks:**

**Title** I would precise that the isotopes in questions are (stable) water isotopes. **L1** I would also mention here that the article discusses stable water isotopes.
Thanks for this suggestion. This is a good idea. I will consider whether to clarify this in the title or in the abstract (or in both places).
I have not modified the title. Instead, I now clarify the matter in the first sentence of the Abstract, by writing "the records of water stable isotopes in ice sheets" there (NL 9).

**L11** I would use "could explain" rather than "can explain".
I decide to write "can" instead of "could".

**L19** Perhaps rephrase to "[…] and variations in δ ranging from 1 to 10% of the amplitude of the bulk isotopic signal. This sets […]"

I prefer the current phrasing, which refers to the order of magnitude. Also, I find it awkward to use percentage for a quantity with the unit "per mil". However, I have extended the sentence to clarify the results applies to both oxygen and deuterium; see NL 20.

**L20** Not sure the mention of the specific technique of laser-ablation mapping is necessary here.

Ok, I have shortened the phrase to "the minimum measurement capability needed to detect them", removing "laser-ablation mapping" from the sentence.

Instead, in the following sentence, I insert "based on laser-ablation mapping" to qualify the experimental procedure which the manuscript describes in Sect. 4.1 and Fig. 13.

**L126-127** "Equation" and "Figures" should be written in full at the start of a sentence. Also both the abbreviations "Eqn." and "Eq." are used in the article. I think TC guidelines require the use of "Eq.".

All corrected. Thanks for spotting these.

**L169** The authors sometimes uses here ≈ when the symbol ~ is used elsewhere to mean what I think is the same idea. Perhaps use the word "about".

Thank you. I have gone through the manuscript to check each instance, making corrections where needed. I write ≈ to mean "approximately equal to", and ~ to mean "on the order of".

**Table 1** If possible perhaps also put the investigated grain boundary thicknesses.

Thanks for this suggestion. I added a new Table 1 to list the grain-boundary thicknesses and their qualitative descriptors (thin, intermediate, thick). The Old Table 1 has been renumbered Table 2. In Table 2, I have added the qualitative descriptors for $D_b$ also.

**L204** corroborates

Corrected

**L206** t is not defined at this point.

On NL 251, the bracketed phrase has been extended to:

(with mean-square displacement of molecules ~ $t^\gamma$, where $t$ denotes time and $\gamma < 1$)

**Eq. 6** I would give the definition of δ a bit above, when it first appears L228.

OL 228 = NL 274.

Thanks for this suggestion. I haven't implemented it, because moving Eq. (6) to the OL 228 would place an unneeded over-emphasis on $N_s$ and $N_{s0}$, requiring them to be defined there, while the thread then needs to backtrack to consider the number densities at the vein and grain boundaries ($N_v$, $N_b$, etc.). The paragraph's flow would be awkward. OL 228

already defines what δ means, and I am happy for OL 228 to work a clean topic sentence for the whole paragraph, without elaborating everything about δ on that line.

**L324** I would rather define BC the first time Boundary Condition is used.
I now write out "boundary conditions" in Eqs. (39) and (40) (on page 16), avoiding this abbreviation.

**L327 and 660** I am not familiar with the use of "/" to mean respectively.
"/" has been removed from these parallel constructions using parentheses. Done also on OL 702.  Please see NL 404 and NL 829-830.

**Figure 4** I It might be a problem from my printing set up, but I cannot easily read the scale and axis values. Perhaps try to increase the resolution of the Figures and to increase the font of the Figures' text. Also the use of dashed could help visualizing the black curve when it covered by the blue one.
Thanks for pointing this out. **I have increased the font sizes of labels and of the numbering on axes and colour scales throughout Figures 4 to 11.**

In Figures 4 and 6, I tried your suggestion of using dashes for each curve where it overlaps with another. Unfortunately it doesn't resolve the issue – the overlap is too close, and verbal description is ultimately needed.

**L545** Remove the hyphen in amplitudes.   Done

**L546** "Underestimation"?
Done; see NL 638.

**L550** Perhaps replace "single-crystal diffusion" by "crystal (or solid) diffusion alone" or by "diffusion in single-crystal".
Changed to "monocrystalline diffusion" (NL 663) for consistency with other parts of the manuscript.

**L640** Is the reference to Ng (2023)?  Yes, year added; see NL 753.

---

## Author Response (AR2)

**Response by the author detailing the revision of manuscript "egusphere-2024-1012"**

15 August 2024

Dear Editor of *TC*,

Thank you for considering the revised manuscript and for posing a further idea for me to consider. Please find below my point-by-point reply to your comments. I took the opportunity to go through the manuscript once again, to spot any glitches. I made a few small changes to correct typos and refine local wording, as listed below, and have uploaded the corresponding revised manuscript to the system. I uploaded also the Supplement file again, even though it has not changed.

Best wishes,
Felix Ng

--- --- --- ---

**ED:** Editor comments

Dear Author,

Thank you for your thorough revision of your manuscript, and your careful and adequate responses to the Reviewers comments. Your revised version is almost ready to be accepted but I would like you to consider addressing the following point, if possible. Line numbers below are those of the tracked changes version.

Line 112, you mention "excess diffusion apparently occurs in those sections and not others". Lines 169-170, you briefly allude that impurity content would enhance grain boundary and triple junction thicknesses, favoring excess diffusion. Interesting data on this aspect is given in Thomson et al. (2013), whom you cite and note that they studied ice "under different conditions". This is a rather vague statement. Isn't the important point "under increasing impurities concentrations", as detailed in Thomson et al.? Further, in your discussion, lines 935-936, you come back to the possible relationship between excess diffusion, and increased impurities contents that would thicken veins and turn on excess diffusion.

Could not all this be substantiated with data? Chemical analyses for most cores you discuss are available. Would it be possible to investigate here a possible correlation between excess diffusion and impurity content? Even a partial investigation on a single core would be valuable.

Thank you for this observation. I too feel that such correlative study – comparing the pattern of excess diffusion and the impurity records along an ice core – is worthwhile and should be explored more.

A major unknown in such study is how the *bulk* concentration of each impurity (e.g. as measured by CFA analysis) is partitioned in the material between ice crystals, the vein network, and grain boundaries, and how this partitioning varies with depth down-core. The potential influence of impurity concentration on isotopic diffusivity described in those passages (Lines 170 and 200-201 (Thomson et al, 2013) and Lines 813-814, "*impurities migrate to grain boundaries and then to the veins, thickening them…*", and thereabout) specifically refers to the impurities at grain boundaries, not to the impurity bulk total. Indeed, the partitioning is a limitation of the experiments of Thomson et al. (2013); see the caveat mentioned on Line 200: "*impurity concentration at grain boundaries was estimated from the bulk concentration as it cannot be measured directly*".

Therefore, when depth records of excess diffusion and (bulk) impurity concentration are compared, one doesn't know the fraction of impurity concentration at grain boundaries; guessing it is unsound. This greatly hampers interpretation. Although LA-ICP-MS mapping can tell us the location of impurities, very few maps are currently available, only for small surface sections spanning a few cm's. Also, how such maps could be used to calculate the impurity fraction at grain boundaries from the total remains to be researched.

More generally, a correlative study needs to involve other factors besides impurities – temperature, grain size, vein-water flow velocity, and the down-core pattern of blockage/disconnection of veins. The last two factors are again major unknowns, as mentioned in the manuscript.

Jones et al. (2017) attempted a correlative study of the kind you suggested, for the WAIS Divide ice core, to query the origin of the anomalous excess diffusion at $\approx$ 15–18 ka BP. They used the profile of diffusion length estimated from the power spectral density of the isotopic record, and involved the bulk records of several ions (calcium, sulphur, magnesium) and acidity; see their Section 3.3. They could not find any convincing correlation between the profiles that would explain the diffusion anomaly. One reason, described above and briefly alluded in their study, is that we do not know what part of the impurity concentration pertain to grain boundaries (or veins).

Finally, I feel that the topic of seeking to explain observed instances of excess diffusion (with bulk records) lies peripheral to the current study, to the subject defined by its title. To maintain a strong focus for the manuscript, I prefer not to extend it to treat or comment on the topic. The avenue of studying the pattern of excess diffusion alongside ice-core proxies has already been pointed in the Conclusions section of Ng (2023; p. 3079), which I think is a better place for the idea, because half of their paper is devoted to modelling the diffusion-rate depth profiles.

Minor points:

Line 40. A link is required.
Done. Please see Line 38-39 in the uploaded manuscript.

Line 254. Change HCL to HCl
Done. Please see Line 237 in the uploaded manuscript.

I look forward to reading your response.

Sincerely,

Florent Domine

**Minor changes applied to the manuscript:**

20, word inserted: "times"
38-39, weblinks now given here
55, removed a comma (… excess diffusion potentially caused by…)
111, minor rephrasing to "models of the enhancement factor"
        [and deletion of "based on the short-circuiting"]
170-171: Before the Thomson et al. (2013) reference, the vague phrase "under different conditions" has now been deleted. The phrase is redundant as their study is described later on Line 200-203.
188, changed "as" to "to be"
190-191, I removed some unnecessary definite articles here

194/195: I have swapped "latter" and "former". (In the last revision, these got mixed up when I constructed the text of this paragraph. This is a typographic mistake. The model has not changed.)

237: "HCL" corrected to "HCl"

493: I simplified this sentence in the caption of Fig. 4

525, typo corrected: "gradents" >> "gradients"

528, missing hyphen inserted

643-645: I edited this long sentence slightly to manage its items better.

672, rephrased "the dependence" to "this dependence"

724, added missing year "(2009)"

751: I changed "give" to "might give"

771, "weakest" changed to "weaker ... than elsewhere"

831, I spelled out "it" as "the section" for clarity

844, idea clarified by writing "fast HCl diffusion" instead of "fast diffusion"

866, word inserted: "times"